# Snow depth mapping with unpiloted aerial system lidar observations: A case study in Durham, New Hampshire, United States

Jennifer M. Jacobs[1,2], Adam G. Hunsaker[1,2], Franklin B. Sullivan[2], Michael Palace[2,3], Elizabeth A. Burakowski[2], Christina Herrick[2], Eunsang Cho[1,2]

[1]Department of Civil and Environmental Engineering, University of New Hampshire, Durham, NH, 03824, USA
[2]Earth Systems Research Center, Institute for the Study of Earth, Oceans, and Space, University of New Hampshire, Durham, NH, 03824, USA
[3]Department of Earth Sciences, University of New Hampshire, Durham, NH, 03824, USA

*Correspondence to*: Jennifer M. Jacobs (Jennifer.jacobs@unh.edu)

**Abstract.** Terrestrial and airborne laser scanning and structure from motion techniques have emerged as viable methods to map snow depths. While these systems have advanced snow hydrology, these techniques have noted limitations in either horizontal or vertical resolution. Lidar on an unpiloted aerial vehicle (UAV) is another potential method to observe field and slope scale variations at the vertical resolutions needed to resolve local variations in snowpack depth and to quantify snow depth when snowpacks are shallow. This paper provides some of the earliest snow depth mapping results on the landscape scale that were measured using lidar on a UAV. The system, which uses modest cost, commercially available components, was assessed in a mixed deciduous and coniferous forest and open field for a thin snowpack (< 20 cm). The lidar classified point clouds had an average of 90 and 364 points/m$^2$ ground returns in the forest and field, respectively. In the field, in-situ and lidar mean snow depths, at 0.4 m horizontal resolution, had a mean absolute difference of 0.96 cm and a root mean squared error of 1.22 cm. At 1 m horizontal resolution, the field snow depth confidence intervals were consistently less than 1 cm. The forest areas had reduced performance with a mean absolute difference of 9.6 cm, a root mean squared error of 10.5 cm, and an average one-sided confidence interval of 3.5 cm. Although the mean lidar snow depths were only 10.3 cm in the field and 6.0 cm in the forest, a pairwise Steel-Dwass test showed that snow depths were significantly different between the coniferous forest, the deciduous forest, and the field land covers (p < 0.0001). Snow depths were shallower and snow depth confidence intervals were higher in areas with steep slopes. Results of this study suggest that performance depends on both the point cloud density, which can be increased or decreased by modifying the flight plan over different vegetation types, and the grid cell variability that depends on site surface conditions.

## 1 Introduction

Over the past two decades, remote sensing methods, providing spatially continuous, high-resolution snow depth maps at local and regional scales, have greatly advanced the ability to characterize the spatiotemporal variability of snow depth over earlier work using snow probes. Spaceborne photogrammetry (e.g. Marti et al. 2016, McGrath et al. 2019, Shaw et al. 2020), airborne laser scanning (ALS) (Deems et al., 2013; Harpold et al., 2014; Kirchner et al., 2014), terrestrial laser scanning (TLS) (Grünewald et al. 2010; Currier et al. 2019), and structure-from-motion photogrammetry (SfM) (Nolan et al., 2015; Bühler et al., 2016; Harder et al., 2016) have emerged as viable methods to map surface elevations with snow-off and snow-on conditions in order to differentially map snow depths.

ALS and TLS both rely on well-established lidar (light detection and ranging) technology. TLS, applied from a fixed ground position, is able to measure snow depth with high vertical accuracy (Fey et al., 2019), and has the advantage of being

relatively low-cost and portable, making repeat observations possible. However, TLS uncertainties are caused by large incident angles, occlusion from hills and trees that can cause data gaps in forested domains (Currier et al., 2019; Palace et al., 2016), and challenges to provide a stable scanner position for the tripod in snow-on conditions (Schweizer et al., 2003). ALS technology such as that deployed on the Airborne Snow Observatory (ASO) (Painter et al., 2016) has the advantage of being
able to cover large areas, but it is extremely expensive and has limited availability and flexibility of deployment, which impacts its use for most studies. ALS also has issues with observation gaps in forested regions (Broxton et al., 2015; Currier and Lundquist, 2018; Mazzotti et al., 2019) but possibly to a lesser extent than TLS (Currier et al., 2019). The typical vertical accuracies from these platforms are on the order of 10 cm (Kraus et al., 2011; Deems et al., 2013) with a relatively low return density (~10 returns/m$^2$) (Cook et al., 2013). These accuracies and densities may not be adequate to observe
spatial variations at point scales (0 to 5 m) to hillslope and field scales (1-100 m) and to detect snow depth changes over short time scales due to single events, densification, wind redistribution, sloughing of snow-off slopes, trapping of snow by vegetation, and forest canopy interception (Clark et al., 2011; Mott et al., 2011; Mott et al., 2018).

SfM can create a digital surface model (DSM) from photographs taken using a standard consumer-grade digital camera.
When using an unpiloted aerial system (UAS), which deploys a camera on an unpiloted aerial vehicle (UAV), SfM is a low cost method that has the capacity for routine snow depth monitoring (Adams et al., 2018; Bühler et al., 2016; De Michele et al., 2016; Harder et al., 2016; Vander Jagt et al., 2015). Reported accuracies range from 8 to 30 cm using UAS SfM (Adams et al., 2018; Bühler et al., 2016; Goetz and Brenning, 2019; Harder et al., 2016; Meyer and Skiles, 2019; Harder et al., 2020). The primary drawbacks of UAS SfM as compared to lidar for mapping snow depth are that the DSM needs to be
georeferenced using ground control points (GCPs) with known coordinates and may require significant manual steps (Tonkin et al., 2016; Meyer and Skiles, 2019), although new techniques are emerging that may reduce field data collection time (Gabrlik et al., 2019; Meyer and Skiles, 2019). Dense canopy or vegetation can reduce performance when snow compresses the vegetation relative to the snow-off imagery or when above-canopy vegetation is falsely interpreted to be the snow surface (Bühler et al., 2017; Cimoli et al., 2017; De Michele et al., 2016; Fernandes et al., 2018; Harder et al., 2016;
Nolan et al., 2015). Canopy effects impact SfM snow mapping capability in regions where snowpacks are masked by dense forest canopies.  The inability to sense portions of the ground/snow surface beneath dense canopies results in fine scale variations in snow depth, such as tree wells, not being accurately represented in UAS SfM snow depth products (Harder et al., 2020).

UAS lidar, a UAV-mounted laser scanning system, has been widely used in forest-related research (e.g. canopy height and forest change detection) (Wallace et al., 2012; 2014) and appears to offer the advantages of both the UAS SfM and lidar for snow depth mapping. UAS lidar also eliminates many of the drawbacks that arise from ALS and TLS systems discussed earlier. However, to date there is only one previous study that estimates snow depth using UAS-based lidar (Harder et al., 2020). Harder et al. (2020) compared snow depth estimates between lidar versus SfM techniques using in-situ snow depth
observations in mountain and prairie environments, focusing on sub-canopy snow, which has been a challenge to measure in the snow remote sensing community. Using a considerably more expensive UAS-based lidar system (~$300K Canadian), they found that the lidar system tends to have lower errors than the SfM to capture sub-canopy snow distributions at moderate depth of snowpack (up to 2 m and 1 m of the maximum depth for mountain and prairie areas, respectively). In this study, we assess the ability of a more modest cost UAS lidar system (~$70K U.S. dollars) to map snow depth focusing on
shallow and ephemeral snowpack (< 20 cm). The pilot study described here serves as a proof-of-concept for providing a high vertical resolution snowpack dataset in open terrain and forests in the north-eastern United States. Snow depth magnitude and variability are mapped and analyzed for differences by land use and slope. The study highlights results from the 2019 winter season that provide insights as to the potential for UAS lidar mapping of snow depth as well as details about the

system, its deployment, and operational and validation challenges. We explore the capability of UAS through the
comparison of contemporary field-based snow depth measurements collected in a landscape containing fields and forests.

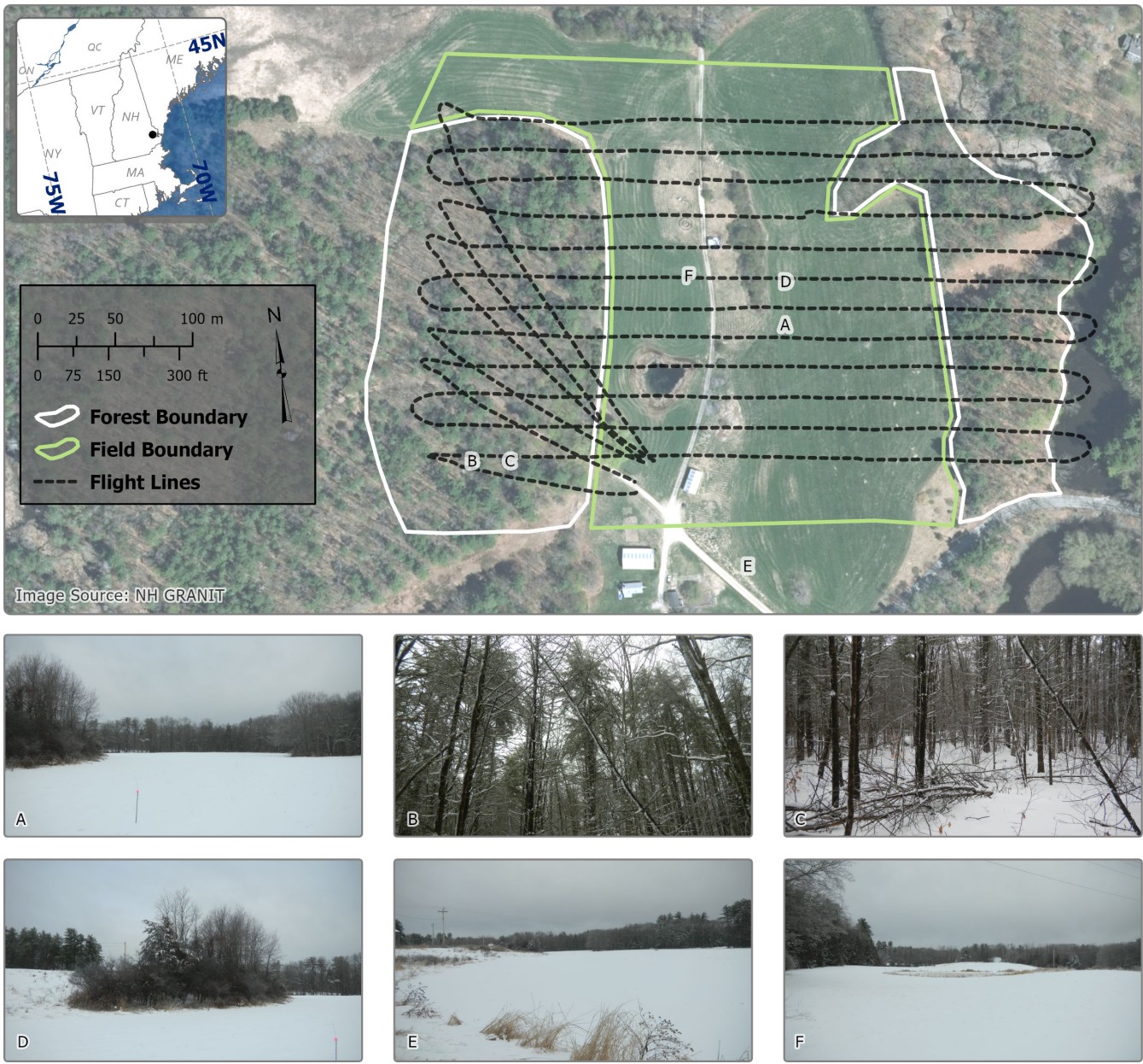

**Figure 1.** 2015 aerial imagery of Thompson Farm, Durham NH showing both forest and field region with lidar flight lines (top). Ground imagery (a to f), collected in December 2019, locations are noted on the top map and show the surface and leaf off forest conditions (bottom panels).

## 2 Site, Data, and Methods

### 2.1 Site

The test flights were conducted at the University of New Hampshire's Thompson Farm Research Observatory in southeast New Hampshire, United States (N 43.10892°, W 70.94853°, 35 m above sea level, ASL), which was chosen for its mixed hardwood forest and open field land covers (Burakowski et al., 2015; Burakowski et al., 2018) that are characteristic of the region (Figure 1). Thompson Farm has an area of 0.83 km$^2$ and little topographic relief (Perron et al., 2004). The agricultural fields are actively managed for pasture grass. The mixed deciduous and coniferous forest is composed primarily of white pine (*Pinus strobus*), northern red oak (*Quercus rubra*), red maple (*Acer rubrum*), shagbark hickory (*Carya ovata*), and white oak (*Quercus alba*) (Perron et al., 2004). There are two logging access roads that run north-south through the pasture and into the western forest section.

## 2.2 UAS Laser Scanning

A series of UAS lidar surveys were conducted over approximately a 0.1 km$^2$ (9.8 hectares) area (430 by 225 m) within the farm during the winter 2018/2019 (Figure 1). Here, we focus on the snow-on flight conducted on January 23, 2019 and the snow-off flight conducted on April 11, 2019. We selected the January 23 flight because it had snowed approximately 11.5 cm with 1.8 cm of snow water equivalent from January 19th to January 20th and the air temperature was persistently below freezing prior to the flight. For the April 11, 2019 snow-off flight, the deciduous component of the canopy and understory were both dormant.

We used an Eagle XF UAS manufactured by UAV America, which carried a small, light-weight lidar sensor (Velodyne VLP-16) suitable for UAS deployment (see Table S1 in supporting information). The VLP-16 is a 16-channel lidar sensor with a 30-degree vertical field of view with rotating lasers that are spaced evenly between -15 to +15 degrees. Each channel rotates to provide a horizontal field of view of 360-degrees. The VLP-16 collects up to 300,000 points per second with an accuracy of +/- 3 cm at a range of 100 m. The sensor was mounted with the vertical field of view parallel with the ground. The payload is equipped with an Applanix APX-15 UAV inertial navigation system (INS), which has 2-5 cm positional, 0.025-degree roll and pitch, and 0.08-degree true heading uncertainties following post-processing. The INS has a measurement rate of 200 Hz, allowing for a timestamp to associate each lidar pulse with the closest data for latitude, longitude, altitude, and perspective information (roll/pitch/yaw), which is required for georeferencing returns.

Flights were conducted to maximize spatial coverage while conserving batteries due to the limited flight time of the Eagle XF (approx. 9 minutes flight time from ascent to descent). Because of the limited flight time, flights were conducted at an altitude of 81 m for greater spatial coverage and multiple return flight lines were necessary for battery exchanges (Figure 1). Automated flights were conducted using UgCS flight planning software. Flight speed was 7 m/s, with a total of 12 parallel flight lines with targeted overlap of 40 percent. A complete survey of the study area took approximately 2 hrs. This includes the time required to calibrate the INS and set-up and break-down the UAS. Because of degrading accuracy at distances >100 m with the VLP-16, returns acquired outside of +/- 30 degrees of nadir view angles in the horizontal field of view were filtered to limit target distance and improve overall accuracy.

Applanix APX-15 INS data were post-processed to a Smoothed Best-Estimate Trajectory (SBET) file using POSPac MMS UAV (v. 8.2.1), resulting in approximately 3 cm positional accuracy for both the snow-on and snow-off flights. Lidar returns were individually georeferenced by synching timestamps of returns from the lidar sensor with timestamps of position and attitude data from the post-processed INS data. Georeferenced point clouds were produced and output to LAS files using Headwall Photonics, Inc.'s LidarTools software. The bare-earth and snow-on point clouds were georeferenced solely using the INS data respective to each flight. The point clouds were not co-registered to each other as there were no reliable common ground control points between surveys. For UAS lidar snow depth surveying, co-registration between point clouds would likely be unattainable due to insufficient common ground control. We determined results would be more meaningful when bare-earth and snow-on point clouds were processed solely relying on the capability of the INS. Boresighting calibration was performed using returns from the first two parallel flight lines that were collected in opposite directions (i.e., antiparallel). A roll offset was determined using 10 m cross sections along the flight lines over flat terrain, and a pitch offset was determined using 1 m cross sections across the flight lines over terrain with moderate relief (see Figure S2 in supporting information). Resulting LAS (LASer) point clouds were generated for the entire study area and projected in WGS84 UTM Zone 19N (EPSG 32619). Flight and filtering parameters of the raw point cloud resulted in return densities of approximately 150 returns/m$^2$ for each of the two flights.

## 2.3 Lidar Classification and Gridding

Three-dimensional point clouds were processed using the progressive morphological filter algorithm (PMF), in the lidR package (https://github.com/Jean-Romain/lidR) of R (v. 3.4, Team, 2018) to identify ground returns. Briefly, the PMF operates iteratively on sets of two parameters, window size and elevation thresholds to erode and dilate point cloud data sets to estimate surface topography. The result of the PMF is that non-ground returns (i.e., trees, shrubs, and noise) are filtered out of point cloud data sets, so that only returns from ground surfaces remain. The two data sets, non-ground returns and ground returns from the original point cloud, are coded according to LAS specifications and merged. For a full explanation of PMF, see Zhang et al., (2003). For ground classification, point clouds were chunked into 100 m square tiles with a 15 m buffer on all sides using catalog options in lidR to ensure returns near tile edges were classified. Processing was distributed across 8 computing cores to improve efficiency. PMF was parameterized using a set of window sizes of 1, 3, 5, and 9 m, and elevation thresholds of 0.2, 1.5, 3, and 7 m, which were determined by varying value sets and assessing digital terrain models (DTMs) to determine the parameter sets that produced a visually smooth surface over a dense grid (*sensu* Muir et al., 2017). Following ground classification for each tile, returns within the 15 m tile buffers were removed, and all resulting 100 m square ground classified tiles were merged. The resulting point clouds for each data set included both the classified ground returns and the non-ground returns. Snow-on and snow-off ground point clouds were gridded at 0.1, 0.2, 0.4, 0.5, and 1.0 m spatial resolutions using the average of all grid points within each grid cell (Currier et al., 2019). Gridded products for each data set were forced to the same coordinate grid to generate DTMs as raster files.

## 2.4 Slope and Vegetation Cover Classification and Analysis

The snow-off DTM was used to develop a 1 m resolution map of slope (Horn, 1981). Vegetation cover type (field/forest) was determined from optical imagery. A Canopy Height Model (CHM) was developed by subtracting the DTM produced using ground-classified points from the DSM produced using all lidar points. This results in a digital model consisting solely of canopy heights with no topography. The CHM generation used raster images with a 1 m resolution. The forested area was further classified as coniferous or deciduous for the study region. Within the forested area, the CHM was used to distinguish the upper canopy that did not lose needles/foliage from other forested regions with trees with no leaves using our snow-off survey that was collected with leaf off in the spring. A 3 by 3 maximum convolve filter was used to enhance the edges of canopy crowns and expand smaller regions that might have just one pixel of an intact canopy or a hole in a larger canopy (Palace et al., 2008). A 15 m threshold was used to differentiate between the upper level intact coniferous canopy and canopies that had lost their leaves. CHM pixels that were below this threshold were deemed deciduous canopies (see Figure S3 in supporting information for intermediate figure). The 5.6 ha forested area has a forest type that is 65% deciduous and 35% coniferous.

Once the vegetation forest type was classified, three sets of 5000 points were extracted respectively in the field, in the eastern forest and in the western forest (Palace et al., 2017). At each of these random points, slope, vegetation type (field, deciduous, coniferous), snow depth, and snow depth confidence interval values were extracted. Because of missing values in the raster images, not all random points extracted values. Slope was assigned to one of three categories: 0-10 degrees, 10-20 degrees, and greater than 20 degrees. Because the extracted datasets (i.e., snow depth, confidence interval, and slope) were not normally distributed, the non-parametric Steel-Dwass Method test was used to test for differences. The Steel-Dwass test has been previously used in geophysical work to examine non-parametric datasets (Slotznick et al., 2019). This non-parametric method is useful when sample numbers are large and groups are small, because it allows type I errors to be controlled (Dolgun and Demirhan, 2017).

**2.5 *In Situ* Observations**

A magnaprobe (Sturm and Holmgren, 2018) was used to compare to the UAS lidar survey over two transects. The first transect consisted of 12 sample locations in the field and 5 locations in the eastern forest of our study site. The second transect consisted of 11 sample locations in the western forest. Sample locations were separated by approximately 10 m. The field transect follows the prevailing westerly wind direction with its west side at the foot of a modest depression (approximately 3-4 m below the land further to the west) and the east side transitioning into a wooded area. Following (Harder et al. 2016) and (Bühler et al. 2016), each sample location includes 5 samples in a cross pattern with the four ordinal directions sampled approximately 20 cm from the center sampling location in the cross. The five samples are used to provide a measure of snow depth central tendency and variation over a 0.4 x 0.4 m pixel. Because the magnaprobe GPS has an absolute accuracy of 8 m, a Trimble© Geo7X GNSS Positioning Unit with Zephr™ antenna was used to collect each sampling location's center point with an estimated horizontal uncertainty of 2.51cm (standard deviation $\sigma$ 0.95 cm) and 4.17cm ($\sigma$ 4.60 cm) for the field and forest, respectively after differential correction. Along the same forest and field transects, a federal snow tube sampler was used to collect a single sample of snow depth and snow water equivalent (SWE) at each magnaprobe sample location for a total of 12 field samples and 16 forest samples. SWE was measured by inserting the aluminium tube vertically into the snowpack and a core was extracted and weighed using a spring scale.

An independent study collected soil frost depth from three locations at the Thompson Farm Research Observatory using Gandahl-Cold Regions Research and Engineering Laboratory (CRREL) style frost tubes. The frost tubes have flexible, polyethylene inner tubing filled with methylene blue dye whose color change is easy to differentiate when extruded from ice (Gandahl 1957). A nylon string housed inside the polyethylene tubing affixes ice during periods of thaw. The outer tubing consists of PVC pipe installed between 0.4 to 0.5 m below soil surface (Ricard et al., 1976; Sharratt and McCool, 2005). Prior to the January 19[th] and 20[th], 2019 snowfall event, soil frost was 23.5 to 25.5 cm in the field and 5.5 to 8.5 cm in the west forest.

**2.6 Snow Depth Uncertainty Assessment**

The snow depth accuracy was assessed by comparing the lidar snow depth measurements to the magnaprobe measurements. Here, accuracy is the measure of the agreement of the lidar snow depth measurements relative to the in situ measurements (Eberhard et al., 2020; Maune and Nayegandhi, 2018). Error statistics were calculated and the results were summarized by forest and field locations. At each magnaprobe location, the average and standard deviation of the five magnaprobe samples were calculated. The average lidar snow depth was determined for a 0.4 x 0.4 m cell centered on the center magnaprobe location. The mean absolute difference (MAD) and root mean square deviation (RMSD) were used to characterize the differences between the magnaprobe snow depths and the lidar snow depths.

The one-sided width of the 95% confidence limits ($CI_{95\%,+/-}$) for each grid cell's lidar derived estimate of the mean snow depth is a measure of uncertainty. The $CI_{95\%,+/-}$ values are used to compare the reliability of the snow depth estimates among cells. The $CI_{95\%,+/-}$ values were calculated using each grid cell's bare-earth and snow-on pooled sample standard deviation ($s_d$) and the number of bare-earth and snow-on lidar returns (n and m respectively) (Helsel and Hirsh, 2002).

$$CI_{95\%+/-} = t_{crit} s_d \sqrt{\left(\frac{1}{n} + \frac{1}{m}\right)} \qquad (1)$$

where $t_{crit}$ is the critical value of the Student's t-distribution with a significance level of 0.05 and $s_d$ is the cell's pooled sample standard deviation which was calculated as

$$s_d = \sqrt{\frac{(n-1)s_{0ff}^2 + (m-1)s_{on}^2}{(n+m-2)}} \qquad (2)$$

where $s_{on}$ and $s_{off}$ are the standard deviations of the snow-on and snow-off lidar ground return elevations, respectively. The $s_{on}$ and $s_{off}$ values are a measure of the grid cell variability. This variability depends on the lidar instrument's relative accuracy (Maune and Nayegandhi, 2018), which includes intra-swatch accuracy (i.e., precision or repeatability of measurements) and inter-swath accuracy (i.e., differences in elevations between overlapping swaths), as well as surface elevation variations and terrain induced errors (Deems et al., 2013). The contribution from the individual sources of 230 variability was not assessed in the current study.

## 3 Results and discussion

### 3.1 Snow Depth Survey

The snow-on and snow-off lidar ground returns yielded an average point cloud density of 90 and 364 points/m$^2$ in the forest and field, respectively, with 6.7% of the 1 m$^2$ forest cells and 0.03% of the 1 m$^2$ field cells having less than 5 point/m$^2$ (Figure 2). There was a wide range of the point cloud densities (Figure 2b). The highest point cloud density occurred for those cells sampled by both the regular flight lines and the multiple return flight lines conducted for the three battery exchanges. The vast majority of field cells (82%) have more than 100 points/m$^2$. Only 1% of the field cells had less than 25 240 points/m$^2$ and most of those cells were in shrubbery or dense vegetation surrounding the small pond in the center of the study site (Figure 1). In contrast, 41% of the forest cells had more than 100 points/m$^2$ and nearly 20% of the forest cells had less than 25 points/m$^2$ with 8% having fewer than 10 points/m$^2$ (Figure 2b). Only 0.086% and 0.95% of the 1 m resolution field and forest cells, respectively, had no ground returns. The number of points per cell decreases with decreasing cell size (Figure 2a). In the field, reducing the gridded resolution from 1 m to 0.5 m lowers the mean cell return count to 91 points per 245 cell on average. Thus a 0.5 m field cell has approximately the same number of returns as a 1 m forest cell. At a 0.2 m spatial resolution, the mean number of ground returns is 14.6 and 3.6 in the field and forest, respectively.

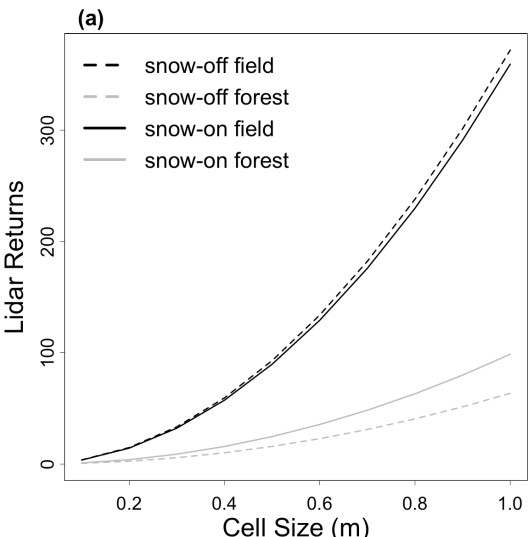
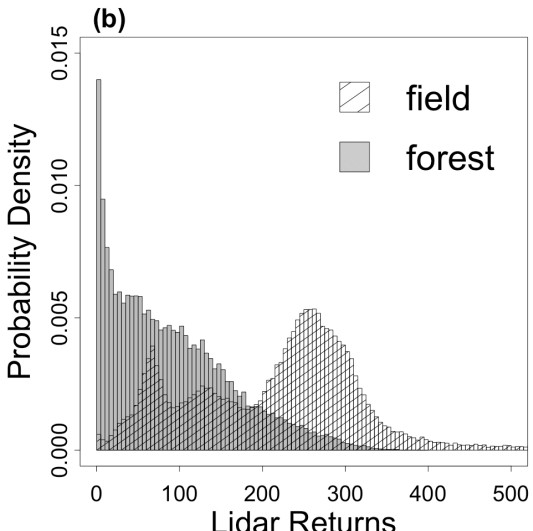

**Figure 2**. (a) Average lidar point cloud density of the ground returns versus cell size by land cover, and snow-on and snow-off state. (b) Probability density function for the number of lidar ground returns by square meter for the forest (gray) and the field (white)

## 3.2 Lidar and *In Situ* Snow Depth Comparison

Based on the magnaprobe snow depth and UAS-mapped snow depth measurements, the accuracy of lidar snow depth measurements differed between field and forest cells (Figure 3). In the field, the mean snow depth from the magnaprobe (12.2 cm ±0.56 cm) was only slightly greater than that from the lidar (11.2 cm ± 0.72 cm) and the MAD and RMSD values were 0.96 cm and 1.22 cm, respectively. In the forest, the mean snow depth from the magnaprobe (15.2 cm ± 2.3 cm) was twice as large as the lidar snow depths (7.8 cm± 6.3 cm) and the MAD and RMSD were 9.6 cm and 10.5 cm, respectively. The mean snow depth from the Federal snow tube was (12.9 cm ±0.71 cm) and (13.1 cm ±1.9 cm) in the field and forest, respectively. There is a notable absolute low bias in the lidar forest snow depth relative to the magnaprobe and snow tube for west forest in particular with the exception of one site.

To provide insight to differences between the forest and field observations, height profiles of classified returns were calculated for 25 m$^2$ square regions centered on all forest (n=12) and field (n=7) study plots from lidar data. Height profiles were averaged for each site type, from here on referred to as mean height profiles (Figure 4). To do this, all lidar returns were extracted from the bounding box of each plot, then the mean elevation of ground returns was calculated within each plot. Return height profiles for each plot were determined by subtracting the mean ground elevation of the plot from each return, then the normalized return elevations were binned in 0.1 m height increments. Within the forests, an average of 2142 and 2889 returns were classified as ground and non-ground, respectively, in snow-free conditions for each 25 m$^2$ plot. Snow-on conditions had a comparable number of ground returns (2218), but fewer non-ground returns (1721). In field plots, an average of 5666 ground returns and 154 non-ground returns in snow-free conditions were obtained for each 25 m$^2$ plot, with 7567 ground returns and 25 non-ground returns in snow-on conditions. Figure 4 also shows that there is a greater range of ground return elevations in the forest as compared to the field. In forest plots, ground return elevations had an average standard deviation of 0.157 m and 0.154 m in snow-free and snow-on conditions, respectively, while in field plots, ground return elevations had standard deviations of 0.058 m and 0.050 m in snow-free and snow-on conditions, respectively.

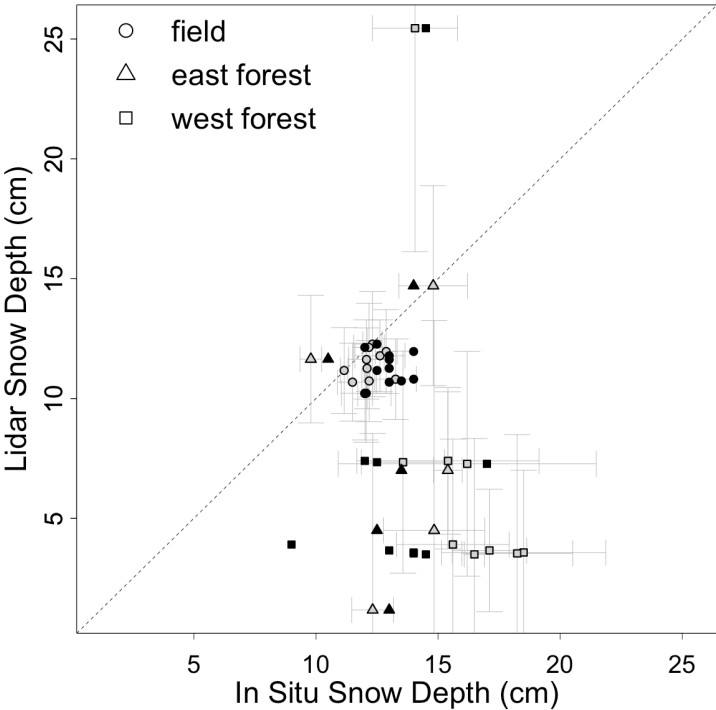

**Figure 3.** Comparison between the magnaprobe (gray fill) and snow tube (black fill) versus the lidar snow depth measurements by location. The mean and 95% confidence intervals were calculated using the five magnaprobe snow depths and the lidar snow depths, equation 1, averaged over a 0.4 x 0.4 m grid cell. Single snow tube snow depth measurements are shown without confidence intervals.

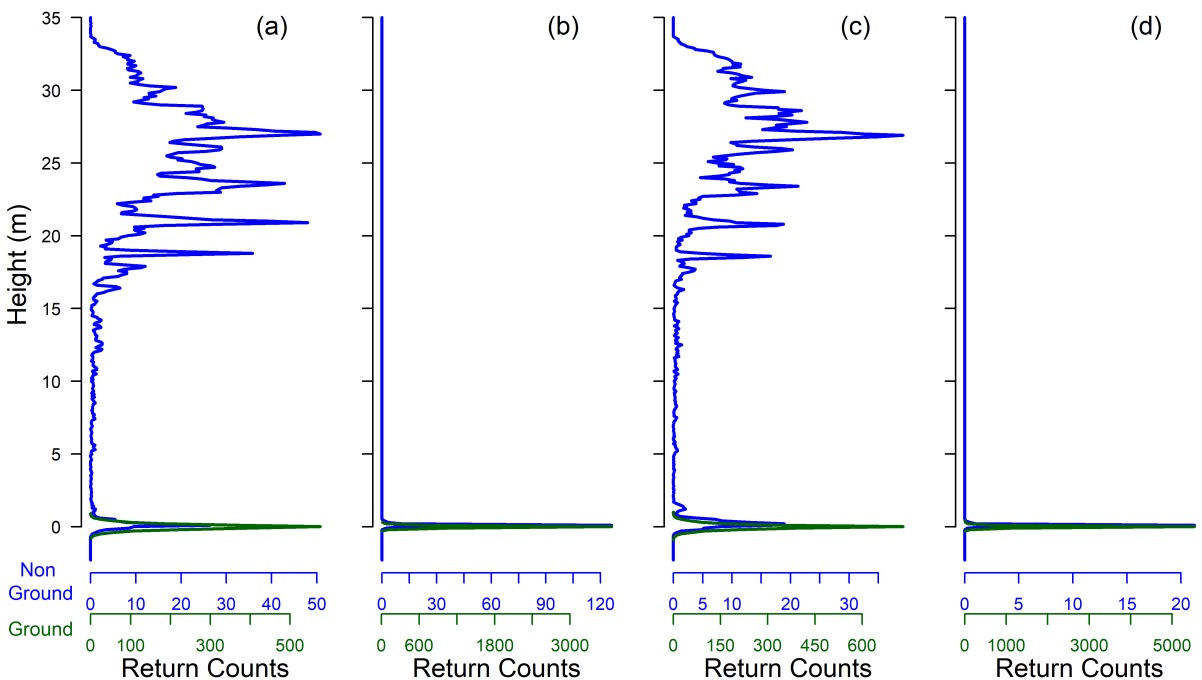

**Figure 4.** Mean height profiles for all ground (green) and non-ground (blue) lidar returns within a 5 m x 5 m region centered on each transect plot in snow-free conditions (a, b) and snow-on conditions (c, d) in forest (a, c) and field (b, d) study plots.

### 3.3 Snow Depth Maps from UAS Lidar

The UAS-mapped snow depth, mapped by subtracting snow-off DTMs from snow-on DTMs, revealed a shallow snowpack whose depth ranges from less than 2 cm to over 18 cm (Figure 5). The mean lidar snow depth was 10.3 cm in the field and 6.0 cm in the forest. Despite the shallow conditions, spatially coherent patterns are readily discernible. The field snowpack depth had higher spatial variability than the west forest snowpack and more spatial organization. In the field, the deepest snow was in the low-lying northeast areas that are sheltered from westerly winds. A relatively moderate and consistent snowpack occurred in southern part of the east field and west of the small pond. The shallowest snowpack was found in the center portion of the field, which is slightly elevated and, unlike most of the field, was not mowed. Lower snow depth at the forest edge distinguishes the field to forest transition. A non-parametric Steel-Dwass test found significant variation for the mean snow depth among the two forest types and field ($p < 0.0001$) (Figure 6a). Figures 6a and 6b also reveal that there are some negative snow depths in the two forest types that is due to the uncertainty of the snow-on and snow-off DTMs.

A pairwise Steel-Dwass test showed that snow depths were significantly different between the three pairs of field and forest types ($p < 0.0001$). When comparing just field and forest as categories, the test also found significant differences for snow depth ($p < 0.0001$). Snow depth was also determined to be significantly different among the three slope group categories using the Steel-Dwass test where regions with a limited slope (Group 1) had more decidedly different snow than steeper regions ($p < 0.0001$) (Figure 6b).

The one-sided confidence interval values of the mean snow depth estimate are remarkably consistent in the field and typically are between 0.5 to 1 cm regardless of snow depth (Figure 5b). Modestly larger confidence intervals occur adjacent to the north-south road where the fields were not mowed prior to winter as well as the northern and southern extents of the flight lines likely due to the reduced sampling density. The forest had an average one-sided confidence interval of 3.5 cm, which was considerably higher than the field. Where the forest is predominantly comprised of deciduous trees, the typical

one-sided confidence intervals of the mean snow depth were as low as 1 to 2 cm. The largest one-sided confidence interval values occurred in the middle of the field where there was dense shrubbery, at the edge of the fields, and in clusters within the forest where the forest sections were dominated by coniferous trees (*Pinus strobus*). The nexus of flight lines in the take-off and landing area resulted in a local area with very high confidence. A non-parametric Steel-Dwass test found significant variation for confidence intervals of the mean snow depth among the two forest types and field (p < 0.0001) (Figure 6c). A pairwise Steel-Dwass test showed that confidence intervals were significantly different between the three pairs of field and forest types and (p < 0.0001). Confidence intervals were also significantly different among the three slope categories as determined using a Steel-Dwass test  (p < 0.0001) (Figure 6d).

## 3.4 Point Cloud Density, Spatial Resolution, and Canopy Profiles

Confidence intervals for the mean snow depths by grid cell were examined in light of the point cloud density and the spatial resolution at which lidar returns were aggregated. The confidence interval width for a mean snow depth of a 1 m$^2$ area decreased dramatically as the lidar point cloud density increased (Figure 7a). Except for the cells with fewer than 10 point/m$^2$, forest cells had larger confidence intervals for the mean depths than field cells for a given sample size. When the density exceeds 25 point/m$^2$ in the field and 50 point/m$^2$ in the forest, confidence intervals were typically 2 cm. The cells with the highest point cloud densities had one-sided confidence intervals of about 1 and 1.5 cm for the field and forest cells, respectively. The field cells with more than 50 point/m$^2$ did not have noticeably smaller confidence intervals, but the increased density did reduce the number of cells with anomalously small confidence intervals. Given the high lidar point cloud density for the field cells, it is possible that reasonable estimates of snow depth can be made at scales finer than 1 m (Figure 7b).

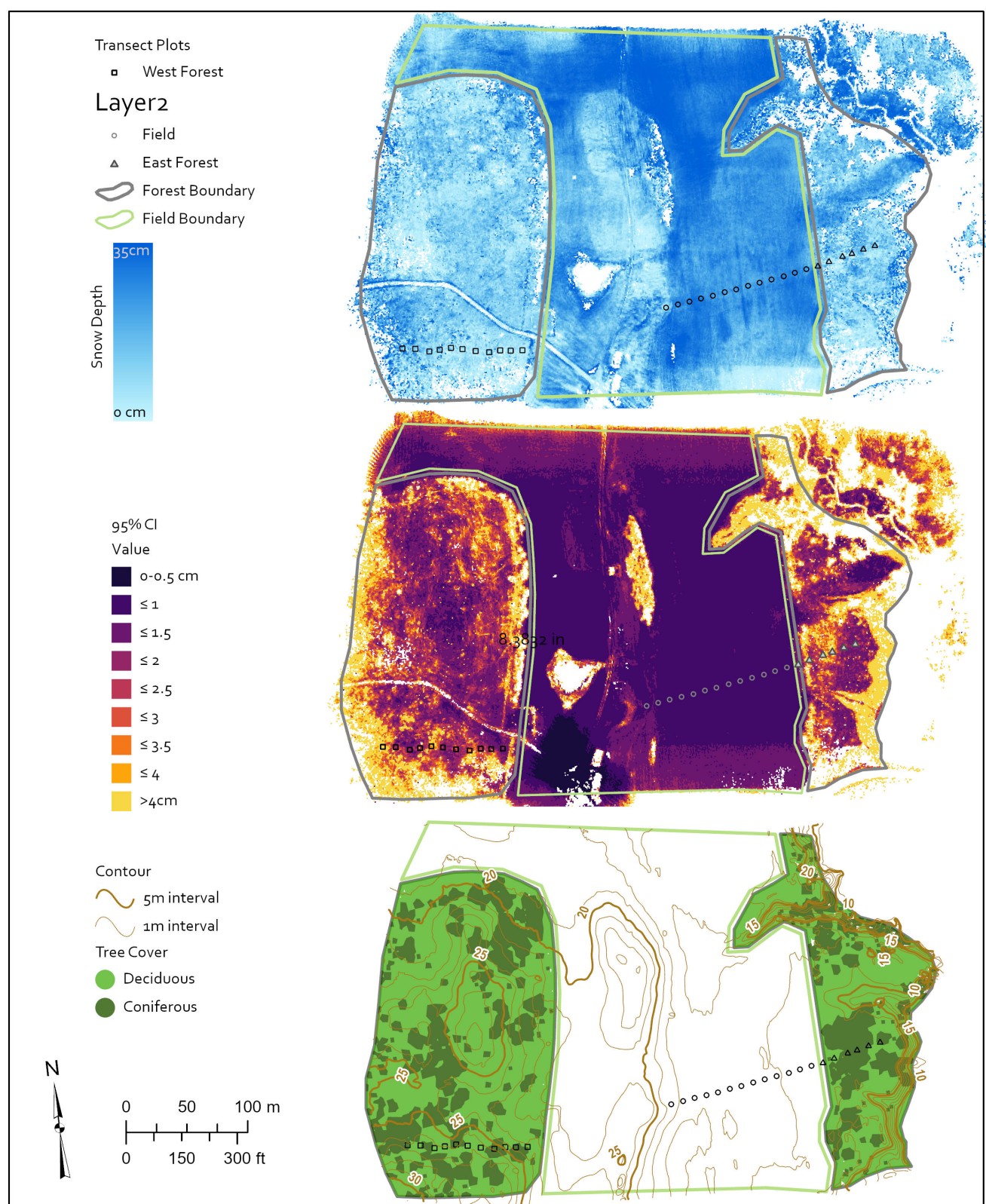

**Figure 5.** Average (top) snow depth values, (middle) one sided confidence intervals, and (bottom) topography and forest cover type. Snow depth and confidence intervals calculated from the snow-on and snow-off lidar point clouds for 1 m$^2$ cells at Thompson Farm, Durham, NH. Topography and forest cover type determined from snow-off lidar point clouds on snow-off flight for 1 m$^2$ cells conducted on April 11, 2019.

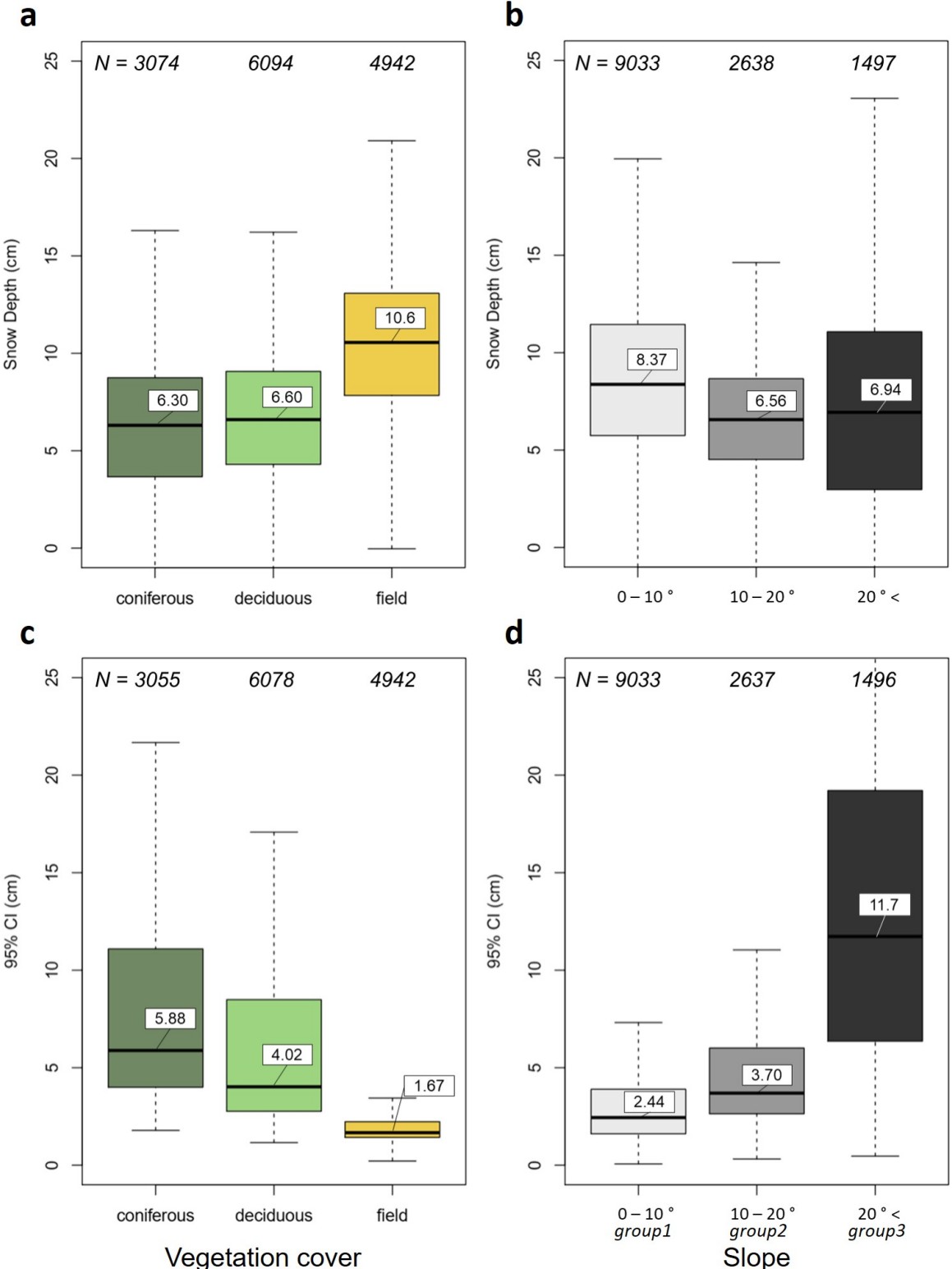

**Figure 6.** Snow depths (a,b) and their one sided confidence intervals (c,d) from the random sample points of the field and forest at Thompson Farm, Durham, NH on January 23, 2019 from the individual cells for 1 m² cells by vegetation cover (a,c) and slope group (b,d). Boxplots show the lower quartile, median, upper quartile, and whiskers with the median value noted. Because of missing values in the raster images, not all random points extracted values and resulted in different numbers of samples points for vegetation cover classes.

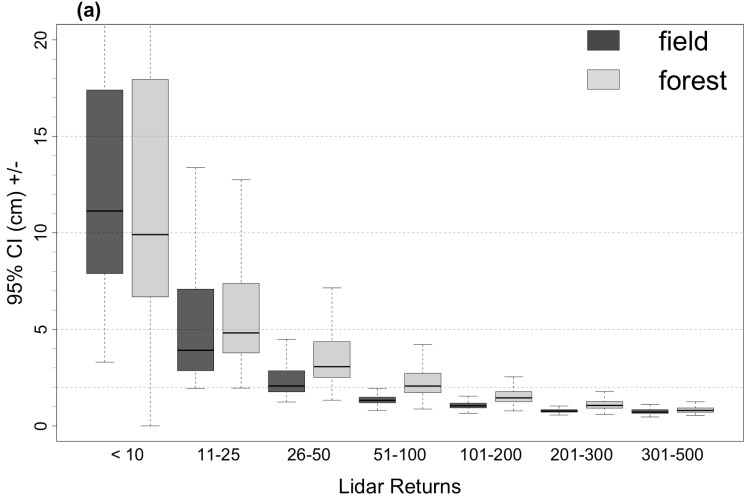

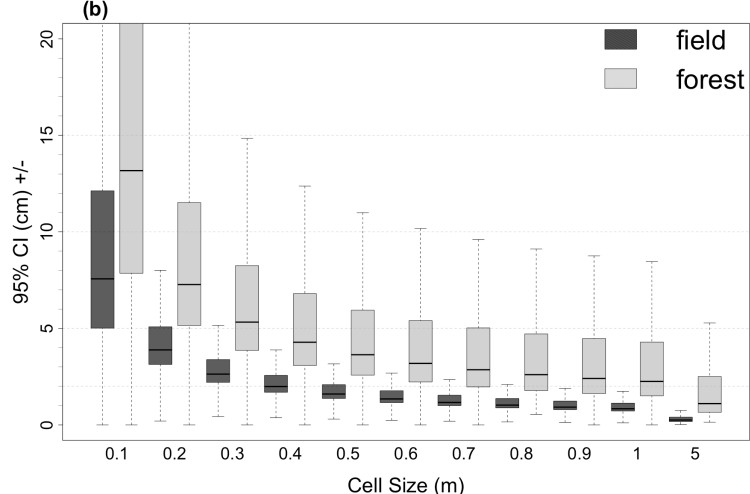

**Figure 7.** One sided confidence intervals of the mean snow depth values in the field and forest at Thompson Farm, Durham, NH on January 23, 2019 from the individual cells for 1 m$^2$ cells by land cover and point cloud density (a) and for grid resolutions ranging 0.1 to 5 m (b). Boxplots show the lower quartile, median, upper quartile, and whiskers.

In addition to the lidar point cloud density, the ability to reduce the confidence interval of the mean snow depth also depends on the ground surface variability within a cell as well as the lidar performance. For this site and its shallow snowpack, the grid cell variability of the ground surface elevation, estimated by calculating the standard deviation of the lidar elevation values, and found to depend primarily on the cell size and, to a more limited extent, on land cover and snow cover (Figure 8a). Snow cover reduced the grid cell variability in the field by about 1 cm, but has a limited effect in the forest. It is possible that the modest snowpack was able to flatten the higher grass in the field, while the forest's vegetation and ground surface features that dominate the grid cell variability were only minimally compacted by the snow. Within the 1 m grid cells, snow depth variability was much lower in the field than the forest (Figure 8b). Both distributions had a positive skew. Typical standard deviations of the lidar surface elevation values within a 10 cm cell were on the order of 1.5 and 2 cm for the field and forest, respectively. That variability doubled for a 20 cm cell. The grid cell variability increased gradually to about 3 to 4 cm in the field, and to about 6 cm in the forest.

Thus, confidence intervals largely depended on the point cloud density in the lidar cloud because the standard deviation of a cell's surface elevation is relatively constant for snow depth resolutions from 0.5 to 1 m (Figure 8a). In the field, reducing the cell size from 1 m to 0.5 m still yields about 100 points/m$^2$ and provides snow depth estimates within +/- 1.5 cm. Because the forest cells required a higher ground return density to capture these snow depths within a 1 cm, any reduction in cell size below 1 m greatly increased the cell mean snow depth's confidence intervals.

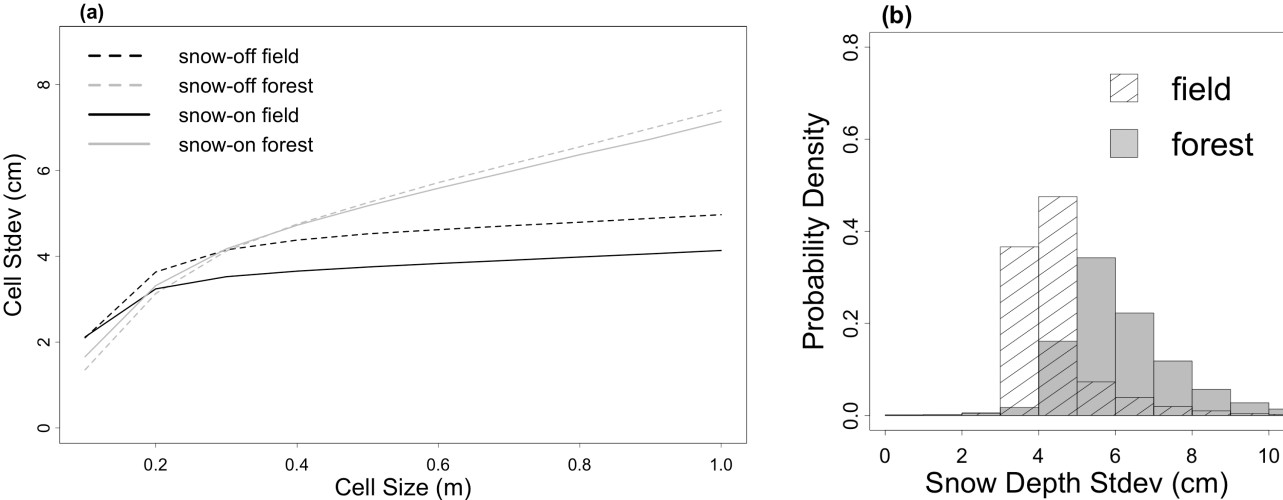

**Figure 8.** Lidar surface elevation standard deviations by (a) cell size and land cover. Cell standard deviations are the average of the individual cells standard deviation values calculated for cells with side dimensions ranging from 0.1 to 1.0 m. (b) Probability density function of the pooled snow depth standard deviation for each 1 m$^2$ cell in the forest (gray) and field (hashed).

## 4. **Challenges and Recommended Improvements to UAS Lidar Snow Depth Mapping**

Despite UAS-based lidar's increasing use in the natural sciences and capacity to make high-resolution snow maps, there are many operational and technical challenges that require consideration prior to successfully conducting UAS-based lidar surveys that produce research grade, high-resolution snow depth data. For the lidar surveys, the hardware and supporting software analysis tools can be expensive and require trained pilots and lidar data analysis specialists. In this section, we present some general considerations regarding validation of the lidar snow depth maps, selection and deployment of a lidar sensor on a UAV for snow depth mapping as well as specific insights that we experienced when using our system.

### 4.1 *In Situ* and UAS Sampling

This study's lidar snow depth performance metrics are comparable to those from the more extensive lidar surveys made Harder et al. (2020), In the field, our snow depth errors, 1 cm bias and 1.2 cm RMSD, were modestly better than those from their open sites snow depth 3 cm bias and RMSE values on the order of 10 cm. In the forest, our snow depth errors, 7 cm bias and 10 cm RMSD, were also modestly lower than those from their forest sites 9 to 13 cm bias and 15 cm RMSE. While it is difficult to make direct comparison across different study sites, snow conditions, and ground validation approaches, these early findings indicate that UAS lidar has the capability of mapping snow depths in open and forested regions and has improved performance as compared to previous SfM results particularly for vegetated surface. It is also noteworthy that this study's mapping was conducted using the Velodyne Puck series, a laser scanner adapted from the assisted and autonomous vehicle applications, rather than the specialized Riegl miniVUX-1UAV used by Harder et al. (2020) resulting in a complete mapping system that was approximately one-third the costs of their Riegl system.

While UAS-based lidar surveys can measure snow depth to within a centimeter at high spatial resolutions, validation of those observations is challenging. A time consuming collection of high accuracy GNSS survey points was required to co-locate magnaprobe and lidar observations. Surveying and marking sample locations prior to the winter season might reduce this effort. However, the use of sampling stakes risks modifying snow processes at the sample locations and potentially biasing the snowpack and incurring destructive sampling impacts if same location is being repeatedly visited over a season.

It is also challenging to make *in situ* snow depth measurements that provide centimeter accuracy. In this study, the magnaprobe *in situ* snow depth observations made in the forest were considerably higher than the lidar observations as compared to the open field where the magnaprobe and lidar measurements were within 1 cm. Previous studies also found that snow depth observations from ALS measurements are biased lower than those from snow-probe observations in the forest (Hopkinson et al., 2004, Currier et al., 2019; Harder et al., 2020). In past studies, the causes of these differences have been partially attributed to the snow probe's ability to penetrate the soil and vegetation, human observers tending to make snow depth measurements in locations with relatively high snow (Sturm and Holmgren, 2018) and the reduced accuracy of the GNSS in the forest. Our study suggests additional issues in forest sampling including enhanced ground surface variability in forested areas relative to adjacent field areas and reduced lidar returns in forested areas as compared to field areas combine with sampling issues to contribute to the higher uncertainty in the forest snow depths observed in our study.

In this study, the cold temperatures and snow-free conditions prior to the January 19[th] and 20[th] snowfall event resulted in deeper frozen soils (23.5 to 25.5 cm) in the field and shallower soil frost depth (5.5 to 8.5 cm) in the west forest, which would have limited the probe penetration into soils at both sites. However, the forest has a 1-4 cm thick organic leaf litter layer that may have been penetrated by the magnaprobe. The average Federal snow sampler tube depths (13.1 cm) were not as deep as the magna probe (15.2 cm) and thus more closely match the lidar snow depth (7.8 cm; see Figure 3), though a considerable low bias (~5.3 cm) similar to that found by Harder et al. (2020) persists in the lidar snow depth relative to the federal snow sampler snow depths. Additional factors such as downed logs, thick understory, and fine-scale topographic features (ie: small boulders and hummocky terrain) as well as reduced ground return density may contribute to the lidar snow depth errors in a forest, whereas these factors are absent in the field.

An improved understanding of forest canopies impacts on lidar returns is also warranted. Recent work has demonstrated that lidar pulses are "lost" at a much higher rate in forest canopies than open ground due to interception, absorption, and scattering through canopy transmission, with the loss ratio largely influenced by the range of the target from the sensor (Liu et al., 2020). The data that we presented in this paper were acquired using constant flight speed and at consistent altitude above target areas. Because of this, it is feasible that forest canopy conditions and variable understory vegetation density may have resulted in lost pulses and increased uncertainty in our data set. Indeed, we did observe lower return densities for both ground and all returns in forested areas in our data set (Figure 4).

One possible outcome of these lidar sampling issues in forests was a significant difference in snow depth confidence intervals between field and forest types and among slope groups. Confidence intervals were highest in conifer stands and on steep slopes and lowest in the field. This is likely partially the result of lower ground return density in forests due to the combined effects of lost pulses and canopy occlusion in forested areas. Additionally, this observation may be driven by increased variability in snow-off ground surface due to higher variability in the subnivean terrain in the forested areas of the study site (e.g., pockets of duff and woody debris). On cells where slopes exceed 20°, there is more variability in ground return elevations over shorter distances due to errors in horizontal directions and spreading of the laser spot (Deems et al., 2013), which would partially drive larger confidence intervals of ground surface elevation for pixels located in high relief areas. These relatively high slope areas were more common in forested areas of the study site, and the DTM uncertainty resulting when there are high slopes also carries through snow depth estimation. Snow depth was significantly different between field and forested areas, as well as between conifer and deciduous forest types, despite the relatively high uncertainty. This indicates the possible influence of tree canopies on snow accumulation due to enhanced snow interception in forests (see reviews in Clark et al., 2011), and particularly in conifer stands, but also could be the result of an under-sampled ground surface in forested areas relative to field areas. Despite challenges with sampling in the forest area, some

degree of coherence for snow depth in the forest is apparent. The forest interception effects may be captured on average through forest structure parameters such as canopy closure and leaf area index that have traditionally used in snow models with canopy-snow interactions (see reviews in Snow model inter-comparison project – SNOWMIP2 by Essery et al., 2009; Rutter et al., 2009). However, the finer scale heterogeneity may benefit from additional parameters such as the mean distance to canopy and total gap area (Moeser et al., 2016) or modifications that reflect variations in canopy structure (Mazzotti et al., 2019). Snow depth also was significantly different among the three slope groups, possibly due to wind-driven snow displacement and sloughing on slopes during accumulation.

4.2 Flight Planning

Because high lift UAVs capable of carrying a lidar sensor package have challenges that may differ from small consumer grade UAVs used mainly for optical photogrammetry surveying, a well-formulated flight plan that addresses weather conditions, logistics of flying at a proposed site, flight lines, UAS equipment, and personnel is clearly needed. Weather impacts operations. UAS surveys cannot be conducted when there is any type of precipitation or in dense fog/clouds because moisture can cause electronic components to malfunction and moisture build-up on the propellers can also adversely affect lift production. Depending on the UAS, wind speeds exceeding 7 to 10 m/s may make flights more difficult. This project's Eagle XF high lift capacity UAV cannot be flown comfortably in winds greater than 8 m/s. At the study site, wind speeds often exceeded this threshold in the days immediately following snowfall except early in the morning. High wind speeds can also significantly reduce battery life as well as impact the accuracy of sensor observations. Low air temperatures can cause batteries to rapidly discharge. For winter UAS surveys, all flight and operational batteries were kept warm in a building, vehicle, or insulated cooler prior to the UAS survey. This also applies to the computer used to upload flight lines and relay telemetry information. A MIL-STD-810 certified Panasonic Toughbook was used in this study to handle the anticipated cold temperatures. Additionally, cold temperatures can severely limit the dexterity of the person manipulating the flight controls.

These high lift UAVs also have the potential to cause significant damage to person and property. The selection of a survey site not only needs to meet the scientific objectives of the UAS lidar survey, but also must have the proper attributes for safe and legal UAV operation including permission to operate the UAV at the site. Visual line of sight (VLOS) of the UAS needs to be maintained throughout the flight. When it is difficult to maintain VLOS (e.g., flying over forested or mountainous sites), spotters can be used if there is constant two-way communication between the spotters and the person operating the flight controls. For this study, an on-site, walk up tower with a spotter was necessary while the UAS was flown over the forest.

The deployment of a UAS lidar requires additional flight patterns designed for boresighting to ensure that point clouds are aligned (Painter et al., 2016). Provided that GNSS data are accurate, the most common reason for misalignment of point clouds is boresight angle errors (Li et al., 2019). Boresighting is the process of calculating the differences between lidar sensor and IMU roll, pitch, and yaw angle measurements to correct those errors in point clouds. Due to battery flight time limitations, we were unable to complete the flight pattern that is commonly used for boresighting alignment. Because of this, we leveraged our first two antiparallel flight lines for boresighting calibration. Additional details on boresighting calibration, our technique due to the flight time limitations, and examples of roll and pitch alignment errors observed during this field campaign appear in the supplemental materials.

4.3 UAS Sampling Strategies

While lidar calibration and data post-processing requirements are quite similar for UAS and airborne surveys, the UAS lidar surveys presented in this study have key differences from previous ALS surveys. As noted above, UAS flight durations are

considerably shorter, resulting in limited spatial coverage as compared to previous ALS snow depth surveys. This study took approximately 2 hours to map a 0.1 km$^2$ area. An advantage of UAS over ALS surveys is that the average point cloud density is much higher and has fewer missing pixels in the forest. This study's sampling densities and the proportion of areas with no ground returns are quite different from previous airborne lidar snow depth studies. This study had ground returns of 90 and 364 points/m$^2$ in the forest and field, respectively, and had no ground returns in only 0.086% and 0.95% of the 1 m resolution field and forest cells, respectively. In contrast, ALS surveys typically report surface model densities between 8 to 16 points/m$^2$ (Broxton et al., 2015; 2019; Currier et al., 2019; Kirchner et al., 2014) and ground returns between 3 and 6 points/m$^2$ (Broxton et al., 2019; Kirchner et al., 2014). ALS derived snow depth maps have a much greater proportion of areas that are masked due to no ground returns, particularly under trees, with masking areas ranging from less to 10% to more than 23% (Harpold et al., 2014; Mazzotti et al., 2019). While gap filling is possible, interpolation using measured snow depth values to fill under tree can overestimate snow depth (Zheng et al., 2016). Based on our work comparing field and forest lidar collections from a UAS, we suggest testing alternative flight plans, including reduced flight speed over forest canopies to account for lost pulses and canopy returns to produce ground return density that is comparable to field ground return density and to further reduce the number of missing pixels in an acquisition area. It is worth noting that as the capabilities of UAVs, power supplies and lightweight sensors continue to advance at an accelerated rate, UAS platforms will shortly rival the spatial coverage attainable by manned aircraft while maintaining improved efficiency and cost effectiveness.

A well understood challenge exists when developing a spatial sampling strategy in which, for given resources, there is a trade-off between spatial extent and sampling density (Clark et al. 2011). Increasing flight altitude can expand the spatial extent of an aerial survey. However, flying at higher altitudes results in a decreased point density. In theory, a higher point density could be achieved by slower speeds and increased swath overlap. The targeted spatial extent of an aerial survey dictates whether a manned aircraft or a UAV should be used. If the targeted area has a limited domain then using a manned airborne platform is probably overkill and inefficient for many studies and the use of a UAV would be more cost effective. However, as the domain increases in size, additional batteries would be required, much of the battery power would be used to reach the outer limits of the domain, and the ability to maintain the required line of sight could be difficult. Thus, there are end-members for survey site or regions where it is self-evident as to whether a UAV or an airborne platform should be used, but that leaves considerable gray areas where an appropriate choice of UAV platform with a well-designed mission could stretch the domain. Future research and technological advances are needed to offer insights for snow science observation platforms and trade-offs.

## 5. Conclusions

This paper describes and demonstrates a UAS-based lidar survey for snow depth mapping using commercially available components. The snow depth map was assessed in a mixed deciduous and coniferous forest and open field with little relief over a thin snowpack. The UAS includes an Eagle XF UAV manufactured by UAV America, a small, lightweight VLP-16 lidar (Velodyne, Inc.), and an Applanix APX-15 UAV INS. The INS has a measurement rate of 200 Hz, allowing returns to be georeferenced without ground control points. Data, post-processed to a SBET) file, resulted in approximately 3 cm positional accuracy. Flights were conducted at an altitude of 81 m and flight speed of 7 m/s, with a total of 12 parallel flight lines with targeted overlap of 40 percent. Once the point clouds were classified as ground and non-ground points, the flights yielded an average of 90 and 364 ground points/m$^2$ in the forest canopy and field, respectively, with 6.7% of the forest and 0.03% of the field cells having less than 5 point/m$^2$.

The snow depth map, generated by subtracting snow-off from snow-on DTMs derived from the resultant point clouds, reveals a snowpack whose depth ranges from less than 2 cm to over 18 cm. For both snow depth and confidence intervals,

differences were found between vegetation cover types and slope, indicating complex snow-vegetation interaction that can be observed by UAS lidar return numbers. For 0.4 x 0.4 m cells, the *in situ* and lidar mean snow depths in the field were nearly identical with the MAD and RMSD values of approximately 1 cm. In the forest, the *in situ* mean snow depths from a magnaprobe were twice as large as the lidar snow depths with a correspondingly high RMSD. These forest differences have numerous possible explanations; 1) the snow probe's ability to penetrate the soil and vegetation resulting in random errors,

2) higher uncertainty in areas with canopy cover, variable ground surface, and high slope that occur more commonly in forested areas, 3) reduced total and ground return density in forests due to occlusion and lost pulses. Nevertheless, the results support previous findings indicating that there are limits to lidar snow depth validation at high horizontal and vertical spatial resolutions in some land covers and conditions. Mapped at 1 $m^2$ cells, a 0.5 to 1 cm snow depth confidence interval was achieved consistently in the field with confidence intervals increasing to median values of 4.0 cm in the deciduous forest and

5.9 cm in the coniferous forest. In the field, snow depth can be mapped at finer spatial resolutions with limited reduction in performance when reducing the cell size to 0.5 x 0.5 m and still achieving snow depth confidence intervals of less than 5 cm for a 0.2 x 0.2 m. Performance depends on both the point cloud density, which can be increased or decreased by changing the flight plan, and the grid cell variability that depends on site surface conditions.

**Acknowledgements**

This material is based upon work supported by the Broad Agency Announcement Program and the Cold Regions Research and Engineering Laboratory (ERDC-CRREL) under contract number W913E5-18-C-005. Any opinions, findings and conclusions or recommendations in this material are those of the author(s) and do not necessarily reflect the views of the Broad Agency Announcement Program and the Cold Regions Research and Engineering Laboratory.

The authors are grateful to Lee Friess for providing a technical review of the draft manuscript, Mahsa Moradi Khaneghahi for supporting manuscript preparation, and Ronny Schroeder contributing to the field data collection.

**Data Availability**

The UAS-based lidar point clouds and in-situ snow observations are available from the corresponding author upon reasonable request.

**Author Contributions**

JJ, AH, FS, and MP designed research and performed analysis. JJ, AH, FS, MP, EB, and EC conducted field work to obtain lidar and/or in-situ snow observations. AH, FS, CH, and EC produced figures. JJ wrote the initial draft. All authors contributed to manuscript review and editing.

**Competing Interests**

The authors declare that they have no conflict of interest.

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
