# Peer review of "Snow depth mapping with unpiloted aerial system lidar observations: A case study in Durham, New Hampshire, United States"

_The Cryosphere, 2020_

## Referee Comment (RC1) · Anonymous Referee #1 · 4 May 2020

Jacobs et al. present snow depth maps measured with a lidar onboard an unmanned aerial system (UAS). The snow depth are calculated as the difference between a snow-on and a snow-off DTM. They study a shallow snowpack with snow depth inferior to 20 cm in a flat open terrain and forested terrain. The lidar snow depth are compared to in situ magnaprobe measurements. They also provide some insights on what controls the lidar precision. The article is innovative as results are obtained with a new combination of sensors and platform which is lidar and UAS. This was, to my knowledge, only suggested by Vander Jagt et al. (2015) but not yet tested. Although this article focuses on shallow snowpack, it can be inferred that this method is promising for deeper snowpacks in open terrain. I see two points which should be addressed before I would recommend this article for publication.

1. The novelty of this work is not well highlighted. L 95, the authors state: "However, to date there are few previous studies that estimate snow depth using UAS-based lidar (Vander jagt et al., 2013(5!)).". In my understanding Vander Jagt et al. did not use UAS-based lidar and no other study ever did. The authors should verify the method in Vander Jagt et al. (2015) and cite the "few previous studies" that did similar work, if they exist. If this article is the first to present snow depth maps measured with UAS-based lidar, this should be clearly stated.

2. The main drawback which should be resolved is the way the "precision" and "accuracy" of the lidar snow depth maps are presented through the article. First, these two terms are not clearly defined. "Precision of the mean snow depth" is found first at L 232 and compared to "one-sided confidence interval". However, this last term is defined as equivalent to "the uncertainty of the lidar estimate of the snow depth" L181 in a confusing paragraph. Following this, it seems like we end up comparing "accuracy" and "precision" of the snow depth (L232) which I do not think was the initial goal. I rather understood that the authors intend to compare i) the accuracy calculated by comparing lidar and magnaprobe snow depth to ii) the lidar precision defined as the one-sided confidence interval. If I understood correctly, this need to be clearly stated, terms to be defined and consistently used. The definition of precision and accuracy proposed in Eberhard et al. (2020) found in Maune and Naygandhi (2018) might help. Related to this topic, the authors use within-cell standard deviation of the elevation twice: in equation (1) in what seems related to the accuracy of the lidar and L 262 to define "the within-cell variability". It seems like in the first case, the standard deviation results from error in the lidar while in the second case, the standard deviation results from the natural variability of the snow pack. As long as this is not clarified, it is hard to understand the point of the paragraph starting L260 in which the authors state that "In

addition to the lidar point cloud density, the ability to precisely capture the snow depth also depends on the within cell variability".

Minor comments are listed below. L21 : better repeat snow probe instead of "in situ"

L21 : "with" instead of "from" ?

L 34 : Make clear that the albedo is "higher" than the ground albedo not than the deeper snowpacks albedo.

L 55 : precise "point measurements"

L 55-57 : Could you clarify this sentence. Maybe split it in two. Plus, I do not understand the opposition you see between increasing spatial variability and small-scale feature. Finally, is it so sure that spatial variability "naturally increases with spatial scale"? Fig. 4. of Deems et al. (2006) seems to show that spatial variability stops increasing above a typical distance of the order of 10 m.

L 63: If you list the methods using difference of surface elevation, you may want to include spaceborne photogrammetry (e.g. Marti et al. 2016, McGrath et al. 2019, Shaw et al. 2019). Otherwise, if you prefer focusing on airborne method, you should remove references to terrestrial laser scanning.

L 76 : what is "micro scale" and "field scale" ?

L 96 : Vander Jagt 2015

L 135 : How do such angles occur since the channels are between -15/+15°? Is it because of the roll and pitch of the UAS?

L 151 : Please indicate what kind of "non-ground point" you observe in this area. Trees, artifacts.. ?

L 153 : Do you further use th and w notations ?

L 154 : "mean" without s?

L 159 :What do you mean with "Following processing"? The sentence is not clear.

L 181 : This paragraph is confusing. It seems that lines 181 and 187 are not consistent. Is the "uncertainty" from L181 the same as the one from L187? See main comment about precision and accuracy. L 181 : you state "uncertainty of the lidar estimate of the snow depth" is the "one-sided 95 % confidence interval" L 185 : you define a "pooled standard deviation" not used after. L 187 : you combine "snow depth uncertainty", "number of lidar return" and "pooled degrees of freedom" to calculate "the one-sided width of the 95 % confidence limits"

L 185 : Does this assume that the spatial variability within the cell is negligible? See main comment on precision, accuracy.

L 191 and following : Please make clear for what resolution these percentages hold.

L 198 : You state "0.95 %" of the forest cells are empty for the 1 m resolution grid. Does that correspond to the white areas in the western forest (Fig. 4) ? In case it is, this seems to be more than 1 % of the forested area. In case it is not, what are these white areas?

L 212 : In "(12.2 cm +-0.56 cm)", is 0.56 cm the standard deviation of the population of mean snow depth ? Or is it related to the standard deviation described in L 185 ?

L 215 : First time the word "tube" is used. Was it the "federal snow sampling tube" (L 172) ?.

L 232-233 : "precision" is not defined above. This sentence is thus hard to understand.

L. 260 : " In addition to the lidar point cloud density, the ability to precisely capture the snow depth also depends on the within cell variability. " Why? Is it a statement based on the way you calculate the lidar precision or an assumption which should be justified? See main comment on within-cell variability.

L. 260 : this is not mandatory but since you use standard deviation, did you check

whether the distribution is normal or not ?

L 319 "boresighting"

L 319. Could you explain what boresighting is ? Not sure The Cryosphere readers know what it is.

L. 368 : Could you provide details about the "simple penetration test" ? If this not it, do you think it would be possible to dig a snow pit at the location of the magnaprobe measurement to evaluate probe penetration ?

L. 389 : "moderately" please give values.

L 510. Missing a carriage return before "Starkloff"

Fig. 1: what's the reason for the buffer around the forest polygon, especially why is the forest peninsula out of both zones (east of the field, west of the western forest) ?

Fig. 2.a The number of returns per cell seems to follow a relationship of type $y=kx^2$ with k the average density of the point cloud and x the cell resolution. Could you comment on that? Did you expect that?

Fig. 2.b It is not so easy to distinguish the two distributions. Maybe remove the vertical lines of the bars ?

Fig. 5, what are the gray points/area on panel a. It seems absent in panel b.

Fig. 6.a. Isn't that surprising that the STD per cell is the same with snow on and off in the forest? Could you comment on that?

Fig 7. Label the panels a,b,c,d instead of A/B top/bottom. Zoom in the panel b. Keep a. as it is and add a square showing where b. is. It is really not clear what is shown in A,B. Are we in 2D view from top in A and from profile in B?

Reference:

Deems JS, Fassnacht SR and Elder KJ (2006) Fractal Distribution of Snow Depth from

Lidar Data. J. Hydrometeorol. 7(2), 285–297 (doi:10.1175/JHM487.1)

Eberhard LA, Sirguey P, Miller A, Marty M, Schindler K, Stoffel A, Bühler Y (2020) Inter-comparison of photogrammetric platforms for spatially continuous snow depth mapping Cryosphere Discussions (https://doi.org/10.5194/tc-2020-93)

Marti R, Gascoin S, Berthier E, De Pinel M, Houet T and Laffly D (2016) Mapping snow depth in open alpine terrain from stereo satellite imagery. Cryosphere 10(4), 1361–1380 (doi:10.5194/tc-10-1361-2016)

Maune DF (Ed.) and Naygandhi A (Ed.) Digital Elevation Model Technologies and Applications: The DEM Users Manual, 3rdEdition, 3 ed., 652 pp., 2018.

McGrath D, Webb R, Shean D, Bonnell R and Marshall HP (2019) Spatially Extensive Ground ‐ Penetrating Radar Snow Depth Observations During NASA ' s 2017 SnowEx Campaign : Comparison With In Situ, Airborne, and Satellite Observations. Water Resour. Res. 10 (doi:10.1029/2019WR024907)

Shaw TE, Gascoin S, Mendoza PA, Pellicciotti F and McPhee J Snow depth patterns in a high mountain Andean catchment from satellite optical tri- stereoscopic remote sensing. Water Resour. Res. di (doi:10.1029/2019WR024880)

Vander Jagt B, Lucieer A, Wallace L, TUrner D and Durand M (2015) Snow Depth Retrieval with UAS Using Photogrammetric Techniques. Geosciences, 264–285 (doi:10.3390/geosciences5030264)

---

## Referee Comment (RC2) · Anonymous Referee #2 · 5 May 2020

In this study, the investigators mounted a small airborne lidar on a drone and flew several test flights to map snow depths across a small flat farm in New Hampshire that contained fields and forest. They then chose one flight to examine in detail. Most of the paper is concerned with the accuracy of the resultant snow depth maps, with comparison of those derived depths against on-the-ground probing (n=130), and with an extensive analysis of accuracy vs. ground point spacing from the lidar.

My overall impression of the paper is that a single acquisition flight in a single landscape, with a quite limited ground collection campaign, is too thin a reed on which to

base a full journal publication. Such a limited comparison leaves open too many questions, like what the results would be if the ground was sloped, how the results would vary if the forest canopy was conifer vs. deciduous, what would happen if the snow had surface relief or other characteristics not tested in this work. In fact, the authors Figure 1 indicates a complex forest with openings and variable canopy density (a snow season air photo here would have been nice), but no attempt has been made to see if the results from one part of the forest look like those from another. No attempt was made to test how well the ground and air results match each other as a function of canopy and ground characteristics. Lastly, while the lidar and ground measurements matched beautifully in the open field, they showed a large discrepancy in the forest, which was then ascribed to over-probing through a duff layer. Perhaps that is the case, but this then ought to have been the focus of more analysis and scrutiny. The conclusion is certainly possible, but Figure 2b suggests there is also lidar sampling bias problem in the forests, and the core depths referred to in the text against which the depth probe depth was compared are never discussed, even to the extent of how many were made.

The other problem with the paper is that it is too equipment/system specific. Not everyone reading this paper will have the same drone, the same lidar etc., so what does the paper offer them? It is perhaps necessary to be equipment-specific in this type of paper to some extent, but to maximize its use to the wider community, the authors need to strive to separate what is inherent in the methodology used with the specific equipment test to what might be more universal. They try this in the discussion section with some lessons-learned statements, but these too general and read a bit like "be careful when you drive" rules. I am not sure what would be best in this regard, but some improvement is definitely needed.

Lastly, considerable space in the text is given to thin, shallow snow covers, and other lidar and airborne methods of mapping snow. While clearly when there is a fixed error in snow depth mapping (e.g., $\pm 3$ cm), it is a more serious problem in thin snow. Ultimately this is a methods paper, and nothing described in the accuracy and operation of the

lidar is limited or specific to thin snow.

I am going to recommend that this paper be returned for major revisions and specifically the inclusion of more extensive testing across a wider set of snow and terrain conditions. In revision, I would suggest that the focus of the paper be honed to be squarely focused on the methodology and not waste journal space on issues related to thin snow covers, for which no real new information was presented.

Recommendation: Return for major revisions and strengthen with more flights over a wider range of terrain and vegetation.

Detailed Comments

Abstract: First three sentences could be deleted.

Lines 1 to 98 could readily be deleted with no loss to the topic of the paper (thin snow discussion).

Figure 1: Nice graphic. . .very clear.

Line 84: Ground control points are mentioned, but I don't see any indication that they used control points for the SfM maps beyond the 200hz measurement rate, and I don't understand how that works.

Line 158: DTM not defined, which reflects a certain unevenness in the technical level of the paper. Who is this paper for? The new practitioner or the veteran GIS and UAV group? There are many acronyms in the paper all of which should when first presented be defined.

Line 166: Ground probe sampling method was a 5-sample cross pattern, with a GNSS GPS point in the center of the cross, but the authors wait until line 175 to tells us they averaged these 5 samples. What was the logic behind the sampling protocol and why only 5 points per 0.4 m sampling pixel, when the lidar was producing between 25 and 90? Surely more could have been measured? Also, later in the paper a core tube

(Federal s ampler?) is mentioned but no other details about it. About here in the paper it would also be good to mention the nature of the ground surface and depth of freeze, instead of later when trying to explain the discrepancy between the forest and field measurements errors.

Line 240-Figure 4: The maps look quite good, and the inclusion of the confidence map is to be commended. But several aspects shown on this figure go unremarked. Specifically, how was the location of the ground validation determined, and why so few ground data? It is unfortunate that for the field ground data, other data from the shallower area bracketing the road wasn't obtained so that a second thinner field comparison could be made. As for the confidence map, the very high confidence area in the center of western forest is at the nexus of all the flight lines. . ..is that why the confidence is high there? Conversely, comparing Fig. 1 to 4a and 4b, there are gaps and openings in the trees in both east and west forest where the confidence drops considerably, yet one might have expected these to function like the open filed. Why does it drop?

Figure 7: OK. . .but anyone new to airborne lidar will not understand it, and anyone already doing SfM or lidar will not need it. Think of who you are writing for.

Line 286 to 316: This is the first time that large vs. small UAVs are differentiated, though the weight of the lidar package would suggest a larger UAV was in use. But a quick scan of the web suggest that the drone used can handle about 14 kg. . .and recent some heavy lift drones are getting near 100 kg. Much of the discussion here seems like lessons learned that anyone trying to fly these larger drones probably already knows. It could be helpful, but they aren't detailed enough to really guide a newcomer to a successful mission. See the general point of trying to write a paper that is generic rather than specific. . ... which for rapidly changing tech can be challenging.

Lines 333 to 334: Heavy payload=short flight duration=small area mapped, hence better ground point density. While that makes sense, can't that be achieved by slower speed, closer passes etc.? And mapping extent, of course can be larger if more missions are used. So, I was puzzled what this paragraph was really trying to say.

---

## Author Response (AR1)

Response to Editor
Many thanks to the two reviewers for their detailed and constructive comments on the manuscript. Our revisions include major modifications to the introduction and discussion, new analyses to capture the impact of forest canopy on snow depth, and improvements to the clarity, importance, and target audience of the paper.

As outlined in more detail in our responses to the review comments, the most major revisions are new analyses to test how well the ground results match each other as a function of canopy and ground characteristics. The results reveal distinct snow conditions by vegetation cover (field, coniferous forest, and deciduous forest) as well as slope. We use methods from forest ecology to use our snow-off lidar survey to construct maps of vegetation cover type using a Canopy Height Model (CHM) to distinguish the upper level intact coniferous canopy from other forest cover. Ground and canopy height profiles derived from the lidar dataset are also used to explain differences in lidar derived observations and performance. The use of lidar returns to characterize the forest canopy along side estimates of snow depth is an important strategy for the snow community seeking insight to snow-vegetation interactions and is now highlighted in the revised manuscript.

We also note that Harder et al.'s (2020) UAV lidar manuscript was published on June 15[th] and found by the author team as we were finalizing our comments. The author team has included references to this Harder et al. (2020) in our revisions.

Our response to each comment is outlined below in **bold**. Revised text is in red with line numbers referenced to the manuscript without track changes. We hope these responses are clear, and we look forward to submitting the revised manuscript.

Harder, P., Pomeroy, J. W., and Helgason, W. D.: Improving sub-canopy snow depth mapping with unmanned aerial vehicles: lidar versus structure-from-motion techniques, The Cryosphere, 14, 1919-1935, 2020.

Anonymous Referee #1

*Thank you for the detailed comments and the opportunity to clarify that this article is the first to present snow depth maps measured with UAS-based lidar. We have provided detailed responses to the reviewer following each of the reviewer's comments.*

Jacobs et al. present snow depth maps measured with a lidar onboard an unmanned aerial system (UAS). The snow depth are calculated as the difference between a snow- on and a snow-off DTM. They study a shallow snowpack with snow depth inferior to 20 cm in a flat open terrain and forested terrain. The lidar snow depth are compared to in situ magnaprobe measurements. They also provide some insights on what controls the lidar precision. The article is innovative as results are obtained with a new combination of sensors and platform which is lidar and UAS. This was, to my knowledge, only suggested by Vander Jagt et al. (2015) but not yet tested. Although this article focuses on shallow

snowpack, it can be inferred that this method is promising for deeper snowpacks in open terrain. I see two points which should be addressed before I would recommend this article for publication.

1. The novelty of this work is not well highlighted. L 95, the authors state: "However, to date there are few previous studies that estimate snow depth using UAS-based lidar (Vander jagt et al., 2013(5!)).". In my understanding Vander Jagt et al. did not use UAS-based lidar and no other study ever did. The authors should verify the method in Vander Jagt et al. (2015) and cite the "few previous studies" that did similar work, if they exist. If this article is the first to present snow depth maps measured with UAS-based lidar, this should be clearly stated.

**A. We reviewed the earlier manuscript and concur that our manuscript is the first UAS-based lidar snow depth mapping manuscript when it was reviewed and during our revision. Shortly prior to resubmission, on June 15[th], Harder et al. 2020 was published. There is a notable difference in systems between our study and their study. They also used a considerably more expensive system (~$300K Canadian). We modified the abstract and the introduction to clarify.**

**Lines 17-19** This paper provides some of the earliest snow depth mapping results on the landscape scale that were measured using lidar on a UAV. The system, which uses modest cost, commercially available components, was assessed in a mixed deciduous and coniferous forest and open field for a thin snowpack (< 20 cm).

**Lines 99-103** However, to date there is only one other published study that estimated snow depth using UAS-based lidar (Harder et al., 2020). However, to date there are no published studies that estimate snow depth using UAS-based lidar. The purpose of this paper is to assess the ability of a UAS platform to provide snow depth using a modest cost UAS-based lidar. The pilot study described here serves as a proof-of-concept for providing a high vertical resolution snowpack dataset in open terrain and forests in the north-eastern United States.

2. The main drawback which should be resolved is the way the "precision" and "accuracy" of the lidar snow depth maps are presented through the article. First, these two terms are not clearly defined. "Precision of the mean snow depth" is found first at L 232 and compared to "one-sided confidence interval". However, this last term is defined as equivalent to "the uncertainty of the lidar estimate of the snow depth" L181 in a confusing paragraph. Following this, it seems like we end up comparing "accu- racy" and "precision" of the snow depth (L232) which I do not think was the initial goal. I rather understood that the authors intend to compare i) the accuracy calculated by comparing lidar and magnaprobe snow depth to ii) the lidar precision defined as the one-sided confidence interval. If I understood correctly, this need to be clearly stated, terms to be defined and consistently used. The definition of precision and accuracy proposed in Eberhard et al. (2020) found in Maune and Naygandhi (2018) might help. Related to this topic, the authors use within-cell standard deviation of the elevation twice: in equation (1) in what seems related to the accuracy of the lidar and L 262 to define "the within-cell variability". It seems like in the first case, the standard deviation results from error in the

lidar while in the second case, the standard deviation results from the natural variability of the snow pack. As long as this is not clarified, it is hard to understand the point of the paragraph starting L260 in which the authors state that "In addition to the lidar point cloud density, the ability to precisely capture the snow depth also depends on the within cell variability".

**A. Good point and this comment warranted considerable consideration and clarification for the reader. The reviewer's interpretation of our intent regarding accuracy is correct. However, our measure of variability is a combination of the instrumentation precision and the sample-to-sample variability within the grid cell (due to variations in surface elevation). Unlike Eberhard et al. (2020), the lidar returns in this study are only a sample of the entire surface. Thus, even if repeated lidar measurements agreed perfectly, there would still be variability within the pixel. We entirely rewrote section 2.6 Snow Depth Uncertainty Assessment. We have revised the definition of "accuracy" and provided a detailed context to the meaning of the confidence intervals of the lidar snow depth maps. Regarding the pooled standard deviation, this is a measure of the variability of the snow on and snow off lidar returns within a grid cell. This variability would depend on both lidar instrument and surface elevation variations. We also clarified the paragraph on L 260 to match the language (now lines 303 to 305). The text has been modified throughout to remove the term precision unless it specifically refers to a measure of the lidar instrumentation variability and to replace it with the "confidence interval".**

**Lines 219 to 237** The snow depth accuracy was assessed by comparing the lidar snow depth measurements to the magnaprobe measurements. Here, accuracy is the measure of the agreement of the lidar snow depth measurements relative to the in situ measurements (Eberhard et al., 2020; Maune and Nayegandhi, 2018). Error statistics were calculated and the results were summarized by forest and field locations. At each magnaprobe location, the average and standard deviation of the five magnaprobe samples were calculated. The average lidar snow depth was determined for a 0.4 x 0.4 m cell centered on the center magnaprobe location. The mean absolute difference (MAD) and root mean square difference (RMSD) were used to characterize the differences between the magnaprobe snow depths and the lidar snow depths.

The one-sided width of the 95% confidence limits for each cell's snow depth is a measure of the lidar snow depth variability. Confidence intervals are calculated using a cell's pooled standard deviation, the number of lidar returns, and the pooled degrees of freedom (Helsel and Hirsh, 1992) to calculate. A cell's snow depth pooled standard deviation $\sigma_d$ of the snow on and snow off elevations was calculated as

$$\sigma_d = \sqrt{\sigma_{on}^2 + \sigma_{off}^2}$$
(1)

where $\sigma_{on}$ and $\sigma_{off}$ are the standard deviation of the snow-on and snow-off lidar return elevations, respectively. This pooled standard deviation is a measure of the variability of the snow on and snow off lidar returns within a grid cell. This variability depends on the lidar instrument's relative accuracy (Maune and Nayegandhi, 2018), which includes

intra-swatch accuracy (i.e., precision or repeatability of measurements) and inter-swatch accuracy (i.e., differences in elevations between overlapping swaths), as well as surface elevation variations. The contribution from the individual sources of variability was not assessed.

**Lines 347 to 348** In addition to the lidar point cloud density, the ability to precisely capture the snow depth also depends on the ground surface variability within a cell variability as well as the lidar performance.

Minor comments are listed below. L21 : better repeat snow probe instead of "in situ" L21 : "with" instead of "from" ?
**A. Modified.**

L 34 : Make clear that the albedo is "higher" than the ground albedo not than the deeper snowpacks albedo.
**A. This line was removed when the introductory paragraph was modified significantly to reflect reviewer 2's comments about shallow snowpack and this reviewer's more general statement about the value of high-resolution snow depth measurements beyond shallow snowpacks.**

**Lines 30 to 48** Snowpacks are highly dynamic, accumulating and ablating throughout the winter with associated changes in snowpack density, grain size, and albedo (Adolph et al., 2017) as well as ice formation. Wind redistribution, sloughing of snow-off slopes, trapping of snow by vegetation, and forest canopy interception result in a range of spatial features at varying scales (Clark et al., 2011; Mott et al., 2011; Mott et al., 2018). Modest differences in snowpack depth can differentially impact many hydrologic, agricultural, and ecosystem processes. Differences in snowpack meltwaters can alter streamflow volumes (Gichamo and Tarboton, 2019), change the likelihood of spring floods (Tuttle et al., 2017) and intensify overland nutrient transport and soil erosion (Seyfried et al., 1990; Singh et al., 2009).

High-resolution snow depth measurements are also needed to discern processes that depend on the snow state. Insulation by seasonal snow in the Arctic and Antarctic slows sea ice growth (Sturm et al., 2002). High-resolution Arctic snow depths from ICE-Sat2 revealed seasonal snow on ice that would be missed when using coarser snow information (Kwok et al. 2020). Thin, ephemeral snowpacks have limited insulation and allow the underlying soils to freeze more readily in the winter (Groffman et al., 2001; Starkloff et al. 2017; Yi et al. 2019). Soil frost severity impacts soil respiration, carbon sequestration, nutrient retention, and microbial communities as well as a plant root health and tree growth (Aase and Siddoway, 1979; Isard and Schaetzel, 1998; Monson et al., 2006; Henry, 2008; Aanderud et al., 2013; Tucker et al., 2016; Sorensen et al., 2018; Reinmann and Templer, 2018). Detection and mapping of rapid thinning of snowpacks followed by frigid cold during "winter whiplash" events (Casson et al. 2019) is therefore important for understanding ecosystem impacts of soil freezing events, which are otherwise not well quantified (Kraatz et al. 2018; Prince et al. 2019). High vertical resolution snow mapping would greatly improve our understanding of these unique

habitats.

L 55 : precise "point measurements"
**A. Modified.**

L 55-57 : Could you clarify this sentence. Maybe split it in two. Plus, I do not understand the opposition you see between increasing spatial variability and small-scale feature. Finally, is it so sure that spatial variability "naturally increases with spatial scale"? Fig. 4. of Deems et al. (2006) seems to show that spatial variability stops increasing above a typical distance of the order of 10 m.
**A. Thank you for the Deems et al. (2006) reference, which points to a short-range fractal segment and a long-range with a break between 15 and 40 m. The referenced lines were split as recommended to make two separate points as follows:**

**Lines 54 to 57** Using traditional, precise point measurements with a limited sample size, the experimental design requires a balance between the sampling extent and sample spacing (Clark et al. 2011). However, the choice of sampling resolution may yield different measures of snow depth spatial variability when the snow exhibits multifractal behaviour (Deems et al. 2006).

L 63: If you list the methods using difference of surface elevation, you may want to include spaceborne photogrammetry (e.g. Marti et al. 2016, McGrath et al. 2019, Shaw et al. 2019). Otherwise, if you prefer focusing on airborne method, you should remove references to terrestrial laser scanning.
**A. The list of methods was modified to include spaceborne references provided by the reviewer.**

**Lines 61 to 66** Spaceborne photogrammetry (e.g. Marti et al. 2016, McGrath et al. 2019, Shaw et al. 2019), airborne laser scanning (ALS) (Deems et al., 2013; Harpold et al., 2014; Kirchner et al., 2014), terrestrial laser scanning (TLS) (Grünewald et al. 2010; Currier et al. 2019), and structure-from-motion photogrammetry (SfM) (Nolan et al., 2015; Bühler et al., 2016; Harder et al., 2016) have emerged as viable methods to map surface elevations with snow-off and snow-on conditions in order to differentially map snow depths.

L 76 : what is "micro scale" and "field scale" ?
**A. We clarified the scales and now Clark et al.'s definitions where they define point scales as less than 5 m and associated with topographic depressions and trapping or interception by individual vegetation features; hillslope or field scales are 1-100 m and associated with drifting and forest canopy interception and sublimations.**

**Lines 76 to 80** For some snowpack features, the typical vertical accuracies from these platforms, on the order of 10 cm (Kraus et al., 2011; Deems et al., 2013), as well as relatively low return density ($\sim$10 returns/m$^2$) (Cook et al., 2013) may not be adequate to

observe spatial variations at point scales (0 to 5 m) to hillslope and field scales (1-100 m) or to detect snow depth changes over short time scales.

L 96 : Vander Jagt 2015
**A. We removed the Vander Jagt reference in this statement.**

L 135 : How do such angles occur since the channels are between -15/+15°? Is it because of the roll and pitch of the UAS?
Because of degrading accuracy at distances >100 m with the VLP-16, returns acquired outside of +/- 30 degrees of nadir view angles were filtered to limit target distance and improve overall accuracy
**A. We clarified in the text that there are two different field of views on this sensor: 1) the vertical field of view (channels between -15/+15$^o$) and 2) the horizontal field of view (rotation angle of channel, 0-360$^o$). While returns from all vertical field of view channels were used, returns from wide angle views retrieved by each channel (outside of +/- 30$^o$ of nadir) were removed.**

**Lines 131 to 133** The VLP-16 is a 16-channel lidar with a 30-degree vertical field of view with rotating lasers that are spaced evenly between -15 to +15 degrees, with each channel rotating to provide a horizontal field of view of 360-degrees.

**Lines 144 to 146** Because of degrading accuracy at distances >100 m with the VLP-16, returns acquired outside of +/- 30 degrees of nadir view angles in the horizontal field of view were filtered to limit target distance and improve overall accuracy.

L 151 : Please indicate what kind of "non-ground point" you observe in this area. Trees, artifacts.. ?
**A. The progressive morphological filter only identifies ground returns– remaining returns are assumed to be primarily from vegetation (trees and understory shrubs). We clarified this discussion by rewording the first sentence and specified that remaining points are assumed to be from trees and vegetation with minimal artifacts.**

**Lines 159 to 162** The PMF operates iteratively on sets of two parameters, window size and elevation thresholds to erode and dilate point cloud data sets to estimate surface topography as an approach to filter out non-ground returns (i.e. trees, shrubs, and noise) from point cloud data sets (Zhang et al., 2003).

L 153 : Do you further use th and w notations ?
**A. No, these were removed.**

L 154 : "mean" without s?
**A. No. "means" was replaced with "approach"**

L 159 :What do you mean with "Following processing"? The sentence is not clear.
**A. The text was clarified.**

**Lines 167 to 169** Following ground classification for each tile, returns within the 15 m tile buffers were removed, and all resulting 100 m square ground classified tiles were merged. The resulting point clouds for each data set included both the classified ground returns and the non-ground returns.

L 181 : This paragraph is confusing. It seems that lines 181 and 187 are not consistent. Is the "uncertainty" from L181 the same as the one from L187? See main comment about precision and accuracy. L 181 : you state "uncertainty of the lidar estimate of the snow depth" is the "one-sided 95 % confidence interval" L 185 : you define a "pooled standard deviation" not used after. L 187 : you combine "snow depth uncertainty", "number of lidar return" and "pooled degrees of freedom" to calculate "the one-sided width of the 95 % confidence limits"
**A. This paragraph has been rewritten to address the confusion and word choice after a careful review of the Reviewer's comments and reading Maune and Nayegandhi (2018). The within cell variability is not negligible. The confidence interval reflects the within cell variability and, when combined with the lidar precision Please see the earlier comment for additional information. Please see the earlier comment for additional information.**

**Lines 226 to 237** The one-sided width of the 95% confidence limits for each cell's snow depth is a measure of the lidar snow depth variability. Confidence intervals are calculated using a cell's pooled standard deviation, the number of lidar returns, and the pooled degrees of freedom (Helsel and Hirsh, 1992) to calculate. A cell's snow depth pooled standard deviation $\sigma_d$ of the snow on and snow off elevations was calculated as

$$\sigma_d = \sqrt{\sigma_{on}^2 + \sigma_{off}^2}$$

(1)

where $\sigma_{on}$ and $\sigma_{off}$ are the standard deviation of the snow-on and snow-off lidar return elevations, respectively. This pooled standard deviation is a measure of the variability of the snow on and snow off lidar returns within a grid cell. This variability depends on the lidar instrument's relative accuracy (Maune and Nayegandhi, 2018), which includes intra-swatch accuracy (i.e., precision or repeatability of measurements) and inter-swath accuracy (i.e., differences in elevations between overlapping swaths), as well as surface elevation variations. The contribution from the individual sources of variability was not assessed.

L 185 : Does this assume that the spatial variability within the cell is negligible? See main comment on precision, accuracy.
**A. Please see the response to the previous comment.**

L 191 and following : Please make clear for what resolution these percentages hold.
**A. The resolution was clarified.**

**Lines 241 to 243** The snow-on and snow-off flights lidar ground returns yielded an average point cloud density of 90 and 364 points/m$^2$ in the forest and field, respectively, with 6.7% of the forest and 0.03% of the 1 m$^2$ field cells having less than 5 point/m$^2$

L 198 : You state "0.95 %" of the forest cells are empty for the 1 m resolution grid. Does that correspond to the white areas in the western forest (Fig. 4)? In case it is, this seems to be more than 1 % of the forested area. In case it is not, what are these white areas?
**A. Thank you for catching an issue with the eastern forest boundary. The white areas in the eastern forest are empty cells resulting for a river that runs along the forest. Infrared energy is absorbed by water; therefore, no lidar data were collected over the river. The forest boundaries in Fig 1 and 4 were updated to reflect the eastern forest boundary that was used in the analysis, which excludes the river.**

L 212 : In "(12.2 cm +-0.56 cm)", is 0.56 cm the standard deviation of the population of mean snow depth ? Or is it related to the standard deviation described in L 185?
**A. The 0.56 cm standard deviation is the standard deviation of the in situ Magnaprobe measurements in the field. The mean snow depth was calculated at each in situ sampling location. Then the average and standard deviation of the field locations (N = 11) was calculated. It is not related to the pooled standard deviation described on L 185. The pooled standard deviation described on L185 was used to calculate the 95% confidence intervals of the lidar derived snow depth.**

L 215 : First time the word "tube" is used. Was it the "federal snow sampling tube" (L 172) ?.
A. Yes, it is the federal snow sampling tube.

**Lines 263 to 265** The mean snow depth from the Federal snow tube was (12.9 cm ±0.71 cm) and (13.1 cm ±1.9 cm) in the field and forest, respectively. There is a notable low bias in the lidar forest snow depth relative to the magnaprobe and snow tube for west forest in particular with exception of one site.

L 232-233 : "precision" is not defined above. This sentence is thus hard to understand.
**A. See previous response and definition in section 2.5.**

**Lines 233 to 237** This variability depends on the lidar instrument's relative accuracy (Maune and Nayegandhi, 2018), which includes intra-swatch accuracy (i.e., precision or repeatability of measurements) and inter-swath accuracy (i.e., differences in elevations between overlapping swaths), as well as surface elevation variations. The contribution from the individual sources of variability was not assessed.

L. 260 : " In addition to the lidar point cloud density, the ability to precisely capture the snow depth also depends on the within cell variability. " Why? Is it a statement based on the way you calculate the lidar precision or an assumption which should be justified? See main comment on within-cell variability.
**A. This was clarified in the initial comment on the topic. We modified this sentence to clarify that there are two sources of variability in the cell. "**

**Line 347 to 348** In addition to the lidar point cloud density, the ability to precisely

capture the snow depth also depends on the ground surface variability within a cell as well as the lidar precision.

L. 260 : this is not mandatory but since you use standard deviation, did you check whether the distribution is normal or not ?
**A. We didn't check normality on a cell-by-cell basis, but did calculate the moments including skew values on a cell-by-cell basis at various scales. At the 10 and 20 cm cell size, there was not a notable skew. Larger cell sizes had increasingly negative skews with skew values typically less than -1.**

L 319 "boresighting"
L 319. Could you explain what boresighting is ? Not sure The Cryosphere readers know what it is.
**A. Considerable additional explanatory text and figures were added to the discussion on boresighting in order to provide a specific example to anyone who is new to airborne lidar. Our goal is to provide a specific example using a snow depth survey that will provide information beyond that available in a standard textbook discussion of boresighting. The new text and revised figure were moved to supplemental material. This location change was in response to Reviewer 2's comment about Figure 7: "OK...but anyone new to airborne lidar will not understand it, and anyone already doing SfM or lidar will not need it. Within the supplemental material we define boresighting.**
**Lines S8 – S9** Boresighting is the process of calculating the differences between the lidar sensor and IMU roll, pitch, and yaw angle measurements to correct those errors in point clouds.

L. 368 : Could you provide details about the "simple penetration test" ? If this not it, do you think it would be possible to dig a snow pit at the location of the magnaprobe measurement to evaluate probe penetration?
**A. Based on Reviewer 2's comments, this sentence was removed and replaced with additional details about the soil frost depth. In the future, it would be possible to dig a snow pit at the magnaprobe locations to determine. We did not do this during the experiment because we did not observe the bias until the lidar datasets had been post-processed.**

**Line 211 to 217** An independent study collected soil frost depth from three locations at the Thompson Farm Research Observatory using Gandahl-Cold Regions Research and Engineering Laboratory (CRREL) style frost tubes. The frost tubes have flexible, polyethylene inner tubing filled with methylene blue dye whose color change is easy to differentiate when extruded from ice (Gandahl 1957). A nylon string housed inside the polyethylene tubing affixes ice during periods of thaw. The outer tubing consists of PVC pipe installed between 0.4 to 0.5 m below soil surface (Ricard et al., 1976; Sharratt and McCool, 2005). Prior to the January 19[th] and 20[th], 2019 snowfall event, soil frost was 23.5 to 25.5 cm in the field and 5.5 to 8.5 cm in the west forest.

L. 389 : "moderately" please give values.

**A. Values were added to the text.**

**Line 508 to 510** Mapped at 1 m$^2$ cells, a 0.5 to 1 cm snow depth confidence interval was achieved consistently in the field with confidence intervals increasing to within 4 cm in the forest and heavily vegetated areas.

L 510. Missing a carriage return before "Starkloff"
**A. Carriage return was added.**

Fig. 1: what's the reason for the buffer around the forest polygon, especially why is the forest peninsula out of both zones (east of the field, west of the western forest) ?
**A. Thank you for the keen eye. The buffer around the forest polygon was removed and the peninsula is now included in the eastern forest. All plots and figures were updated to reflect any changes to the field/forest boundaries.**

Fig. 2.a The number of returns per cell seems to follow a relationship of type y=kx$^2$ with k the average density of the point cloud and x the cell resolution. Could you comment on that? Did you expect that?
**A. Yes, this nonlinear relationship could be expected because the counts are based on area of the DTM (length squared) rather than the resolution (length). For example, if a 1 m x 1 m areas (1m$^2$) have 100 returns, then a 2 m x 2 m areas (4m$^2$) should have 400 returns.**

Fig. 2.b It is not so easy to distinguish the two distributions. Maybe remove the vertical lines of the bars?
**A. The hatched fill pattern has been removed. Also, the line weight of the field distribution has been increased to more easily distinguish between the two distributions.**

[Figure]

**Figure 2**. (a) Average lidar point cloud density of the ground returns with versus cell size by land cover, and snow-on and snow-off state (top). (b) Probability density function for the lidar ground returns point cloud density for 1 m$^2$ cell for the forest (gray) and the field (hashed) (bottom).

Fig. 5, what are the gray points/area on panel a. It seems absent in panel b.
**A. The points showed the individual outliers of the distributions. They have now been removed from figure 5a.**

[Figure]

**Figure 5.** One sided confidence intervals of the mean snow depth values in the field and forest at Thompson Farm, Durham, NH on January 23, 2019 from the individual cells for 1 m² cells by land cover and point cloud density (top) and for grid resolutions ranging 0.1 to 5 m (bottom). Boxplots show the lower quartile, median, upper quartile, and whiskers.

Fig. 6.a. Isn't that surprising that the STD per cell is the same with snow on and off in the forest? Could you comment on that?
**A. Yes, this is somewhat surprising and we had not seen this effect noted in previous studies. A comment was added to offer an explanation for the difference.**

**Line 350 to 353** Snow cover reduces the within cell variability in field by about 1 cm, but has a limited effect in the forest. It is possible that the modest snowpack was able to flatten the higher grass in the field, while the forest's vegetation and terrain features that dominate the within cell variability are only minimally compacted by the snow.

Fig 7. Label the panels a,b,c,d instead of A/B top/bottom. Zoom in the panel b. Keep a. as it is and add a square showing where b. is. It is really not clear what is shown in A,B. Are we in 2D view from top in A and from profile in B?
**A. Figure was heavily modified, with many clarifications included in the figure caption and the text. All boresighting figures and text are now in supplemental materials part 3.**

[Figure]

**Figure S3.** Uncalibrated boresight angles between the INS and lidar sensor can result in poorly aligned point clouds (a1 and b1). Arrows in (a) and (b) show approximate flight direction during data acquisition. The lidar returns within the box marked in red in (a) are shown in (a1) and (a2) at an oblique view angle. Figure (a1) shows how boresight errors of roll angles present, while (a2) shows proper boresight alignment for roll. Roll alignment errors present well in anti-parallel flight lines (flight lines flown parallel to each other but in the opposite direction), flown over **flat** terrain. Figure (b) shows the approximate location of returns used for pitch boresight alignment error demonstration (b1) and its correction (b2). Pitch misalignment presents well in anti-parallel flight lines in areas with terrain relief while viewing across the flight track, as opposed to along the flight track as with roll alignment. For (b, a1, a2, b1, and b2), only ground returns are shown for each flight line, while in (a), all returns are shown.

**The following references were added to the manuscript based on Reviewer 1's input:**

Deems JS, Fassnacht SR and Elder KJ (2006) Fractal Distribution of Snow Depth from Lidar Data. J. Hydrometeorol. 7(2), 285–297 (doi:10.1175/JHM487.1)

Eberhard LA, Sirguey P, Miller A, Marty M, Schindler K, Stoffel A, Bühler Y (2020) Inter- comparison of photogrammetric platforms for spatially continuous snow depth mapping Cryosphere Discussions (https://doi.org/10.5194/tc-2020-93)

Marti R, Gascoin S, Berthier E, De Pinel M, Houet T and Laffly D (2016) Mapping snow depth in open alpine terrain from stereo satellite imagery. Cryosphere 10(4), 1361–1380 (doi:10.5194/tc-10-1361-2016)

Maune DF (Ed.) and Naygandhi A (Ed.) Digital Elevation Model Technologies and Applications: The DEM Users Manual, 3rdEdition, 3 ed., 652 pp., 2018.

McGrath D, Webb R, Shean D, Bonnell R and Marshall HP (2019) Spatially Extensive Ground Penetrating Radar Snow Depth Observations During NASA's 2017 SnowEx Campaign: Comparison With In Situ, Airborne, and Satellite Observations. Water Resour. Res. 10 (doi:10.1029/2019WR024907)

Shaw TE, Gascoin S, Mendoza PA, Pellicciotti F and McPhee J Snow depth patterns in a high mountain Andean catchment from satellite optical tri- stereoscopic remote sensing. Water Resour. Res. di (doi:10.1029/2019WR024880)

Vander Jagt B, Lucieer A, Wallace L, TUrner D and Durand M (2015) Snow Depth Retrieval with UAS Using Photogrammetric Techniques. Geosciences, 264–285 (doi:10.3390/geosciences5030264)

**Interactive comment on The Cryosphere Discuss., https://doi.org/10.5194/tc-2020-37, 2020.**

Anonymous Referee #2

*Thank you for the detailed comments and the opportunity refine the original submission and to consider variations across land-use and terrain. We have provided detailed responses to the reviewer following each of the reviewer's comments.*

In this study, the investigators mounted a small airborne lidar on a drone and flew several test flights to map snow depths across a small flat farm in New Hampshire that contained fields and forest. They then chose one flight to examine in detail. Most of the paper is concerned with the accuracy of the resultant snow depth maps, with comparison of those derived depths against on-the-ground probing (n=130), and with an extensive analysis of accuracy vs. ground point spacing from the lidar.

My overall impression of the paper is that a single acquisition flight in a single landscape, with a quite limited ground collection campaign, is too thin a reed on which to base a full journal publication. Such a limited comparison leaves open too many questions, like what the results would be if the ground was sloped, how the results would vary if the forest canopy was conifer vs. deciduous, what would happen if the snow had surface relief or other characteristics not tested in this work. In fact, the authors Figure 1 indicates a complex forest with openings and variable canopy density (a snow season air photo here would have been nice), but no attempt has been made to see if the results from one part of the forest look like those from another. No attempt was made to test how well the ground and air results match each other as a function of canopy and ground characteristics. Lastly, while the lidar and ground measurements matched beautifully in the open field, they showed a large discrepancy in the forest, which was then ascribed to over-probing through a duff layer. Perhaps that is the case, but this then ought to have been the focus of more analysis and scrutiny. The conclusion is certainly possible, but Figure 2b suggests there is also lidar sampling bias problem in the forests, and the core depths referred to in the text against which the depth probe depth was compared are never discussed, even to the extent of how many were made.

**A. The reviewer makes a number of reasonable points regarding the long-term value of limited flights over limited landscapes. We entirely agree that this submission leaves open questions, particularly given the strong contrast in performance between the field and the forest. Based on the reviewer's comments, we reconsidered this paper's contribution in light of the early structure from motion (SfM) papers that used a UAS platform to characterize snow depth. A summary of those studies in light of the reviewer's comments appears in Table R1 (below). These recent papers share many commonalities with the current study in that they seek to understand how a recent technological development might contribute to improved understanding of the snow depth. The table shows how the literature and experiments evolved over time. These papers also demonstrate that the experimental design and results from this study equals or exceeds that of these early SfM studies that also sought to demonstrate the value of a new combination of sensors and platform.**

| Yr Flown | Location | Area | # flights | Site (# and Description) | Validation | Error | Method Detail | Study |
|---|---|---|---|---|---|---|---|---|
| 2013 | Tasmania, Australia | 0.0069 km$^2$ | 1 | 1. strong gradient in elevation, thick vegetation and various soil/rock | Survey pole 37 measured, N= 20 due to vegetation, survey at snow surface, then ground surface | 0.10 m (acc) RMSE = 9.6 cm | Yes & Workflow | (Vander Jagt et al. 2015) |
| 2014 | Lombardy region, northern Italy | 0.3 km$^2$ | 1 | 1. sparse grass coverage and rocks, with no tree, firn, or glacier ice. | 12 probe measurements horiz. accuracy 2–3 cm. | Bias 0.073 m and aRMSE = 0.14 m | Yes | (De Michele et al. 2016) |
| 2015 | Rosthern, Saskatchewan, Canada

 Canadian Rocky Mountains | 0.65km$^2$

 0.32 km$^2$ | 22, 18 | 1. Canadian prairie; tall stubble (35 cm) and short stubble (15 cm) Sparsely vegetated 2. Rocky Mountain alpine ridgetop grasses, shrubs and coniferous trees in gullies | Ruler with 17 snow stakes - horiz.accuracy ±2.5 cm. 34 points

 Alpine: 3 to 19 pts per flight. 5 SD measurements in a 0.4 m × 0.4 m square at that point | 8.8 cm for a short stubble, 13.7 cm for a tall stubble 8.5 cm alpine mean SD must be > 30 cm | Yes | (Harder et al. 2016) |
| 2015 | Davos, Switzerland | 0.057 – 0.091 km$^2$
 0.29 km$^2$ | 3/1 | 1. Tschuggen: flat alpine meadows and hilly alpine terrain 2. Brämabühl: an exposed location meadow and bushes | 60, 95, 95 and 110 (5 pts per site) 5 SD measurements in a 1 m × 1 m square - center pt horiz. accuracy < 10 cm | Overall RMSE = 0.25 m bias = 0.2 m Short grass RMSE 0.07 m bias 0.05 m Bushes/high grass RMSE 0.30 m bias 0.29 m alpine RMSE 0.15 m bias 0.11 m | Yes | (Bühler et al. 2016) |
| 2016 | Piedmont region, Italy | 0.0067 km$^2$ | 1 | 1. sparse rocks and grass, with no trees | 135 pts and TLS UAS, a multi station survey, and manual probing | RMSE = 0.31 m overall RMSE = 0.17 m areas of likely water accumulation removed | limited | (Avanzi et al. 2017) |
| 2016 | Canada | 0.02 km$^2$ | 13/16 | 1 and 2. G Gatineau: N. Shrubs up to 1m. S. Shrubs and sm. forested area southwest corner S. 3 to 5. Acadia A. grass (< 5 cm) and stumps (< 20 cm). B stumps (< 20 cm) and brush and shrubs (< 1 m). C 1 – 5 m balsam fir | Transects of ~ 50 m in length; 12 48″ × 2″ × 1″ wooden stakes; no horiz. accuracy | 2 to 11 cm RMSD for SD change | Yes | (Fernandes et al. 2018) |
| 2015 | Alps West | 0.12 km$^2$ | 12 | 1. alpine meadow | 149 M... | 0.25 m | Yes | (Adams et al. |

Table R1. Review of early structure from motion papers

In brief, Table R1 indicates that the studies that used SfM to map SD were first published in 2015 and 2016. In those early studies, the number of flights was extremely limited, the surveyed area was typical quite small, there was often only a single site, and the cover conditions were typically relatively short grass, stubble, with limited shrubs and no or limited trees. Studies that estimate SfM SDs in sites having significant tree canopies were published approximately three years after the initial studies. These SfM papers are an example where the early papers use targeted, focused studies to provide the broader community with an approach that is now embraced, and which has been subsequently refined and used to explore a range of landscapes, terrain, and forest canopy.

The submitted manuscript, as noted by Reviewer 1, "is the first to present snow depth maps measured with UAS-based lidar" and the novel contribution is its results that were obtained with a new combination of sensors and platform. Our manuscript also sets the stage for further research by including results that demonstrate a sharp contrast between the field and the forest findings as well as considerable variability of metrics within in the forests. We expect the broader community will contribute the additional studies that the reviewer desires, with more extensive campaigns over a wide range of landscapes, following a similar trajectory of UAV-based SfM in the embracement of new technology. Note that Harder et al.'s (2020) UAV lidar manuscript was published on June 15[th]. The author team has included references to this study.

We revised the manuscript to be clearer about the contribution including in the abstract, the last paragraph of the introduction, and the conclusion. Specifically, we have added context of our work in the Discussion to emphasize the refinement of methodology and new questions that emerged from our work. Our work highlights, unknown at the time of study implementation, sampling and collection finding that are useful for planning for future snow depth studies.

Regarding the lidar sampling bias problem in the forests, the reviewer makes a number of reasonable points including that ascribing the errors to over-probing is likely a gross simplification of the complexity of measuring forest SD. Based on this comment, the discussion section that discusses these issues has been revised, the snow core observation information has been expanded in Section 2.4 (renamed "*In Situ* Observations") and in Section 4 (renamed "Challenges and Recommended Improvements to UAS Lidar Snow Depth Mapping", last paragraph), and a preliminary assessment of variations in forest canopy has been added. Even with high ground return lidar that is collected with a UAS, forest canopies still generate collection issues that complicate interpretation and characterization of snow. When collecting data over a region, forest type and canopy characteristics and their impacts on a lidar snow depth survey may not known in advance. We have added a section in the Discussion that describes issues found with forests in our study, including reduced total and ground return density in forests compared to open fields, and we make suggestions on how data collection strategies might be modified for forested areas. Additionally, we suggest that further studies may be warranted to understand how forest vegetation (e.g. canopy species, understory vegetation density, and duff layer quality) contributes to snow depth measurement bias, while pointing to recent

**evidence in the literature of challenges inherent to sampling mixed land-use landscapes with airborne lidar sensors.**

**Lines 378 to 425** 4.1 *In Situ* and UAS Sampling
While UAS-based lidar surveys can measure snow depth to within a centimeter at high spatial resolutions, validation of those observations is challenging. A time consuming collection of high accuracy GNSS survey points was required to co-locate magnaprobe and lidar observations. Surveying in sample locations prior to the winter season might reduce this effort. It is also challenging to make *in situ* snow depth measurements that provide centimeter accuracy. In this study, the magnaprobe *in situ* snow depth observations made in the forest were considerably higher than the lidar observations as compared to the open field where the magnaprobe and lidar measurements were within 1 cm. Previous studies also found that snow depth observations from ALS measurements are biased lower than those from snow-probe observations in the forest (Hopkinson et al., 2004, Currier et al., 2019; Harder et al., 2020). In past studies, the causes of these differences have been partially attributed to the snow probe's ability to penetrate the soil and vegetation, human observers tending to make snow depth measurements in locations with relatively high snow (Sturm and Holmgren, 2018) and the reduced accuracy of the GNSS. Our study suggests additional issues in forest sampling including enhanced terrain variability in forested areas relative to adjacent field areas and reduced lidar returns in forested areas as compared to field areas combine with sampling issues to contribute to the higher uncertainty in the forest snow depths observed in our study.

In this study, the cold temperatures and snow-free conditions prior to the January 19[th] and 20[th] snowfall event resulted in deeper frozen soils (23.5 to 25.5 cm) in the field and shallower soil frost depth (5.5 to 8.5 cm) in the west forest, which would have limited the probe penetration into soils at both sites. However, the forest has a 1-4 cm thick organic leaf litter layer that may have been penetrated by the magnaprobe. The average Federal snow sampler tube depths (13.1 cm) were not as deep as the magna probe (15.2 cm) and thus more closely match the lidar snow depth (7.8 cm; see Figure 3), though a considerable low bias (~5.3 cm) similar to that found by Harder et al. (2020) persists in the lidar snow depth relative to the federal snow sampler snow depths. Additional factors such as downed logs, thick understory, and fine-scale topographic features (ie: small boulders and hummocky terrain) as well as reduced ground return density may contribute to the lidar snow depth errors in a forest, whereas these factors are absent in the field.

An improved understanding of forest canopies impacts on lidar returns is also warranted. Recent work has demonstrated that lidar pulses are "lost" at a much higher rate in forest canopies than open ground terrain due to interception, absorption, and scattering through canopy transmission, with the loss ratio largely influenced by the range of the target from the sensor (Liu et al., 2020). The data that we presented in this paper were acquired using constant flight speed and at consistent altitude above target areas. Because of this, it is feasible that forest canopy conditions and variable understory vegetation density may have resulted in lost pulses and increased uncertainty in our data set. Indeed, we did observe lower return densities for both ground and all returns in forested areas in our data set (Figure 4).

One possible outcome of these lidar sampling issues in forests was a significant difference in snow depth confidence intervals between field and forest types and among slope groups.

Confidence intervals were highest in conifer stands and on steep slopes and lowest in the field. While this result is not entirely surprising, it is likely partially the result of lower ground return density in forests due to the combined effects of lost pulses and canopy occlusion in forested areas. Additionally, this observation may be driven by increased variability in snow depth due to pockets of duff and woody debris, and due to higher variability in subnivean terrain in the forested areas of the study site. Areas of high terrain relief are expected to have more variability in ground return elevations over shorter distances, which would partially drive higher confidence intervals of ground surface elevation for pixels located in high relief areas. High relief areas of the study site were more common in forested areas of the study site, and the uncertainty resulting around high slopes also carries through snow depth estimation. Snow depth was significantly different between field and forested areas, as well as between conifer and deciduous forest types, despite the relatively high uncertainty. This indicates the possible influence of tree canopies on snow accumulation due to enhanced snow interception in forests, and particularly in conifer stands, but also could be the result of an under-sampled ground surface in forested areas relative to field areas. Snow depth also was significantly different among the three slope groups, possibly due to wind-driven snow displacement and sloughing on slopes during accumulation.

**A. The core depth procedures originally described briefly in Section 2.4 were expanded. The core accuracy values appeared in section 3.2.**

**Lines 206 to 209** Along the same forest and field transects, a federal snow sampler was used to collect a single sample of snow depth and snow water equivalent at each magnaprobe sample location for a total of 12 field samples and 16 forest samples. Snow depth was measured by inserting the aluminium tube vertically into the snowpack and a core was extracted and weighed using a spring scale.

The other problem with the paper is that it is too equipment/system specific. Not everyone reading this paper will have the same drone, the same lidar etc., so what does the paper offer them? It is perhaps necessary to be equipment-specific in this type of paper to some extent, but to maximize its use to the wider community, the authors need to strive to separate what is inherent in the methodology used with the specific equipment test to what might be more universal. They try this in the discussion section with some lessons-learned statements, but these too general and read a bit like "be careful when you drive" rules. I am not sure what would be best in this regard, but some improvement is definitely needed.

**A. We have embraced the reviewer's comment "the authors need to strive to separate what is inherent in the methodology used with the specific equipment test to what might be more universal." and have rewritten the discussion section to more keenly focus on what we believe are the most useful lessons learned, broken them into more manageable units and clearly indicated what are generalizable lessons versus those that are instrument specific. The first paragraph in the discussion and sections 4.2 and 4.3 respond to the reviewer's comments.**

[revised manuscript text omitted]

**If the comment that the paper is "too equipment/system specific" is intended to mean that we should reduce the description of the equipment, we would push back because the authors strongly believe that the audience who is interested in replicating the experiment should be provided with adequate details to be able to do so. Authors who are interested in conducting similar studies with different instrumentation should be able to understand difference due to instrumentation versus those due to snow differences. Similarly, every experiment is equipment specific and most experiments across research groups do not use identical equipment. This author team has found papers very informative when methods and equipment are described in detail and not just overall results. When new methods and equipment are deployed in studies, the ability to recreate a study or examine the methods is important. This knowledge allows for repeatability, criticism of the experiment, and also can save a research team many hours when learning a new method or developing an experimental plan with technological equipment. Early SfM, airborne lidar, and UAS optical work included specific equipment details and methodologies.**

**We have slightly reduced our equipment description in the body of the text and reference supplemental material with a new table of technical specifications. We hope that this will balance out the reviewer's concern.**

Table S1. Technical specifications of the project UAS

| **UAS** | |
| --- | --- |
| UAS type | quadcopter |
| Manufacturer/Model | UAV-America / Eagle X8 |
| Diameter | 130 cm |
| Height | 70 cm |
| Number of rotors | 4 |
| Rotor diameter | 27.5 in (~70cm) |
| Motor Manufacturer/Model | KDE Direct / 7208 |
| RPM/Volt (KV rating) | 110 KV |
| Aircraft empty weight | 8 kg |
| Aircraft weight at take-off (with payload) | 16 kg |
| Flight time at take-off weight | ~7 minutes |
| Tolerable wind speed (with payload) | 5 m/s |
| Flight controller | Pixhawk PX4 |
| Flight Batteries | 22,000 mAh 6 Cell Lipo (2X) |
| | |
| **Sensor Payload** | |
| Gimble | Gremsy H7 |

| IMU/GPS | Applanix APX-15 |
| Lidar | Velodyne VLP-16 |
| Payload weight | 3 kg |

Lastly, considerable space in the text is given to thin, shallow snow covers, and other lidar and airborne methods of mapping snow. While clearly when there is a fixed error in snow depth mapping (e.g., ±3 cm), it is a more serious problem in thin snow. Ultimately this is a methods paper, and nothing described in the accuracy and operation of the lidar is limited or specific to thin snow.

**A. The reviewer makes a reasonable point that this work is more about pushing the envelope by reducing SD errors as opposed to thin snow per se and is relevant to any research that needs snow depth with a high vertical resolution. Based on the reviewer's comment, we have broadened the motivation to include a range of scenarios where an improved vertical resolution of SD beyond the existing 10+ cm resolution would be welcome. We have also discussed where the lidar observations are likely specific to thin snow.**

**Lines 30 to 57** Snowpacks are highly dynamic, accumulating and ablating throughout the winter with associated changes in snowpack density, grain size, and albedo (Adolph et al., 2017) as well as ice formation. Wind redistribution, sloughing of snow-off slopes, trapping of snow by vegetation, and forest canopy interception result in a range of spatial features at varying scales (Clark et al., 2011; Mott et al., 2011; Mott et al., 2018). Modest differences in snowpack depth can differentially impact many hydrologic, agricultural, and ecosystem processes.  Differences in snowpack meltwaters can alter streamflow volumes (Gichamo and Tarboton, 2019), change the likelihood of spring floods (Tuttle et al., 2017) and intensify overland nutrient transport and soil erosion (Seyfried et al., 1990; Singh et al., 2009).

High-resolution snow depth measurements are also needed to discern processes that depend on the snow state. Insulation by seasonal snow in the Arctic and Antarctic slows sea ice growth (Sturm et al., 2002). High-resolution Arctic snow depths from ICE-Sat2 revealed seasonal snow on ice that would be missed when using coarser snow information (Kwok et al. 2020).  Thin, ephemeral snowpacks have limited insulation and allow the underlying soils to freeze more readily in the winter (Groffman et al., 2001; Starkloff et al. 2017; Yi et al. 2019). Soil frost severity impacts soil respiration, carbon sequestration, nutrient retention, and microbial communities as well as a plant root health and tree growth (Aase and Siddoway, 1979; Isard and Schaetzel, 1998; Monson et al., 2006; Henry, 2008; Aanderud et al., 2013; Tucker et al., 2016; Sorensen et al., 2018; Reinmann and Templer, 2018). Detection and mapping of rapid thinning of snowpacks followed by frigid cold during "winter whiplash" events (Casson et al. 2019) is therefore important for understanding ecosystem impacts of soil freezing events, which are otherwise not well quantified (Kraatz et al. 2018; Prince et al. 2019). High vertical resolution snow mapping would greatly improve our understanding of these unique habitats.

Distributed modeling and mapping of snowpacks can increasingly provide output at fine spatiotemporal scales but snow state change validation typically relies on in situ observations (Gichamo and Tarboton 2019; Starkloff et al., 2017). Despite importance, few spatially continuous high-resolution snowpacks datasets are available to support modelling, and mapping

efforts.  Because snowpacks have considerable spatiotemporal variability, a large number of snow depth measurements are often needed to characterize the snowpack (Dickinson and Whiteley, 1972). Using traditional, precise point measurements with a limited sample size, the experimental design requires a balance between the sampling extent and sample spacing (Clark et al. 2011).  However, the choice of sampling resolution may yield different measures of snow depth spatial variability when the snow exhibits multifractal behaviour (Deems et al. 2006).

I am going to recommend that this paper be returned for major revisions and specifically the inclusion of more extensive testing across a wider set of snow and terrain conditions. In revision, I would suggest that the focus of the paper be honed to be squarely focused on the methodology and not waste journal space on issues related to thin snow covers, for which no real new information was presented.
Recommendation: Return for major revisions and strengthen with more flights over a wider range of terrain and vegetation.

**Thank you for the recommendations here and in the following sections.  We have refined the focus and the thin snow covers discussion is now only one aspect of the broader motivation for a new combination of sensors and platform to provide higher vertical resolution SD measurements.  Please see the previous comment and response.**

**The reviewer requested consideration of canopy and terrain variations. At this site, there are notable variations in slope as well as forest type. We conducted a new analysis to better quantify the canopy variations and to determine if the mean snow depth and the confidence intervals differ by slope or land-use. We found statistically significant differences for all combinations. Land-use differences include a new delineation of the forest by coniferous or deciduous trees. A new methods section 2.4 Slope and Vegetation Cover Classification and Analysis was added. The findings are reported in results section 3.3 Snow Depth Maps from UAS Lidar with an additional figure showing boxplots.**

**Lines 172 to 194 2.4 Slope and Vegetation Cover Classification and Analysis**
The snow-off DTM was used to develop a 1 m resolution map of slope (Horn, 1981). Vegetation cover type (field/forest) was determined from the known boundaries of field and forest.  The forested area was further classified as coniferous or deciduous for the study region using the following methodology (Figure 1). Within the forested area (Figure 1), a Canopy Height Model (CHM) was used to distinguish the intact upper canopy from other forest cover using our snow-off survey, collected with leaf off in the spring (Sullivan et al., 2017). The CHM was generated by subtracting the DTM produced using ground-classified points from the DSM produced using all lidar points. This results in a digital model consisting solely of canopy heights with no terrain or topography.  The CHM generation used raster images with a 1 m resolution.  A 3 by 3 maximum convolve filter was used to enhance the edges of canopy crowns and expand smaller regions that might have just one pixel of an intact canopy or a whole in a larger canopy (Palace et al., 2008).  A 15 m threshold was used to differentiate between the upper level intact coniferous canopy. CHM pixels that were below this threshold were deemed deciduous canopies (see supporting information for intermediate figures). The 5.6 ha forested area has a forest type that is 65% deciduous and 35% coniferous.

Once the vegetation forest type was classified, the raster binary image was vectorized. Within the forest and field regions of our study, a subsample was created from the entire image of 5000 random points in the field and 5000 random points each of the eastern and western forested areas (Palace et al., 2017). At each of these random points, slope, vegetation type (field, deciduous, coniferous), and snow depth and snow depth confidence interval values were extracted. Because of missing values in the raster images, not all random points extracted values and resulted in different numbers of samples points for the forest and forest types. Slope was assigned to one of three categories: 0-10 degrees, 10-20 degrees, and greater than 20 degrees. Because the extracted datasets (i.e., snow depth, confidence interval, and slope) were not normally distributed, the non-parametric Steel-Dwass Method test was used to test for differences. This non-parametric method is useful when sample numbers are large and groups are small, because it allows type I errors to be controlled (Dolgun and Demirhan, 2017).

**Lines 290 to 317 3.3 Snow Depth Maps from UAS Lidar**
The UAS-mapped snow depth, mapped by subtracting snow-off DTMs from snow-on DTMs, reveals a shallow snowpack whose depth ranges from less than 2 cm to over 18 cm (Figure 5). The mean lidar snow depth was 10.3 cm in the field and 6.0 cm in the forest. Despite the shallow conditions, spatially coherent patterns are readily discernible. The field snowpack depth has higher spatial variability than the west forest snowpack and more spatial organization. In the field, the deepest snow is in the low-lying northeast areas that are sheltered from westerly winds. A relatively moderate and consistent snowpack occurs in southern part of the east field and west of the small pond. The shallowest snowpack is found in the center portion of the field, which is slightly elevated and, unlike most of the field, was not mowed. Lower snow depth at the forest edge distinguishes the field to forest transition. A non-parametric Steel-Dwass test found significant variation for the mean snow depth among the two forest types and field ($p < 0.0001$) (Figure 6a). A pairwise Steel-Dwass test showed that snow depths were significantly different between the three pairs of field and forest types ($p < 0.0001$). When comparing just field and forest as categories, the test also found significant differences for snow depth ($p < 0.0001$). Snow depth was also determined to be significantly different among the three slope group categories using the Steel-Dwass test where regions with a limited slope (Group 1) had more decidedly different snow than steeper regions ($p < 0.0001$) (Figure 6b).

The one-sided confidence interval values of the mean snow depth estimate are remarkably consistent in the field and typically are between 0.5 to 1 cm regardless of snow depth (Figure 5b). Modestly larger confidence intervals occur adjacent to the north-south road where the fields were not mowed prior to winter as well as the northern and southern extents of the flight lines likely due to the reduced sampling density. The forest had an average one-sided confidence interval of 3.5 cm, which is considerably higher than the field. Where the forest is predominantly comprised of deciduous trees, the typical one-sided confidence intervals of the mean snow depth were as low as 1 to 2 cm. The largest one-sided confidence interval values occur in the middle of the field where there is dense shrubbery, at the edge of the fields, and in clusters within the forest where the forest sections are dominated by coniferous trees. The nexus of flight lines in the take-off and landing area resulted in a local area with very high confidence. A non-parametric Steel-Dwass test found significant variation for confidence intervals of the mean snow depth among the two forest types and field ($p < 0.0001$) (Figure 6c). A pairwise Steel-Dwass test showed that

confidence intervals were significantly different between the three pairs of field and forest types and (p < 0.0001). Confidence intervals were also significantly different among the three slope categories as determined using a Steel-Dwass test (p < 0.0001) (Figure 6d).

[Figure]

**Figure 6.** Snow depths (a,b) and their one sided confidence intervals (c,d) from the random sample points of the field and forest at Thompson Farm, Durham, NH on January 23, 2019 from the individual cells for 1 m$^2$ cells by vegetation cover (a,c) and slope group (b,d). Boxplots show the lower quartile, median, upper quartile, and whiskers with the median value noted. Because of missing values in the raster images, not all random points extracted values and resulted in different numbers of samples points for vegetation cover classes.

**While more extensive testing across a wider set of snow and terrain conditions would**

certainly be welcome, the previous literature with SfM SD shows that there is a place in the literature for limited, targeted, early studies and that these papers provide tremendous value as evidenced by their heavy citation rate and how they have informed subsequent research. Also, most of the early SfM SD papers were published in The Cryosphere.  The additional analysis on snow depth variations by land cover and slope add novel results for this region. There are very few snow depth studies in the northeast region that attempt to quantify the contribution of land cover and slope in the thin and ephemeral snowpacks that are increasingly characteristic of this region.

Our manuscript closely follows the model used by the early SfM studies and provides early guidance on methods for surveying and ground-based sampling as well as early results that provide insights to potential outcomes, performance and challenges. The requested additional datasets would very much change this submission and, as the request would require an additional winter field season, delay the communication of these early findings by over a year.

We hope our responses and explanations on why this paper is novel and a contribution to the field of shallow snowpack estimation using remotely sensed data warrants consideration of publication. We believe that our work presented in this manuscript is valuable for the community of researcher who are increasingly likely to consider including lidar UAS systems in experiments, with timely information to support decisions regarding whether to proceed with UAS lidar observations, to inform equipment purchases, and to plan field campaigns.

Detailed Comments
Abstract: First three sentences could be deleted.
Lines 1 to 98 could readily be deleted with no loss to the topic of the paper (thin snow discussion).

A. Based on the reviewer's comment, we have revised the motivation to include a range of scenarios where an improved vertical resolution of SD beyond the 10+ cm resolution would be welcome. Beyond shallow snowpacks, lines 54 forward provide a review of the methods used to measure SD and their limitations. A review of this literature is important to put this current new technology and methods in context. Based on the reviewer's comments the introduction section was entirely rewritten.

**Lines 30 to 66** Snowpacks are highly dynamic, accumulating and ablating throughout the winter with associated changes in snowpack density, grain size, and albedo (Adolph et al., 2017) as well as ice formation. Wind redistribution, sloughing of snow-off slopes, trapping of snow by vegetation, and forest canopy interception result in a range of spatial features at varying scales (Clark et al., 2011; Mott et al., 2011; Mott et al., 2018). Modest differences in snowpack depth can differentially impact many hydrologic, agricultural, and ecosystem processes.  Differences in snowpack meltwaters can alter streamflow volumes (Gichamo and Tarboton, 2019), change the likelihood of spring floods (Tuttle et al., 2017) and intensify overland nutrient transport and soil erosion (Seyfried et al., 1990; Singh et al., 2009).

High-resolution snow depth measurements are also needed to discern processes that depend on the snow state. Insulation by seasonal snow in the Arctic and Antarctic slows sea ice growth (Sturm et al., 2002). High-resolution Arctic snow depths from ICE-Sat2 revealed seasonal snow on ice that would be missed when using coarser snow information (Kwok et al. 2020). Thin, ephemeral snowpacks have limited insulation and allow the underlying soils to freeze more readily in the winter (Groffman et al., 2001; Starkloff et al. 2017; Yi et al. 2019). Soil frost severity impacts soil respiration, carbon sequestration, nutrient retention, and microbial communities as well as a plant root health and tree growth (Aase and Siddoway, 1979; Isard and Schaetzel, 1998; Monson et al., 2006; Henry, 2008; Aanderud et al., 2013; Tucker et al., 2016; Sorensen et al., 2018; Reinmann and Templer, 2018). Detection and mapping of rapid thinning of snowpacks followed by frigid cold during "winter whiplash" events (Casson et al. 2019) is therefore important for understanding ecosystem impacts of soil freezing events, which are otherwise not well quantified (Kraatz et al. 2018; Prince et al. 2019). High vertical resolution snow mapping would greatly improve our understanding of these unique habitats.

Distributed modeling and mapping of snowpacks can increasingly provide output at fine spatiotemporal scales but snow state change validation typically relies on in situ observations (Gichamo and Tarboton 2019; Starkloff et al., 2017). Despite importance, few spatially continuous high-resolution snowpacks datasets are available to support modelling, and mapping efforts. Because snowpacks have considerable spatiotemporal variability, a large number of snow depth measurements are often needed to characterize the snowpack (Dickinson and Whiteley, 1972). Using traditional, precise point measurements with a limited sample size, the experimental design requires a balance between the sampling extent and sample spacing (Clark et al. 2011). However, the choice of sampling resolution may yield different measures of snow depth spatial variability when the snow exhibits multifractal behaviour (Deems et al. 2006).

Over the past two decades, remote sensing methods, providing spatially continuous, high-resolution snow depth maps at local and regional scales, have greatly advanced the ability to characterize the spatiotemporal variability of snow depth over earlier work using snow probes (see reviews in Deems et al., 2013; López-Moreno et al., 2017). Spaceborne photogrammetry (e.g. Marti et al. 2016, McGrath et al. 2019, Shaw et al. 2020), airborne laser scanning (ALS) (Deems et al., 2013; Harpold et al., 2014; Kirchner et al., 2014), terrestrial laser scanning (TLS) (Grünewald et al. 2010; Currier et al. 2019), and structure-from-motion photogrammetry (SfM) (Nolan et al., 2015; Bühler et al., 2016; Harder et al., 2016) have emerged as viable methods to map surface elevations with snow-off and snow-on conditions in order to differentially map snow depths.

Figure 1: Nice graphic. . .very clear.
**A. Thank you.**

Line 84: Ground control points are mentioned, but I don't see any indication that they used control points for the SfM maps beyond the 200hz measurement rate, and I don't understand how that works.
**A. We are not sure what the reviewer means. We did not create any SfM maps for this paper. Line 84 is the literature review not methods. The 200hz referred to in the methods**

and conclusion is the sampling rate of the inertial navigation system (INS), which measures the position of the UAS during acquisition flights. Those data are then used to calculate the location of lidar returns. We do use GCPs in the same sentence as 200hz once, and it is to point out that one of the benefits of our lidar payload over SfM approaches is that a payload that relies on an INS does not require GCPs, while SfM does.

Line 158: DTM not defined, which reflects a certain unevenness in the technical level of the paper. Who is this paper for? The new practitioner or the veteran GIS and UAV group? There are many acronyms in the paper all of which should when first presented be defined.
**A. The acronyms were reviewed and defined. We apologize for the original omission of the definition of digital terrain model (DTM) and now include it.**

**Lines 164 to 167** PMF was parameterized using a set of window sizes of 1, 3, 5, and 9 m, and elevation thresholds of 0.2, 1.5, 3, and 7 m, which were determined by varying value sets and assessing digital terrain models (DTMs) to determine the parameter sets that produced a visually smooth surface over a dense grid (*sensu* Muir et al., 2017).

Line 166: Ground probe sampling method was a 5-sample cross pattern, with a GNSS GPS point in the center of the cross, but the authors wait until line 175 to tells us they averaged these 5 samples. What was the logic behind the sampling protocol and why only 5 points per 0.4 m sampling pixel, when the lidar was producing between 25 and 90? Surely more could have been measured? Also, later in the paper a core tube (Federal s ampler?) is mentioned but no other details about it. About here in the paper it would also be good to mention the nature of the ground surface and depth of freeze, instead of later when trying to explain the discrepancy between the forest and field measurements errors.
**A. Because the lidar observations were anticipated to give very high-resolution observations, we used an approach that would provide very high spatial precision for the in situ observation coordinates. The ground sampling protocol was informed by the methods used to validate SfM SDs. Harder (2016), Bühler et al. (2016), and Adams et al. (2018) used the same 5-sample cross pattern with a GNSS GPS point in the center of the cross. Our in situ SD observations were measured using the magna probe and then the center point was surveyed to a horizontal uncertainty of 2.51cm and 4.17cm for the field and forest, respectively, that meets or exceeds previous studies. The downside is that this procedure limits the number of in situ validation points.**

**The federal snow sampling tube was originally described on lines 172 and 173 (2.4 Snow Depth Ground Truth) and the later reference to the "tube" has been clarified. The section 2.4 Snow Depth Ground Truth section has been modified to 2.4 *In Situ* Observations. This section now includes requested a discussion of the ground surface and depth of freeze as well as additional details on the sampling methods.**

**Lines 195 to 217 2.5 *In Situ* Observations**
A 1.2-m Global Positioning System (GPS)-equipped magnaprobe (Sturm and Holmgren, 2018) was used to compare to the unmanned aerial system (UAV) lidar surveys (hereafter noted as ALS measurements) over two transects. The first transect consisted of 12 sample locations in the field and 5 locations in the eastern forest of our study site. The second transect consisted of 11

sample locations in the western forest. Sample locations were separated by approximately 10 m. The field transect follows the prevailing westerly wind direction with its west side at the foot of a modest depression (approximately 2 m below the land further to the west) and the east side transitioning into a wooded area. Following (Harder et al. 2016) and (Bühler et al. 2016), each sample location includes 5 samples in a cross pattern with the four ordinal directions sampled approximately 20 cm from the center sampling location in the cross. The five samples are used to provide a measure of SD central tendency and variation over a 0.4 x 0.4 m pixel. Because the magnaprobe GPS has an absolute accuracy of 8 m, a Trimble[©] Geo7X GNSS Positioning Unit with Zephr[™] antenna was used to collect each sampling location's center point with an estimated horizontal uncertainty of 2.51cm (standard deviation $\sigma$ 0.95 cm) and 4.17cm ($\sigma$ 4.60 cm) for the field and forest, respectively after differential correction. Along the same forest and field transects, a federal snow tube sampler was used to collect a single sample of snow depth and snow water equivalent at each magnaprobe sample location for a total of 12 field samples and 16 forest samples. Snow depth was measured by inserting the aluminium tube vertically into the snowpack and a core was extracted and weighed using a spring scale.

An independent study collected soil frost depth from three locations at the Thompson Farm Research Observatory using Gandahl-Cold Regions Research and Engineering Laboratory (CRREL) style frost tubes. The frost tubes have flexible, polyethylene inner tubing filled with methylene blue dye whose color change is easy to differentiate when extruded from ice (Gandahl 1957). A nylon string housed inside the polyethylene tubing affixes ice during periods of thaw. The outer tubing consists of PVC pipe installed between 0.4 to 0.5 m below soil surface (Ricard et al., 1976; Sharratt and McCool, 2005). Prior to the January 19[th] and 20[th], 2019 snowfall event, soil frost was 23.5 to 25.5 cm in the field and 5.5 to 8.5 cm in the west forest.

Line 240-Figure 4: The maps look quite good, and the inclusion of the confidence map is to be commended. But several aspects shown on this figure go unremarked. Specifically, how was the location of the ground validation determined, and why so few ground data? It is unfortunate that for the field ground data, other data from the shallower area bracketing the road wasn't obtained so that a second thinner field comparison could be made. As for the confidence map, the very high confidence area in the center of western forest is at the nexus of all the flight lines. . ..is that why the confidence is high there? Conversely, comparing Fig. 1 to 4a and 4b, there are gaps and openings in the trees in both east and west forest where the confidence drops considerably, yet one might have expected these to function like the open filed. Why does it drop?

**A. Thank you. Additional remarks about this figure were added based on the reviewer's comments including the point about the nexus of flight lines resulting in high confidence. The forest locations having a marked decreased confidence are locations where there is a dense canopy and limited lidar penetration combined with increased pulse loss. The higher variability in confidence in the forest is likely due to the heterogeneity of the forest structure, not canopy gaps as this is a continuous forest canopy. Instead, what the reviewer perceives to be gaps are more likely areas with more deciduous trees and variable terrain. A new analysis was conducted and added to the paper to examine the variability within the forest. The areas with marked decreased confidence are locations where there is a dense canopy and limited lidar penetration.**

**We were intrigued by the reviewer's comments about the confidence in the forests and revisited the forest locations. A new analysis of the forest canopy profiles and the ground versus nonground returns in the forest and field for both snow on and snow-off conditions was added.**

**Lines 312 to 314** The nexus of flight lines in the take-off and landing area resulted in a local area with very high confidence.

**Lines 266 to 278** To provide insight to differences between the forest and field observations, mean height profiles were calculated for a 25 m$^2$ square region centered on forest and field study plots from lidar data (Figure 4). To do this, all lidar returns were extracted from the bounding box of each plot, then the mean elevation of ground returns was calculated within each plot. Return heights for each plot were determined by subtracting the mean ground elevation of the plot, then the normalized return elevations were binned in 0.1 m height increments. Within forests, an average of 2142 and 2889 returns were classified as ground and non-ground in snow-free conditions per 25 m$^2$ plot, respectively with 2218 ground returns and 1721 non-ground returns in snow-on conditions. In field plots, an average of 5666 ground returns and 154 non-ground returns in snow-free conditions were obtained per 25 m$^2$ plot, with 7567 ground returns and 25 non-ground returns in snow-on conditions. Figure 4 also shows that there is a greater range of ground return elevations in the forest as compared to the field. In forest plots, ground return elevations had an average standard deviation of 0.157 m and 0.154 m in snow-free and snow-on conditions, respectively, while in field plots, ground return elevations had standard deviations of 0.058 m and 0.050 m in snow-free and snow-on conditions, respectively.

**The limited number of ground sampling points is discussed in the response to the previous section. We agree it is unfortunate that our field data didn't capture more of the variability. Unfortunately, because lidar post-processing takes some time, it is not possible to develop a sampling plan based on the lidar observations because the field data needs to be collected at nearly the same time as the lidar data. Similarly, field data collection occurs after the lidar acquisition because snow sampling and movement of people across the landscape alters the snow field.**

**Regarding how was the location of the ground validation determined: Our working hypothesis that informed the ground sampling design was that there would be limited local variations in precipitation in the field and that wind redistribution would drive variations in snow depth across the field. The field transect was set up along the prevailing wind direction with the west side at the foot of a modest depression (approximately 3-4 m below the land further to the west) and the east side transitioning into a wooded area in an effort to capture wind driven variations. The results instead showed limited SD variations along the transect as compared to notable SD variations and patterns that were readily evident from the lidar SD maps. This suggests opportunities for further research and will inform future in situ sampling strategies. We updated the methods to describe how the field transect was located.**

**Lines 199 to 201** The field transect follows the prevailing westerly wind direction with its west side at the foot of a modest depression (approximately 3-4 m below the land further to the west) and the east side transitioning into a wooded area.

Figure 7: OK...but anyone new to airborne lidar will not understand it, and anyone already doing SfM or lidar will not need it. Think of who you are writing for.
**A. This is a reasonable point and was also noted by Reviewer #1. We moved this to supplemental materials and modified the text. Because our target audience will likely include readers who are new to airborne lidar, this figure has been revised and the supporting text have been rewritten to make this important information for accessible to that audience. Additional explanatory text and figures were added to the discussion on boresighting in order to provide a specific example to anyone who is new to airborne lidar. Our goal is to provide a specific example using a snow depth survey that will provide information beyond that available in a standard textbook discussion of boresighting. We hope that the placement in supplementary material will allow readers who are new to lidar to have a specific example that is linked to this analysis, but will remove the material from the main body of the paper for those who do not need it.**

**S2 Boresight Calibration**

The deployment of a lidar system mounted on a UAV platform for snow depth monitoring requires flight patterns designed for calculating boresight alignment and post-processing to ensure that point clouds are properly aligned (Painter et al., 2016). Provided that GNSS data are accurate, the most common reason for misalignment of point clouds is boresight angle errors (Li et al., 2019). Boresighting is the process of calculating the differences between lidar sensor and IMU roll, pitch, and yaw angle measurements to correct those errors in point clouds. Traditionally, boresighting calibration is performed using antiparallel flight lines in addition to a perpendicular flight line (Keyetieu and Seube, 2019). Due to battery flight time limitations, it was not possible to complete the flight pattern that is commonly used for boresighting alignment. Because of this, the first two antiparallel flight lines were leveraged for boresighting calibration. Offsets between sensor and IMU are calculated by observing misalignments between lidar data collected from different flight lines, and iteratively adjusting roll, pitch, and yaw angles of the IMU data to produce sub-datasets into the same planes. To determine roll offset, broad (10 m) along-path cross-sections over flat terrain were assessed, and to determine pitch offset narrow (1 m) across-path cross-sections in sloped terrain where the point clouds overlapped were used (Figure S3). Though not shown here, unique features were leveraged within the data acquisition region, including barn roofs and deciduous tree branches, to assess the resulting boresight angles (Kumari et al., 2011; Li et al., 2005). For this particular study, boresight calibration was performed manually and iteratively. Methods often require extensive user input (Li et al., 2005), however boresight calibration is an increasingly automated process with wide variation in algorithms and approaches (e.g. Maas, 2000; Kumari et al., 2011; Zhang et al., 2019). In future work, automated boresight calibration methods to improve the accuracy of point cloud data sets will be explored.

Figure S2 shows two examples of ground return point clouds before and after calibration in this study's field region. Uncalibrated boresight angles between the INS and lidar sensor can result in

poorly aligned point clouds (i and iii). Red and blue arrows in (A) and (B) show approximate flight direction during data acquisition superimposed on the LAS point cloud. Roll alignment errors present well in anti-parallel flight lines (flight lines flown parallel to each other but in the opposite direction) over flat terrain. The top panel in Figure S3 addresses roll misalignment with (a) showing the LAS point cloud and the two flight lines flown in opposite directions. The lidar returns within the box marked in red in (a) are shown in (a1) and (a2) at an oblique view angle. Figure (a1) shows how boresight errors of roll angles present, while (a2) shows proper boresight alignment for roll. Figure (b) shows the approximate location of returns and flight lines used for pitch boresight alignment error demonstration (b1) and its correction (b2). Pitch misalignment presents well in anti-parallel flight lines in areas with terrain relief while viewing across the flight track, as opposed to along the flight track as with roll alignment.

[Figure]

**Figure S2.** Boresight examples that show how uncalibrated boresight angles between the INS and lidar sensor can result in poorly aligned point clouds (a1 and b1). Arrows in (a) and (b) show approximate flight direction during data acquisition. The lidar returns within the box marked in red in (a) are shown in (a1) and (a2) at an oblique view angle. Figure (a1) shows how boresight errors of roll angles present, while (a2) shows proper boresight alignment for roll. Figure (b) shows the approximate location of returns used for pitch boresight alignment error demonstration (b1) and its correction (b2). Pitch misalignment presents well in anti-parallel flight lines in areas with terrain relief while viewing across the flight track, as opposed to along the flight track as with roll alignment. For (b, a1, a2, b1, and b2), only ground returns are shown for each flight line, while in (a), all returns are shown.

Line 286 to 316: This is the first time that large vs. small UAVs are differentiated, though the weight of the lidar package would suggest a larger UAV was in use. But a quick scan of the web suggest that the drone used can handle about 14 kg. . .and recent some heavy lift drones are getting near 100 kg. Much of the discussion here seems like lessons learned that anyone trying to fly these larger drones probably already knows. It could be helpful, but they aren't detailed

enough to really guide a newcomer to a successful mission. See the general point of trying to write a paper that is generic rather than specific. . .. which for rapidly changing tech can be challenging.

**A. Agreed that additional details are needed to support the target audience. We envision an important audience of this research to be researchers who have used off the shelf systems such as the DJI Phantom IV and are considering instrumentation that would increase the UAV payload beyond that carrying light weight sensors such as optical sensors. We added a table of specifications to the supplemental materials and clearly differentiated this UAV from those used previously in SfM SD studies.**

Table S1. Technical specifications of the project UAS

| UAS | |
|---|---|
| UAS type | quadcopter |
| Manufacturer/Model | UAV-America / Eagle X8 |
| Diameter | 130 cm |
| Height | 70 cm |
| Number of rotors | 4 |
| Rotor diameter | 27.5 in (~70cm) |
| Motor Manufacturer/Model | KDE Direct / 7208 |
| RPM/Volt (KV rating) | 110 KV |
| Aircraft empty weight | 8 kg |
| Aircraft weight at take-off (with payload) | 16 kg |
| Flight time at take-off weight | ~7 minutes |
| Tolerable wind speed (with payload) | 5 m/s |
| Flight controller | Pixhawk PX4 |
| Flight Batteries | 22,000 mAh 6 Cell Lipo (2X) |
| | |
| **Sensor Payload** | |
| Gimble | Gremsy H7 |
| IMU/GPS | Applanix APX-15 |
| Lidar | Velodyne VLP-16 |
| Payload weight | 3 kg |

**There is a total 55lb (~25 kg) limit on UAVs with our specific license. Heavier than that requires additional licensing. Our effort is to provide information on UAVs that can carry a lidar, GPS, and IMU appropriate for shallow snow depth retrieval. Because our work is intended to be helpful to new researchers and even seasoned UAV groups, we have tended on the side of presenting additional equipment attributes and settings.**

**We entirely rewrote the discussion section and separated it into three distinct sections (4.1 *In Situ* and UAS Sampling, 4.2 Flight Planning, and 4.3 UAS Sampling Strategies. Regarding the material on flight planning, this section is now much tighter.**

**Lines 425 to 455** 4.2 Flight Planning
Because larger UAVs that can carry heavier payloads have challenges that may differ from small UAVs, a well-formulated flight plan that addresses weather conditions, logistics of flying at proposed site, flight lines, UAS equipment, and personnel is clearly needed. Weather impacts

operations. UAS surveys cannot be conducted when there is any type of precipitation or in dense fog/clouds because moisture can cause electronic components to malfunction and moisture build-up on the propellers can also adversely affect lift production. Depending on the UAV, wind speeds exceeding 7 to 10 m/s may make flights more difficult. This project's Eagle XF high lift capacity UAS cannot be flown comfortably in winds greater than 8 m/s. At the study site, wind speeds often exceeded this threshold in the days immediately following snowfall except early in the morning. High wind speeds can also significantly reduce battery life as well as impact the accuracy of sensor observations. Low air temperatures can cause batteries to rapidly discharge. For winter UAS surveys, all flight and operational batteries were kept warm in a building, vehicle, or insulated cooler prior to the UAS survey. This also applies to the computer used to upload flight lines and relay telemetry information. A MIL-STD-810 certified Panasonic Toughbook was used in this study to handle the anticipated cold temperatures. Additionally, cold temperatures can severely limit the dexterity of the person manipulating the flight controls.

High lift UAVs capable of carrying a lidar sensor package have the potential to cause significant damage to person and property. The selection of a survey site not only needs to meet the scientific objectives of the UAV survey, but also must have the proper attributes for safe and legal UAV operation including permission to operate the UAV at the site. Visual line of sight (VLOS) of the UAV needs to be maintained throughout the flight. When it is difficult to maintain VLOS (e.g., flying over forested or mountainous sites), spotters can be used if there is constant two-way communication between the spotters and the person operating the flight controls. For this study, an on-site, walk up tower with a spotter was necessary while the UAV was flown over the forest.

The deployment of a UAV lidar system requires additional flight patterns designed for boresighting to ensure that point clouds are aligned (Painter et al., 2016). Provided that GNSS data are accurate, the most common reason for misalignment of point clouds is boresight angle errors (Li et al., 2019). Boresighting is the process of calculating the differences between lidar sensor and IMU roll, pitch, and yaw angle measurements to correct those errors in point clouds. Due to battery flight time limitations, we were unable to complete the flight pattern that is commonly used for boresighting alignment. Because of this, we leveraged our first two antiparallel flight lines for boresighting calibration. Additional details on boresighting calibration, our technique due to the flight time limitations, and examples of roll and pitch alignment errors observed during this field campaign appear in the supplemental materials.

Lines 333 to 334: Heavy payload=short flight duration=small area mapped, hence better ground point density. While that makes sense, can't that be achieved by slower speed, closer passes etc.? And mapping extent, of course can be larger if more missions are used. So, I was puzzled what this paragraph was really trying to say.
**A. This is a reasonable comment led to a modification of section 4.3 UAS Sampling Strategies to include a brief paragraph which appears at the end of the response.**

**This comment reflects a general challenge that occurs when developing a spatial sampling strategy in which, for given resources, there is a trade-off between spatial extent and sampling density. An additional point is that the survey height can also be varied with higher altitudes increasing the spatial extent with trade-offs between the point density and**

**number of missions. The main point was intended to provide the reader with the means to contrast this study's sampling densities and the proportion of areas that are masked due to no ground returns with those from previous airborne lidar SD studies.**

**A second point was added to a separate section to respond to the reviewer's insights that regarding the trade-offs between using a UAV versus an airborne platform. While we agree in theory that "Heavy payload=short flight duration=small area mapped, hence better ground point density." could be achieved by "slower speed, closer passes etc." by an airborne platform, if the mapped area has a limited domain then using an airborne platform is probably overkill and inefficient for many studies. Similarly, the "mapping extent, of course can be larger if more missions are used", but as the domain increases in size, much of the battery power would be used to reach the outer limits of the domain and the ability to maintain the required line of sight could also limit the domain. Thus, there are end-members for survey site or regions where it is self-evident as to whether a UAV or an airborne platform should be used, but that leaves considerable gray areas where an appropriate choice of UAV platform and a well designed mission could stretch the domain. Future research and technological advances is needed to offer insights for snow science observation platforms and trade-offs.**

**Finally, slower flights and lower altitude do increase the point density, but further limit the area covered.  We used three sets of batteries and flew over 2 hr period to collect our images.  Limitations on battery cost and time to fly restrict data collection. Flights over multiple days are not appropriate because snowpacks can change within 24 hours.**

**Lines 475 to 486** A well understood challenge exists when developing a spatial sampling strategy in which, for given resources, there is a trade-off between spatial extent and sampling density (Clark et al. 2011). Increasing flight altitude can expand the spatial extent of an aerial survey. However, flying at higher altitudes results in a decreased point density. In theory, a higher point density could be achieved by slower speeds and increased swath overlap. The targeted spatial extent of an aerial survey dictates whether a manned aircraft or a UAV platform should be used. If the targeted area has a limited domain then using a manned airborne platform is probably overkill and inefficient for many studies and the use of a UAV would be more cost effective. However, as the domain increases in size, additional batteries would be required, much of the battery power would be used to reach the outer limits of the domain, and the ability to maintain the required line of sight could be difficult. Thus, there are end-members for survey site or regions where it is self-evident as to whether a UAV or an airborne platform should be used, but that leaves considerable gray areas where an appropriate choice of UAV platform with a well designed mission could stretch the domain.  Future research and technological advances are needed to offer insights for snow science observation platforms and trade-offs.

**The following references were added to the manuscript based on Reviewer 2's input:**

Adams, M.S., Bühler, Y., & Fromm, R. (2018). Multitemporal accuracy and precision assessment of unmanned aerial system photogrammetry for slope-scale snow depth maps in Alpine terrain. *Pure and Applied Geophysics, 175*, 3303-3324

Bühler, Y., Adams, M.S., Bösch, R., & Stoffel, A. (2016). Mapping snow depth in alpine terrain with unmanned aerial systems (UASs): potential and limitations. *The Cryosphere, 10*, 1075-1088
Harder, P., Schirmer, M., Pomeroy, J., & Helgason, W. (2016). Accuracy of snow depth estimation in mountain and prairie environments by an unmanned aerial vehicle. *The Cryosphere, 10*, 2559

**Snow depth mapping with unpiloted aerial systems lidar observations: A case study in Durham, New Hampshire, United States**

5   Jennifer M. Jacobs[1,2], Adam G. Hunsaker[1,2], Franklin B. Sullivan[2], Michael Palace[2,3], Elizabeth A. Burakowski[2], Christina Herrick[2], Eunsang Cho[1,2]

[1]Department of Civil and Environmental Engineering, University of New Hampshire, Durham, NH, 03824, USA
[2]Earth Systems Research Center, Institute for the Study of Earth, Oceans, and Space, University of New Hampshire, Durham, NH, 03824, USA
10   [3]Department of Earth Sciences, University of New Hampshire, Durham, NH, 03824, USA

*Correspondence to*: Jennifer M. Jacobs (Jennifer.jacobs@unh.edu)

**Abstract.** Terrestrial and airborne laser scanning and structure from motion techniques have emerged as viable methods to map snow depths. While these systems have advanced snow hydrology, these techniques have noted limitations in either

15   horizontal or vertical resolution. Lidar on an unpiloted aerial vehicle (UAV) is also a potential method to observe field and slope scale variations at the vertical resolutions needed to resolve local variations in snowpack depth and to quantify snow depth when snowpacks are shallow. This paper provides some of the earliest snow depth mapping results on the landscape scale that were measured using lidar on a UAV. The system, which uses modest cost, commercially available components, was assessed in a mixed deciduous and coniferous forest and open field for a thin snowpack (< 20 cm). The lidar classified

20   point clouds had an average of 90 and 364 points/m$^2$ ground returns in the forest and field, respectively. In the field, in-situ and lidar mean snow depths, at 0.4 m resolution, had a mean absolute difference of 0.96 cm and a root mean squared error of 1.22 cm. At 1 m resolution, the field snow depth confidence intervals were consistently less than 1 cm. The forest and heavily vegetated areas had modestly reduced performance with typical confidence intervals within 4 cm. Although the mean snow depth was only 10.3 cm in the field and 6.0 cm in the forest, a pairwise Steel-Dwass test showed that snow

25   depths were significantly different between the coniferous forest, the deciduous forest, and the field land covers (p < 0.0001). Snow depths were shallower and snow depth confidence intervals were higher in areas with steep slopes. Results of this study suggest that performance depends on both the point cloud density, which can be increased or decreased by modifying the flight plan over different vegetation types, and the within cell variability that depends on site surface conditions.

**1 Introduction**

Snowpacks are highly dynamic, accumulating and ablating throughout the winter with associated changes in snowpack density, grain size, and albedo (Adolph et al., 2017) as well as ice formation. Wind redistribution, sloughing of snow-off slopes, trapping of snow by vegetation, and forest canopy interception result in a range of spatial features at varying scales (Clark et al., 2011; Mott et al., 2011; Mott et al., 2018). Modest differences in snowpack depth can differentially impact

35   many hydrologic, agricultural, and ecosystem processes. Differences in snowpack meltwaters can alter streamflow volumes (Gichamo and Tarboton, 2019), change the likelihood of spring floods (Tuttle et al., 2017) and intensify overland nutrient transport and soil erosion (Seyfried et al., 1990; Singh et al., 2009).

High-resolution snow depth measurements are also needed to discern processes that depend on the snow state. Insulation by

40   seasonal snow in the Arctic and Antarctic slows sea ice growth (Sturm et al., 2002). High-resolution Arctic snow depths

from ICE-Sat2 revealed seasonal snow on ice that would be missed when using coarser snow information (Kwok et al. 2020). Thin, ephemeral snowpacks have limited insulation and allow the underlying soils to freeze more readily in the winter (Groffman et al., 2001; Starkloff et al. 2017; Yi et al. 2019). Soil frost severity impacts soil respiration, carbon sequestration, nutrient retention, and microbial communities as well as a plant root health and tree growth (Aase and Siddoway, 1979; Isard and Schaetzel, 1998; Monson et al., 2006; Henry, 2008; Aanderud et al., 2013; Tucker et al., 2016; Sorensen et al., 2018; Reinmann and Templer, 2018). Detection and mapping of rapid thinning of snowpacks followed by frigid cold during "winter whiplash" events (Casson et al. 2019) is therefore important for understanding ecosystem impacts of soil freezing events, which are otherwise not well quantified (Kraatz et al. 2018; Prince et al. 2019). High vertical resolution snow mapping would greatly improve our understanding of these unique habitats.

Distributed modeling and mapping of snowpacks can increasingly provide output at fine spatiotemporal scales but snow state change validation typically relies on in situ observations (Gichamo and Tarboton 2019; Starkloff et al., 2017). Despite importance, few spatially continuous high-resolution snowpacks datasets are available to support modelling, and mapping efforts. Because snowpacks have considerable spatiotemporal variability, a large number of snow depth measurements are often needed to characterize the snowpack (Dickinson and Whiteley, 1972). Using traditional, precise point measurements with a limited sample size, the experimental design requires a balance between the sampling extent and sample spacing (Clark et al. 2011). However, the choice of sampling resolution may yield different measures of snow depth spatial variability when the snow exhibits multifractal behaviour (Deems et al. 2006).

[revised manuscript text omitted]

Jennifer Jacobs 7/13/2020 11:32 AM

Jennifer Jacobs 7/13/2020 11:33 AM

Jennifer Jacobs 7/13/2020 11:35 AM

Jennifer Jacobs 7/13/2020 11:35 AM
**Moved down [1]:** Automated flights were conducted using UgCS flight planning software.

Jennifer Jacobs 7/13/2020 11:36 AM

Jennifer Jacobs 7/13/2020 11:35 AM
**Moved (insertion) [1]**

Jennifer Jacobs 7/13/2020 11:36 AM

Jennifer Jacobs 7/13/2020 11:39 AM

Jennifer Jacobs 7/13/2020 11:37 AM

Jennifer Jacobs 7/13/2020 11:38 AM

Jennifer Jacobs 7/13/2020 11:38 AM

Jennifer Jacobs 7/13/2020 11:39 AM

Jennifer Jacobs 7/13/2020 11:40 AM

Jennifer Jacobs 7/13/2020 11:40 AM

Jennifer Jacobs 7/13/2020 11:41 AM

Jennifer Jacobs 7/13/2020 11:41 AM

across 8 computing cores to improve efficiency. PMF was parameterized using a set of window sizes of 1, 3, 5, and 9 m, and elevation thresholds of 0.2, 1.5, 3, and 7 m, which were determined by varying value sets and assessing digital terrain models (DTMs) to determine the parameter sets that produced a visually smooth surface over a dense grid (*in sensu* Muir et al., 2017). Following ground classification for each tile, returns within the 15 m tile buffers were removed, and all resulting 100 m square ground classified tiles were merged. The resulting point clouds for each data set included both the classified ground returns and the non-ground returns. Snow-on and snow-off ground point clouds were gridded at 0.1, 0.2, 0.4, 0.5, and 1.0 m spatial resolutions using the average of all grid points within each grid cell (Currier et al., 2019). Gridded products for each data set were forced to the same coordinate grid to generate DTMs as raster files.

**2.4 Slope and Vegetation Cover Classification and Analysis**

The snow-off DTM was used to develop a 1 m resolution map of slope (Horn, 1981). Vegetation cover type (field/forest) was determined from the known boundaries of field and forest. The forested area was further classified as coniferous or deciduous for the study region using the following methodology (Figure 1). Within the forested area (Figure 1), a Canopy Height Model (CHM) was used to distinguish the intact upper canopy from other forest cover using our snow-off survey, collected with leaf off in the spring (Sullivan et al., 2017). The CHM was generated by subtracting the DTM produced using ground-classified points from the DSM produced using all lidar points. This results in a digital model consisting solely of canopy heights with no terrain or topography. The CHM generation used raster images with a 1 m resolution. A 3 by 3 maximum convolve filter was used to enhance the edges of canopy crowns and expand smaller regions that might have just one pixel of an intact canopy or a hole in a larger canopy (Palace et al., 2008). A 15 m threshold was used to differentiate between the upper level intact coniferous canopy. CHM pixels that were below this threshold were deemed deciduous canopies (see Figure S3 in supporting information for intermediate figure). The 5.6 ha forested area has a forest type that is 65% deciduous and 35% coniferous.

[revised manuscript text omitted]

Jennifer Jacobs 7/13/2020 11:52 AM
**Moved (insertion) [2]**

Jennifer Jacobs 7/13/2020 11:52 AM

Jennifer Jacobs 7/13/2020 11:56 AM

Jennifer Jacobs 7/13/2020 12:00 PM
**Moved (insertion) [3]**

Jennifer Jacobs 7/13/2020 12:00 PM

Jennifer Jacobs 7/27/2020 5:03 PM

Forest Boundary
Field Boundary
Transect Plots
West Forest
Field
East Forest
Snow Depth
35cm
0 cm
0   25   50        100 m          N
0   75  150      300 ft

Unknown

[Figure]

**Unknown**

**Figure 5.** Average (top) snow depth values, (middle) one sided confidence intervals, and (bottom) topography and forest cover type. Snow depth and confidence intervals calculated from the snow-on and snow-off lidar point clouds for 1 m$^2$ cells at Thompson Farm, Durham, NH. Topography and forest cover type determined from snow-off lidar point clouds on snow-off flight for 1 m$^2$ cells conducted on April 11, 2019.

570

[Figure]

[Figure]

**Unknown**

Jennifer Jacobs 7/13/2020 12:04 PM

95% CI

- 0-0.5 cm
- ≤ 1
- ≤ 1.5
- ≤ 2
- ≤ 2.5
- ≤ 3
- ≤ 3.5
- ≤ 4
- >4cm

Forest Boundary
Field Boundary
Transect Plots
□ West Forest
○ Field
△ East Forest

N

0 25 50 100 m
0 75 150 300 ft

[revised manuscript text omitted]

<table>
<tr><td>Page 10: [4] Deleted</td><td>Jennifer Jacobs</td><td>7/13/20 11:52 AM</td></tr>
</table>

<table>
<tr><td>Page 10: [4] Deleted</td><td>Jennifer Jacobs</td><td>7/13/20 11:52 AM</td></tr>
</table>

<table>
<tr><td>Page 10: [5] Deleted</td><td>Jennifer Jacobs</td><td>7/13/20 11:56 AM</td></tr>
</table>

In the field, the precision of the mean snow depth estimate is

<table>
<tr><td>Page 10: [5] Deleted</td><td>Jennifer Jacobs</td><td>7/13/20 11:56 AM</td></tr>
</table>

In the field, the precision of the mean snow depth estimate is

<table>
<tr><td>Page 10: [5] Deleted</td><td>Jennifer Jacobs</td><td>7/13/20 11:56 AM</td></tr>
</table>

In the field, the precision of the mean snow depth estimate is

<table>
<tr><td>Page 10: [5] Deleted</td><td>Jennifer Jacobs</td><td>7/13/20 11:56 AM</td></tr>
</table>

In the field, the precision of the mean snow depth estimate is

<table>
<tr><td>Page 10: [5] Deleted</td><td>Jennifer Jacobs</td><td>7/13/20 11:56 AM</td></tr>
</table>

In the field, the precision of the mean snow depth estimate is

<table>
<tr><td>Page 10: [5] Deleted</td><td>Jennifer Jacobs</td><td>7/13/20 11:56 AM</td></tr>
</table>

In the field, the precision of the mean snow depth estimate is

<table>
<tr><td>Page 10: [5] Deleted</td><td>Jennifer Jacobs</td><td>7/13/20 11:56 AM</td></tr>
</table>

In the field, the precision of the mean snow depth estimate is

<table>
<tr><td>Page 10: [6] Deleted</td><td>Jennifer Jacobs</td><td>7/13/20 12:00 PM</td></tr>
</table>

Snow depth precision was

| Page 10: [6] Deleted | Jennifer Jacobs | 7/13/20 12:00 PM |

Snow depth precision was

| Page 10: [6] Deleted | Jennifer Jacobs | 7/13/20 12:00 PM |

Snow depth precision was

| Page 10: [6] Deleted | Jennifer Jacobs | 7/13/20 12:00 PM |

Snow depth precision was

| Page 10: [6] Deleted | Jennifer Jacobs | 7/13/20 12:00 PM |

Snow depth precision was

| Page 10: [6] Deleted | Jennifer Jacobs | 7/13/20 12:00 PM |

Snow depth precision was

| Page 10: [6] Deleted | Jennifer Jacobs | 7/13/20 12:00 PM |

Snow depth precision was

| Page 10: [6] Deleted | Jennifer Jacobs | 7/13/20 12:00 PM |

Snow depth precision was

[revised manuscript text omitted]

---

## Referee Report (RR1)

**Review #2 for The Cryosphere Discussion - 2020-09-09**
**Snow depth mapping with unpiloted aerial systems lidar observations: a case study in Durham, New Hampshire, United States**

Jacobs et al. provided substantial improvements in the manuscript structure and scientific content. Adding the comparison between forest type is especially interesting (Figure 6 and S3). However, some parts are still a bit confusing and would need some work to ease the readers' comprehension of the work. I think the article can be recommended for publication if the following minor points are addressed.

The two main points are :

1. The comparison with the work of Harder et al. 2020 seems a bit light. I think it is beneficial for the community to have several validation of similar methods published, but it would be interesting to highlight what are the main differences between the two study methods (in the introduction) and between their results (in the discussions).

2. I appreciate the clarification brought by the authors about the terms « precision » and « accuracy » as requested in my previous review. However the use of the «within cell variability » term is still a source of confusion for me. The « local scale variability »(L349)/« within cell variability » (L351) is measured with the « standard deviation of the lidar elevation values ». What is this standard deviation ? Please provide the formula or a clear description of it in the methods. Is there one standard deviation for each DTM or is it a pixel-based metric ? Is it related to sigma_on and sigma_off used to calculate the one-sided confidence interval (equation 1) ? If they are related, please make clear why one is the combination of the lidar accuracy and surface elevation variation (one-sided confidence interval, L234) and the other is the local scale variability of the ground surface elevation (L349).

Below are minor points.

Title : Why was « unmanned » replaced with « unpiloted » ? I cannot find the term « unpiloted » in the snow depth UAS literature cited in the article (De Michele et al., 2016, Bühler et al., 2016, Adams et al., 2018, Harder et al., 2020, Eberhard et al., 2020).

L61 : the citation of Deems 2013 and Lopez-Moreno 2017 suggests that they made a review of remote sensing methods for snow depth mapping while these article focus respectively on ALS and TLS. I would remove these citations as the list of the remote sensing methods is given in the next sentence.

L62 : ALS is airborne laser scanning, fine. L196 : « UAS laser scanning » becomes ALS. Confusing, especially in discussion : « the UAS lidar surveys presented in this study have key differences from previous ALS surveys » L458 and in the following paragraph. I would use two acronyms for airborne laser scanning (ALS since at least Deems et al., 2013) and UAS laser scanning (ULS ?…)

Deems JS, Painter TH and Finnegan DC (2013) Lidar measurement of snow depth : a review. *J. Glaciol.* **59**(215), 467–479 (doi:10.3189/2013JoG12J154)

L76 : This sentence is too long. Split or reduce. Name the « snowpack features » ?

L100 : now we want to know more about this paper...what do they conclude ? What does your work add ?

L141 : interesting information, but please explain the « to 3.6 V per cell » Cells are never mentionned before.

L150 : please expand a bit on the geo-referencement. Is it adjusting one point-cloud relatively to the other ? Or are they completely individually geo-referenced ?

L174 : How are the boundaries of the field and forest « known » ? External dataset ?

L175 : Maybe remove the second « (Figure 1) ».

L173-185 : I am a bit confused : « intact canopy » means with needles/leaves in winter ? Is it that the needless branches backscatter less the laser, making the deciduous tree canopy appear lower ? Add a sentence or two to explain that, although Figure S3 provides a good visual intuition of what is happening.

L186 : Why extract 5000 points and not use all available points ? Even at the highest resolution (0.1 m) the 0.1 km² raster should represent ~$10^7$ points which is not computationally unbearable. To be considered in future work.

L186 : I think you can simplify. « Once the vegetation forest type was classified, the raster binary image was vectorized. » is not really necessary. Something like : « Three sets of 5000 points were extracted respectively in the field, in the eastern forest and in the western forest. At each of these random points... »

L191 : Has this test ever been used in geophysics studies ? It would be good to cite other papers using it. It is otherwise really hard to assess the relevance of this test without extensive statistical background. If it is relevant, this is a very welcome technic to estimate significant snow depth differences.

L229 : Consider giving the one-sided confidence interval equation.

L229 : « of the snow on and snow off elevation » is confusing. Could you remove it ?

L233 : « of the snow-on and snow-off **ground** returns » ?

L264 : «**absolute** low bias ». In absolute yes, in relative no.

L266 : « mean height profiles » does not seem right. The heights are not averaged, they are normalized. It is rather a distribution of the normalized elevation, or something along that.

L272 : bring « respectively » earlier in the sentence.

L321 : confidence interval **decreases**

L346 : « the ability to capture the mean snow depth » I disagree: one-sided confidence interval still includes natural infra-pixel variability. You might perfectly capture the mean of a pixel with high infra-pixel variability and still have a large confidence interval.

L359-L363 : this paragraph is a bit confusing. It would benefit from i) defining better the standard deviation (see main comment 2. above) and ii) explain the relationship between point density per cell and cell size. The last sentence of the paragraph implies that reducing the cell size reduces the ground return density. This should be explained.
L359 : **point cloud density** instead of « the point per cell »
L360 : **snow depth map resolution** instead of « spatial scales »

L387 : « reduced accuracy of the GNSS » : in forest ?

Paragraph from L410 : This paragraph is a bit hard to read. It should be simplified and the conclusions better highlighted. See some suggestions below.

L412 : « While this result is not entirely surprising » : little added value of this sentence. I suggest removing it.
L414 : « in snow depth due to pockets of duff and woody debris, and due to higher variability in subnivean terrain in the forested areas of the study site » are duff and woody debris not part of the subnivean terrain ?
L414 : is it really related to « variability in snow depth » ? or variability in snow-off terrain since this is the only thing discussed after.

L416 : Long sentence. Consider shortening with « drive higher confidence interval **in these areas.** »

L416 : High relief terrain is defined by high elevation variability over short distances. Also, I am not sure what this sentence is stating.

L418 : remove « of the study site »

L419 : Interesting result. Please repeat which is higher than which (deciduous, coniferous, field). Cite literature showing the canopy interception effect. Is it not surprising that deciduous and coniferous trees have the same impact interception ?

L426, L440, you mention large UAV and heavy payloads. L372  the UAV seemed to be light (<25 kg) and small in size. Please homogenize.

L463 : SD ?

L475 : please say what area is typically covered by airborne laser scanning campaign.

Figure 4 : In a and c a certain amount of points classified as non-ground is centered on 0 m height. This seems to be points which are wrongly classified. Is this assumption true ? Are they at specific locations ? Consider adding something about this in the results.

Figure 6 a and b : there are some negative snow depth. This is possible due to the uncertainty of snow-on and snow-off DTM but should be presented in the results.

Figure 8. a.: See main comment about the standard deviation of lidar elevation. Here it seems to be a general metric : one standard deviation per snow depth map.

---

## Referee Report (RR2)

This paper presents a snow depth map methodology for a UAV and LIDAR combination to measure thin snowpack of ephemeral snow. The study site contains a low vegetated field with a mixed forest which is useful to evaluate vegetation interaction. The main question focuses on the capability of LIDAR for snow depth mapping mounted on UAV because the sensor combination of LIDAR and UAV had not been extensively published yet. Most of the paper evaluates the accuracy of the map with respect to point cloud density (link to DTM resolution and LIDAR returns), vegetation cover and slope. A substantial amount of methodology and flight experience makes this paper focus on technical and methodological issues rather than informing us on snow processes of thin and ephemeral snow with vegetation interaction.

I do not think another methodological paper on snow depth mapping would be beneficial to the Cryosphere community. Differential snow depth mapping from LIDAR dates to the beginning of the century (Deems et al., 2006; Hopkinson et al., 2004) with airborne data from plane quickly became a more efficient tool to map large areas. A numerous numbers of article have used airborne LIDAR data for snow depth mapping (Currier et al., 2019; Grünewald et al., 2013; Hopkinson et al., 2012, 2004; Mazzotti et al., 2019; Nolan et al., 2015; Painter et al., 2016) with development on data processing (see review from Deems et al., 2013) and topographic and vegetation induced errors on LIDAR DEM (Spaete et al., 2011). Since the UAV platform only change the altitude and the coverage area, I do not see the point of having another methodological article. Regarding flight experience with UAV, a substantial amount of paper regarding snow mapping has already been published for smaller area with multi-rotor systems (Buhler et al., 2016; Cimoli et al., 2017; Fernandes et al., 2018) and larger area with fixed wing systems (De Michele et al., 2016; Harder et al., 2016; Redpath et al., 2018).

The high error in forested environment clearly need to be more investigated. It is stated that the magnaprobe measurements overestimate the snow depth in forested environment by penetrating low lying vegetation and soil. So, is the LIDAR mean depth of 6cm in more representative than magnaprobe (15 cm) and federal sampling tube (13 cm) which both showed larger depth for forested than field? Some forested areas can have deeper snowpack (Trujillo et al., 2009) due to wind reduction but snow interception by canopy will decrease snow accumulation on the ground as forest cover increases (Varhola et al., 2010). This needs to be sorted out especially when it is know that differential snow depth mapping underestimate snow for small vegetation as the snow off DEM is higher than the bare ground but when snow is measured for truth scene, the same vegetation is compressed by the weight of the snow (Buhler et al., 2016; Nolan et al., 2015).

The paper lacks a novel or in depth analysis of relations or processes regarding snow derived from a snow map that would be beneficial to the community like for e.g. a comparison with SfM photogrammetry in forested environments (Harder et al., 2020), statistical analysis of snow depth (Grünewald et al., 2013; Wainwright et al., 2017) or spatial analysis with variogram or fractal analysis between open field and forested environments (Deems et al., 2006; Redpath et al., 2018; Trujillo et al., 2009). To do this, I would suggest trying to improve with additional dataset over wider areas, only one site is not enough or perhaps different seasons to explore temporality and resubmit to this journal after next field season or go with this version towards a technical journal regarding UAV.

**References**

Buhler, Y., Adams, M.S., Bosch, R., Stoffel, A., 2016. Mapping snow depth in alpine terrain with unmanned aerial systems (UASs): Potential and limitations. Cryosphere 10, 1075–1088. https://doi.org/10.5194/tc-10-1075-2016

Cimoli, E., Marcer, M., Vandecrux, B., Bøggild, C.E., Williams, G., Simonsen, S.B., 2017. Application of low-cost uass and digital photogrammetry for high-resolution snow depth mapping in the Arctic. Remote Sens. 9, 1–29. https://doi.org/10.3390/rs9111144

Currier, W.R., Pflug, J., Mazzotti, G., Jonas, T., Deems, J.S., Bormann, K.J., Painter, T.H., Hiemstra, C.A., Gelvin, A., Uhlmann, Z., Spaete, L., Glenn, N.F., Lundquist, J.D., 2019. Comparing Aerial Lidar Observations With Terrestrial Lidar and Snow-Probe Transects From NASA's 2017 SnowEx Campaign. Water Resour. Res. 55, 6285–6294. https://doi.org/10.1029/2018WR024533

De Michele, C., Avanzi, F., Passoni, D., Barzaghi, R., Pinto, L., Dosso, P., Ghezzi, A., Gianatti, R., Vedova, G. Della, 2016. Using a fixed-wing UAS to map snow depth distribution: An evaluation at peak accumulation. Cryosphere 10, 511–522. https://doi.org/10.5194/tc-10-511-2016

Deems, J.S., Fassnacht, S.R., Elder, K.J., 2006. Fractal Distribution of Snow Depth from Lidar Data. J. Hydrometeorol. 7, 285–297. https://doi.org/10.1175/JHM487.1

Deems, J.S., Painter, T.H., Finnegan, D.C., 2013. Lidar measurement of snow depth: A review. J. Glaciol. 59, 467–479. https://doi.org/10.3189/2013JoG12J154

Fernandes, R., Prevost, C., Canisius, F., Leblanc, S.G., Maloley, M., Oakes, S., Holman, K., Knudby, A., 2018. Monitoring snow depth change across a range of landscapes with ephemeral snowpacks using structure from motion applied to lightweight unmanned aerial vehicle videos. Cryosphere 12, 3535–3550. https://doi.org/10.5194/tc-12-3535-2018

Grünewald, T., Stötter, J., Pomeroy, J.W., Dadic, R., Moreno Baños, I., Marturià, J., Spross, M., Hopkinson, C., Burlando, P., Lehning, M., 2013. Statistical modelling of the snow depth distribution in open alpine terrain. Hydrol. Earth Syst. Sci. 17, 3005–3021. https://doi.org/10.5194/hess-17-3005-2013

Harder, P., Pomeroy, J.W., Helgason, W.D., Helgason, W.D., 2020. Improving sub-canopy snow depth mapping with unmanned aerial vehicles: Lidar versus structure-from-motion techniques. Cryosphere 14, 1919–1935. https://doi.org/10.5194/tc-14-1919-2020

Harder, P., Schirmer, M., Pomeroy, J., Helgason, W., 2016. Accuracy of snow depth estimation in mountain and prairie environments by an unmanned aerial vehicle. Cryosphere. https://doi.org/10.5194/tc-10-2559-2016

Hopkinson, C., Collins, T., Anderson, A., Pomeroy, J., Spooner, I., 2012. Spatial snow depth assessment using LiDAR transect samples and public GIS data layers in the Elbow River Watershed, Alberta. Can. Water Resour. J. 37, 69–87. https://doi.org/10.4296/cwrj3702893

Hopkinson, C., Sitar, M., Chasmer, L., Treitz, P., 2004. Mapping Snowpack Depth beneath Forest Canopies Using Airborne Lidar. Photogramm. Eng. Remote Sens. 70, 323–330. https://doi.org/10.14358/PERS.70.3.323

Mazzotti, G., Currier, W.R., Deems, J.S., Pflug, J.M., Lundquist, J.D., Jonas, T., 2019. Revisiting Snow Cover Variability and Canopy Structure Within Forest Stands: Insights From Airborne Lidar Data. Water Resour. Res. 55, 6198–6216. https://doi.org/10.1029/2019WR024898

Nolan, M., Larsen, C., Sturm, M., 2015. Mapping snow depth from manned aircraft on landscape scales

at centimeter resolution using structure-from-motion photogrammetry. Cryosphere 9, 1445–1463. https://doi.org/10.5194/tc-9-1445-2015

Painter, T.H., Berisford, D.F., Boardman, J.W., Bormann, K.J., Deems, J.S., Gehrke, F., Hedrick, A., Joyce, M., Laidlaw, R., Marks, D., Mattmann, C., McGurk, B., Ramirez, P., Richardson, M., Skiles, S.M.K., Seidel, F.C., Winstral, A., 2016. The Airborne Snow Observatory: Fusion of scanning lidar, imaging spectrometer, and physically-based modeling for mapping snow water equivalent and snow albedo. Remote Sens. Environ. https://doi.org/10.1016/j.rse.2016.06.018

Redpath, T.A.N., Sirguey, P., Cullen, N.J., 2018. Repeat mapping of snow depth across an alpine catchment with RPAS photogrammetry 3477–3497.

Spaete, L.P., Glenn, N.F., Derryberry, D.R., Sankey, T.T., Mitchell, J.J., Hardegree, S.P., 2011. Vegetation and slope effects on accuracy of a LiDAR-derivedDEMin the sagebrush steppe. Remote Sens. Lett. https://doi.org/10.1080/01431161.2010.515267

Trujillo, E., Ramírez, J.A., Elder, K.J., 2009. Scaling properties and spatial organization of snow depth fields in sub-alpine forest and alpine tundra. Hydrol. Process. https://doi.org/10.1002/hyp.7270

Varhola, A., Coops, N.C., Weiler, M., Moore, R.D., 2010. Forest canopy effects on snow accumulation and ablation: An integrative review of empirical results. J. Hydrol. 392, 219–233. https://doi.org/10.1016/j.jhydrol.2010.08.009

Wainwright, H.M., Liljedahl, A.K., Dafflon, B., Ulrich, C., Peterson, J.E., Gusmeroli, A., Hubbard, S.S., 2017. Mapping snow depth within a tundra ecosystem using multiscale observations and Bayesian methods. Cryosph. 11, 857–875. https://doi.org/10.5194/tc-11-857-2017

---

## Author Response (AR2)

Response to Editor

Dear Dr. Marsh

We are grateful to the four reviewers for their detailed and constructive comments on the manuscript and particularly to those reviewers who reviewed the original and revised submission.

In response to your specific requests, we have
- highlighted what are the main differences between the two study methods (in the introduction) and between their results (in the discussion). Details are provided in the first response to Reviewer #1.
- addressed referee #1s question concerning the use of the term "within cell variability" by simplifying the explanation, adding equations to explain our calculations and adding text that references Deems et al. (2013) to explain that the source of the variability is due to sampling limitations. We hope that this will allow readers understand this part of our paper,
- we have addressed referee #4s comments concern a common approach to report on accuracy that is used in much of the snow mapping papers to date that is an additional point that is similar to referee #1, and
- we have addressed the other numerous questions raised by all referees.

Our response to each comment is outlined below in **bold** and revised text is in red. The copy of the track changes manuscript is provided following our responses to the reviewers. We hope these responses are clear, and we look forward to submitting the revised manuscript.

Regards, Jennifer

Anonymous Referee #1

*Thank you for the detailed comments and the opportunity to clarify that this article is the first to present snow depth maps measured with UAS-based lidar. We have provided detailed responses to the reviewer following each of the reviewer's comments.*

Jacobs et al. provided substantial improvements in the manuscript structure and scientific content. Adding the comparison between forest type is especially interesting (Figure 6 and S3). However, some parts are still a bit confusing and would need some work to ease the readers' comprehension of the work. I think the article can be recommended for publication if the following minor points are addressed.
The two main points are :
1. The comparison with the work of Harder et al. 2020 seems a bit light. I think it is beneficial for the community to have several validation of similar methods published, but it would be interesting to highlight what are the main differences between the two study methods (in the introduction) and between their results (in the discussions).
**We reviewed the earlier manuscript and concur that our manuscript is the first UAS-based lidar snow depth mapping manuscript when it was reviewed and during our revision. Their manuscript was not available when we originally submitted our contribution in February 2020. Only days before our resubmission, on June 15th, the Harder et al. 2020 was published. We appreciate the opportunity to now revise our manuscript to include the requested comparison.**

**The introduction was revised as follows:**
Harder et al. (2020) compared snow depth estimates between lidar versus SfM techniques using in-situ snow depth observations in mountain and prairie environments, focusing on sub-canopy snow, which has been a challenge to measure in the snow remote sensing community. Using a considerably more expensive UAS-based lidar system (~$300K Canadian), they found that the lidar system tends to have lower errors than the SfM to capture sub-canopy snow distributions at moderate depth of snowpack (up to 2 m and 1 m of the maximum depth for mountain and prairie areas, respectively). In this study, we assess the ability of a more modest cost UAS lidar system (~$70K U.S. dollars) to map snow depth focusing on shallow and ephemeral snowpack (< 20 cm).

**We also add a comparison of this study's results to their results in the discussion section as follows:**
This study's lidar snow depth performance metrics are comparable to those from the more extensive lidar surveys made Harder et al. (2020), In the field, our snow depth errors, 1 cm bias and 1.2 cm RMSD, were modestly better than those from their open sites snow depth 3 cm bias and RMSE values on the order of 10 cm. In the forest, our snow depth errors, 7 cm bias and 10 cm RMSD, were also modestly lower than those from their forest sites 9 to 13 cm bias and 15 cm RMSE. While it is difficult to make direct comparison across different study sites, snow conditions, and ground validation approaches, these early findings indicate that UAS lidar has the capability of mapping snow depths in open and forested regions and has improved performance as compared to previous SfM results particularly for vegetated surface. It is also noteworthy that this

study's mapping was conducted using the Velodyne Puck series, a laser scanner adapted from the assisted and autonomous vehicle applications, rather than the specialized Riegl miniVUX-1UAV used by Harder et al. (2020) resulting in a complete mapping system that was approximately one-third the costs of their Riegl system.

2. I appreciate the clarification brought by the authors about the terms « precision » and « accuracy » as requested in my previous review. However the use of the «within cell variability » term is still a source of confusion for me. The « local scale variability »(L349)/« within cell variability » (L351) is measured with the « standard deviation of the lidar elevation values ». What is this standard deviation ? Please provide the formula or a clear description of it in the methods. Is there one standard deviation for each DTM or is it a pixel-based metric ? Is it related to sigma_on and sigma_off used to calculate the one-sided confidence interval (equation 1) ? If they are related, please make clear why one is the combination of the lidar accuracy and surface elevation variation (one-sided confidence interval, L234) and the other is the local scale variability of the ground surface elevation (L349).

**A. The authors appreciate the reviewer's recommendations for clarification. A description of the standard deviation was added. The standard deviation is a pixel-based metric and is equivalent to the sigma_on and sigma_off used to calculate the one-sided confidence interval. The term terms referenced above were replaced by a single term "grid cell variability" which is now used consistently throughout the manuscript. The text now reads "The snow-on and snow-off standard deviation is calculated for each individual grid cell using all the individual snow-on and snow-off lidar ground return elevations, respectively. These standard deviations, measures of the variability of the snow-on and snow-off lidar ground returns within a grid cell, are referred as the grid cell variability. This variability depends on the lidar instrument's relative accuracy (Maune and Nayegandhi, 2018), which includes intra-swatch accuracy (i.e., precision or repeatability of measurements) and inter-swath accuracy (i.e., differences in elevations between overlapping swaths), as well as surface elevation variations. The contribution from the individual sources of variability was not assessed."**

Below are minor points.
Title : Why was « unmanned » replaced with « unpiloted » ? I cannot find the term « unpiloted » in the snow depth UAS literature cited in the article (De Michele et al., 2016, Bühler et al., 2016, Adams et al., 2018, Harder et al., 2020, Eberhard et al., 2020).
**A. The author team appreciates the comment. Perhaps it is a minor thing, but we would prefer to keep unpiloted.  There has been a movement away from this term (see https://www.planetary.org/articles/10050900-finding-new-language for a discussion regarding NASA's policies).  It seems harmless to go with unpiloted over unmanned, in our opinion. At worst it demonstrates our interest in engaging in inclusivity in the science world (even if the argument can be made that there's no implication of exclusivity in the word unmanned). There also is precedent in the scientific community (Davis et al. 2020; Wagner et al. 2019, among others).**

**Davis, J., Blesius, L., Slocombe, M., Maher, S., Vasey, M., Christian, P. and Lynch,**

**P., 2020. Unpiloted Aerial System (UAS)-Supported Biogeomorphic Analysis of Restored Sierra Nevada Montane Meadows.** *Remote Sensing*, *12*(11), p.1828.

**Wagner, M., Doe, R.K., Johnson, A., Chen, Z., Das, J. and Cerveny, R.S., 2019. Unpiloted Aerial Systems (UASs) Application for Tornado Damage Surveys: Benefits and Procedures.** *Bulletin of the American Meteorological Society*, *100*(12), pp.2405-2409.

L61 : the citation of Deems 2013 and Lopez-Moreno 2017 suggests that they made a review of remote sensing methods for snow depth mapping while these article focus respectively on ALS and TLS. I would remove these citations as the list of the remote sensing methods is given in the next sentence.
**A. Citations were removed.**

L62 : ALS is airborne laser scanning, fine. L196 : « UAS laser scanning » becomes ALS. Confusing, especially in discussion : « the UAS lidar surveys presented in this study have key differences from previous ALS surveys » L458 and in the following paragraph. I would use two acronyms for airborne laser scanning (ALS since at least Deems et al., 2013) and UAS laser scanning (ULS ?...)

Deems JS, Painter TH and Finnegan DC (2013) Lidar measurement of snow depth : a review. *J. Glaciol.* **59**(215), 467–479 (doi:10.3189/2013JoG12J154)

**A. Good point. After review, it appears that despite our statement "UAS lidar surveys (here after noted as ALS measurements)", we did not in fact use ALS to refer to the UAS lidar surveys. Harder et al. (2020) did not use the term ALS either for the UAS lidar surveys. Thus the statement "UAS lidar surveys (here after noted as ALS measurements)" was removed.**

L76 : This sentence is too long. Split or reduce. Name the « snowpack features » ?
**A. Modified as recommended.**

L100 : now we want to know more about this paper...what do they conclude ? What does your work add ?
**A. As noted above, the introduction was revised as follows:**
Harder et al. (2020) compared snow depth estimates between lidar versus SfM techniques using in-situ snow depth observations in mountain and prairie environments, focusing on sub-canopy snow, which has been a challenge to measure in the snow remote sensing community. Using a considerably more expensive UAS-based lidar system (~$300K Canadian), they found that the lidar system tends to have lower errors than the SfM to capture sub-canopy snow distributions at moderate depth of snowpack (up to 2 m and 1 m of the maximum depth for mountain and prairie areas, respectively). In this study, we assess the ability of a more modest cost UAS lidar system (~$70K U.S. dollars) to map snow depth focusing on shallow and ephemeral snowpack (< 20 cm).

L141 : interesting information, but please explain the « to 3.6 V per cell » Cells are never mentionned before.

A. **The authors understand why confusion would arise here. Lithium polymer batteries are composed polymer cells in series. The batteries used on our system were a pair of batteries, each with 6 cells in series. A single cell at full charge has 4.2 V per cell, and 3.6 V at a safe discharge that the rotors can maintain altitude of the system. Rather than distract from the focus of the paper, we opted to remove the specifics about battery discharge, and note the total flight time from ascent to descent of the aircraft.**

L150 : please expand a bit on the geo-referencement. Is it adjusting one point-cloud relatively to the other ? Or are they completely individually geo-referenced ?

A. **Lidar returns were individually georeferenced by synching timestamps of returns from the lidar sensor with timestamps of position and attitude data from the post-processed INS data. Georeferenced point clouds were produced and output to LAS files using Headwall Photonics, Inc.'s LidarTools software. This is now explained in the text as follows:**

The bare-earth and snow-on point clouds were georeferenced solely using the INS data respective to each flight. The point clouds were not co-registered to each other as there were no reliable common ground control points between surveys. For UAS lidar snow depth surveying, co-registration between point clouds would likely be unattainable due to insufficient common ground control. We determined results would be more meaningful when bare-earth and snow-on point clouds were processed solely relying on the capability of the INS.

**Note: A similar reason for not co-registering bare-earth and snow-on point clouds was presented by Goetz and Brenning (2019).**

**Goetz, J., & Brenning, A. (2019). Quantifying uncertainties in snow depth mapping from structure from motion photogrammetry in an alpine area. *Water Resources Research*, 55, 7772– 7783. https://doi.org/10.1029/2019WR025251**

L174 : How are the boundaries of the field and forest « known » ? External dataset ?
A. **Modified to be more specific.**

L175 : Maybe remove the second « (Figure 1) ».
A. **Removed.**

L173-185 : I am a bit confused : « intact canopy » means with needles/leaves in winter ? Is it that the needless branches backscatter less the laser, making the deciduous tree canopy appear lower ? Add a sentence or two to explain that, although Figure S3
**We thank the reviewer for suggesting to tighten this section up. We tried to make our methodology a little clearer and moved some sentences around as well as changing wording. The following section of reads as follows:**

The snow-off DTM was used to develop a 1 m resolution map of slope (Horn, 1981). Vegetation cover type (field/forest) was determined from optical imagery. We developed a Canopy Height Model (CHM) by subtracting the DTM produced using ground-classified points from the DSM produced using all lidar points. This results in a digital model consisting solely of canopy heights with no topography. The CHM generation used raster images with a 1 m resolution. The forested area was further classified as coniferous or deciduous for the study region. Within the forested area, the CHM was used to distinguish the upper canopy that did not lose needles/foliage from other forested regions with trees with no leaves using our snow-off survey that was collected with leaf off in the spring. A 3 by 3 maximum convolve filter was used to enhance the edges of canopy crowns and expand smaller regions that might have just one pixel of an intact canopy or a hole in a larger canopy (Palace et al., 2008). A 15 m threshold was used to differentiate between the upper level intact coniferous canopy and canopies that had lost their leaves. CHM pixels that were below this threshold were deemed deciduous canopies (see Figure S3 in supporting information for intermediate figure). The 5.6 ha forested area has a forest type that is 65% deciduous and 35% coniferous.

L186 : Why extract 5000 points and not use all available points ? Even at the highest resolution (0.1 m) the 0.1 km2 raster should represent ~107 points which is not computationally unbearable. To be considered in future work.
**A. A 0.1 km$^2$ area is 100,000 m$^2$.**

L186 : I think you can simplify. « Once the vegetation forest type was classified, the raster binary image was vectorized. » is not really necessary. Something like : « Three sets of 5000 points were extracted respectively in the field, in the eastern forest and in the western forest. At each of these random points... »
**A. Modified as recommended.**

L191 : Has this test ever been used in geophysics studies ? It would be good to cite other papers using it. It is otherwise really hard to assess the relevance of this test without extensive statistical background. If it is relevant, this is a very welcome technic to estimate significant snow depth differences.
**A. The Steel-Dwass test is a common test in environmental science and has been used in geophysics. It is a non-parametric test that is often used when data is not normally distributed. We have added the following text and citation to the manuscript as suggested by the reviewer.**

The Steel-Dwass test has been previously used in geophysical work to examine non-parametric datasets (Slotznick et al., 2019).

Slotznick, S.P., Sperling, E.A., Tosca, N.J., Miller, A.J., Clayton, K.E., van Helmond, N.A.G.M., Slomp, C.P. and Swanson-Hysell, N.L., 2020. Unraveling the mineralogical complexity of sediment iron speciation using sequential extractions. *Geochemistry, Geophysics, Geosystems*, *21*(2).

L229 : Consider giving the one-sided confidence interval equation.

**A. Reviewer #4 also gave specific suggestions regarding this section and terminology. Modifications were also made in response to Reviewer #1 and Reviewer #4. The paragraph now reads as follows:**

The one-sided width of the 95% confidence limits ($CI_{95\%,+/-}$) for each grid cell's lidar derived estimate of the mean snow depth is a measure of uncertainty. The $CI_{95\%,+/-}$ values are used to compare the reliability of the snow depth estimates among cells. The $CI_{95\%,+/-}$ values were calculated using each grid cell's bare-earth and snow-on pooled sample standard deviation ($s_d$) and the number of bare-earth and snow-on lidar returns (n and m respectively) (Helsel and Hirsh, 2002).

$$CI_{95\%+/-} = t_{crit}s_d\sqrt{\left(\frac{1}{n}+\frac{1}{m}\right)} \tag{1}$$

A cell's pooled sample standard deviation ($s_d$) was calculated as

$$s_d = \sqrt{\frac{(n-1)s_{off}^2+(m-1)s_{on}^2}{(n+m-2)}} \tag{2}$$

where $s_{on}$ and $s_{off}$ are the standard deviations of the snow-on and snow-off lidar ground return elevations, respectively. The $s_{on}$ and $s_{off}$ values are a measure of the grid cell variability. This variability depends on the lidar instrument's relative accuracy (Maune and Nayegandhi, 2018), which includes intra-swatch accuracy (i.e., precision or repeatability of measurements) and inter-swath accuracy (i.e., differences in elevations between overlapping swaths), as well as surface elevation variations and terrain induced errors (Deems et al., 2013). The contribution from the individual sources of variability was not assessed in the current study.

L229 : « of the snow on and snow off elevation » is confusing. Could you remove it ?
**A. Removed as recommended.**

L233 : « of the snow-on and snow-off **ground** returns » ?
**A. Added the word "ground" to the two sentences noted.**

L264 : «**absolute** low bias ». In absolute yes, in relative no.
**A. Changed to "absolute low bias"**

L266 : « mean height profiles » does not seem right. The heights are not averaged, they are normalized. It is rather a distribution of the normalized elevation, or something along that.
**A. The text in section 3.2 was reworded to clarify:**
To provide insight to differences between the forest and field observations, height profiles of classified returns were calculated for 25 m$^2$ square regions centered on all forest (n=12) and field (n=7) study plots from lidar data. Height profiles were averaged for each site type, from here on referred to as mean height profiles (Figure 4).

L272 : bring « respectively » earlier in the sentence.
**A. Modified as recommended.**

L321 : confidence interval **decreases**
**A. Corrected.**

L346 : « the ability to capture the mean snow depth » I disagree: one-sided confidence interval still includes natural infra-pixel variability. You might perfectly capture the mean of a pixel with high infra- pixel variability and still have a large confidence interval.
**A. Point taken and we agree. The sentence now reads "In addition to the lidar point cloud density, the ability to narrow the confidence interval of the mean snow depth also depends on the ground surface variability within a cell as well as the lidar performance."**

L359-L363 : this paragraph is a bit confusing. It would benefit from i) defining better the standard deviation (see main comment 2. above) and ii) explain the relationship between point density per cell and cell size. The last sentence of the paragraph implies that reducing the cell size reduces the ground return density. This should be explained.

L359 : **point cloud density** instead of « the point per cell »
**A. Modified as recommended.**

L360 : **snow depth map resolution** instead of « spatial scales »
**A. Modified as recommended.**

L387 : « reduced accuracy of the GNSS » : in forest ?
**A. Modified as recommended.**

Paragraph from L410 : This paragraph is a bit hard to read. It should be simplified and the conclusions better highlighted. See some suggestions below.
L412 : « While this result is not entirely surprising » : little added value of this sentence. I suggest removing it.
**A. Modified as recommended.**

L414 : « in snow depth due to pockets of duff and woody debris, and due to higher variability in subnivean terrain in the forested areas of the study site » are duff and woody debris not part of the subnivean terrain ?
**A. Modified.**

L414 : is it really related to « variability in snow depth » ? or variability in snow-off terrain since this is the only thing discussed after.
**A. Modified as recommended.**

L416 : Long sentence. Consider shortening with « drive higher confidence interval **in these areas.** »

L416 : High relief terrain is defined by high elevation variability over short distances. Also, I am not sure what this sentence is stating.
**A. Modified. The sentence now reads "On steep slopes, there is more variability in ground return elevations over shorter distances, which would partially drive higher confidence intervals of ground surface elevation for pixels located in high relief areas." This is similar to the point made about L346.**

L418 : remove « of the study site »
**A. Modified as recommended.**

L419 : Interesting result. Please repeat which is higher than which (deciduous, coniferous, field). Cite literature showing the canopy interception effect. Is it not surprising that deciduous and coniferous trees have the same impact interception ?
**A. We have stated the mean snow depth earlier in the paper for each of the cover types. But in the section L 419, we believe the author is referring to on line 419, we are examining significant differences. We do find it interesting that coniferous regions had lower snow depth and could be likely to interception. What is additionally interesting is that the CI is higher under coniferous regions, highlighting the influence that lower ground returns due to intact canopy has on the estimate of snow depth. Figure 6 highlights the findings and statistical analysis. There is an evolving literature on forest canopy snow interception and impacts on snowpack distribution. However, most of the literature focuses on deciduous or coniferous forests rather than the mixed forests of this study region. The manuscript was modified to include references on snow interception as follows:**

This indicates the possible influence of tree canopies on snow accumulation due to enhanced snow interception in forests (see reviews in Clark et al., 2011), and particularly in conifer stands, but also could be the result of an under-sampled ground surface in forested areas relative to field areas. Despite challenges with sampling in the forest area, some degree of coherence for snow depth in the forest is apparent. The forest interception effects may be captured on average through forest structure parameters such as canopy closure and leaf area index that have traditionally used in snow models with canopy-snow interactions (see reviews in Snow model inter-comparison project – SNOWMIP2 by Essery et al., 2009; Rutter et al., 2009). However, the finer scale heterogeneity may benefit from additional parameters such as the mean distance to canopy and total gap area (Moeser et al., 2016) or modifications that reflect variations in canopy structure (Mazzotti et al., 2019).

Essery, R., Rutter, N., Pomeroy, J., Baxter, R., Stähli, M., Gustafsson, D., Barr, A., Bartlett, P., and Elder, K.: SNOWMIP2: An evaluation of forest snow process simulations, Bulletin of the American Meteorological Society, 90, 1120-1136, 2009.
Mazzotti, G., Currier, W. R., Deems, J. S., Pflug, J. M., Lundquist, J. D., and Jonas, T.: Revisiting Snow Cover Variability and Canopy Structure Within Forest Stands: Insights From Airborne Lidar Data, Water Resources Research, 55, 6198-6216, 2019.

Moeser, D., Mazzotti, G., Helbig, N., and Jonas, T.: Representing spatial variability of forest snow: Implementation of a new interception model, Water Resources Research, 52, 1208-1226, 2016.
Rutter, N., Essery, R., Pomeroy, J., Altimir, N., Andreadis, K., Baker, I., Barr, A., Bartlett, P., Boone, A., and Deng, H.: Evaluation of forest snow processes models (SnowMIP2), Journal of Geophysical Research: Atmospheres, 114, 2009.

L426, L440, you mention large UAV and heavy payloads. L372 the UAV seemed to be light (<25 kg) and small in size. Please homogenize.
**A. This is a reasonable point. These sections were rewritten to homogenize and to differentiate the high lift UAVs capable of carrying a lidar sensor package from a UAV that supports SfM.**

L463 : SD ?
**A. Modified to read "snow depth".**

L475 : please say what area is typically covered by airborne laser scanning campaign.

Figure 4 : In a and c a certain amount of points classified as non-ground is centered on 0 m height. This seems to be points which are wrongly classified. Is this assumption true ? Are they at specific locations ? Consider adding something about this in the results.
**A. The data presented in these figures is from a large footprint around each plot. Because the peak of non-ground returns near 0 m is actually slightly above 0 m height, it is possible that the returns at and around that peak are misclassified ground returns, returns from low-lying vegetation (which is especially prevalent below deciduous canopy at our study site), or some combination of both. This warrants further exploration and could be examined by calculating relative height profiles at finer scales closer to our DTM (~1m), but was outside of the scope of this study.**

Figure 6 a and b: there are some negative snow depth. This is possible due to the uncertainty of snow-on and snow-off DTM but should be presented in the results.
**A. The text was modified based on the reviewer's comment to read:**
Figure 6a also reveals that there are some negative snow depths in the two forest types that is due to the uncertainty of the snow-on and snow-off DTMs.

Figure 8. a.: See main comment about the standard deviation of lidar elevation. Here it seems to be a general metric : one standard deviation per snow depth map.
**A. Agreed. The reviewer makes the point that the text needs to be clearer regarding what is presented. Figure captions were reviewed to ensure clarity about when the individual cell confidence intervals are presented versus when and how those values are being aggregated.**

Anonymous Referee #2

*Thank you for your input and detailed comments. We have considered this reviewer's comments in light of the comments from the other three reviewers.*

This paper presents a snow depth map methodology for a UAV and LIDAR combination to measure thin snowpack of ephemeral snow. The study site contains a low vegetated field with a mixed forest which is useful to evaluate vegetation interaction. The main question focuses on the capability of LIDAR for snow depth mapping mounted on UAV because the sensor combination of LIDAR and UAV had not been extensively published yet. Most of the paper evaluates the accuracy of the map with respect to point cloud density (link to DTM resolution and LIDAR returns), vegetation cover and slope. A substantial amount of methodology and flight experience makes this paper focus on technical and methodological issues rather than informing us on snow processes of thin and ephemeral snow with vegetation interaction.

I do not think another methodological paper on snow depth mapping would be beneficial to the Cryosphere community. Differential snow depth mapping from LIDAR dates to the beginning of the century (Deems et al., 2006; Hopkinson et al., 2004) with airborne data from plane quickly became a more efficient tool to map large areas. A numerous numbers of article have used airborne LIDAR data for snow depth mapping (Currier et al., 2019; Grünewald et al., 2013; Hopkinson et al., 2012, 2004; Mazzotti et al., 2019; Nolan et al., 2015; Painter et al., 2016) with development on data processing (see review from Deems et al., 2013) and topographic and vegetation induced errors on LIDAR DEM (Spaete et al., 2011). Since the UAV platform only change the altitude and the coverage area, I do not see the point of having another methodological article. Regarding flight experience with UAV, a substantial amount of paper regarding snow mapping has already been published for smaller area with multi-rotor systems (Buhler et al., 2016; Cimoli et al., 2017; Fernandes et al., 2018) and larger area with fixed wing systems (De Michele et al., 2016; Harder et al., 2016; Redpath et al., 2018).

The high error in forested environment clearly need to be more investigated. It is stated that the magnaprobe measurements overestimate the snow depth in forested environment by penetrating low lying vegetation and soil. So, is the LIDAR mean depth of 6cm in more representative than magnaprobe (15 cm) and federal sampling tube (13 cm) which both showed larger depth for forested than field? Some forested areas can have deeper snowpack (Trujillo et al., 2009) due to wind reduction but snow interception by canopy will decrease snow accumulation on the ground as forest cover increases (Varhola et al., 2010). This needs to be sorted out especially when it is know that differential snow depth mapping underestimate snow for small vegetation as the snow off DEM is higher than the bare ground but when snow is measured for truth scene, the same vegetation is compressed by the weight of the snow (Buhler et al., 2016; Nolan et al., 2015).

The paper lacks a novel or in depth analysis of relations or processes regarding snow derived from a snow map that would be beneficial to the community like for e.g. a comparison with SfM photogrammetry in forested environments (Harder et al., 2020),

statistical analysis of snow depth (Grünewald et al., 2013; Wainwright et al., 2017) or spatial analysis with variogram or fractal analysis between open field and forested environments (Deems et al., 2006; Redpath et al., 2018; Trujillo et al., 2009). To do this, I would suggest trying to improve with additional dataset over wider areas, only one site is not enough or perhaps different seasons to explore temporality and resubmit to this journal after next field season or go with this version towards a technical journal regarding UAV.

**A. The author team posits that there are significant potential advances in snow hydrology as drones and lidar become more cost-effective, easier to use, and safer to fly. While there is excellent work using optical sensors on a UAV and the airborne lidar work (e.g., references by the reviewer), the use of lidar on a UAV platform is not trivial. Thus, before analyses of snow depth relations or processes can occur, the first challenge is obtaining research grade lidar data. The previous revision modified the submission to support the target audience seeking to collect UAS lidar data for snow hydrology. In addition, a more extensive discussion of the contribution appears in the previous response to reviewer #2.**

**Based on the reviewer's comments regarding observations in the forested environment, the manuscript has been modified to include a more extensive discussion of forest canopy and snow interactions as well as lidar performance.**

**We also agree with the reviewer's comments that "The high error in forested environment clearly need to be more investigated." As new remote sensing platforms lead to higher resolution snow depth measurements, the ability to conduct in situ ground sampling at resolutions needed to validate remote sensing observations will need to be sorted out. We are currently conducting a new study that focuses on sorting out the issues discussed by the reviewer.**

Anonymous Referee #3

*Thank you for the original review and the recommendations for minor revisions. We have provided detailed responses to the reviewer in **bold** following each of the reviewer's comments.*

In my original review, I commented on the limited field data used to produce the paper. I appreciate the effort the authors went to place their study in context (their Table R1), and while I would like to have seen their analysis encompass far more results, I grant that the paper as now re-written has a distinct and useful purpose related to pushing new technology ahead. I particularly found the discussion of UAS/lidar use and limitations quite helpful. The discussion of forest and slope errors is also far more mature and comprehensive. I recommend publication with some minor revisions.

• GENERAL: There are some inconsistencies between how things are discussed in the text and how they are labeled in the figures. The authors should go through and try to make these consistent. For example, cell size vs. DTM resolution, and I think terrain slope vs. terrain variability.
**A. All figure captions and axes were reviewed. Figures 2a, 7b, and 8a x-axes were changed to "Cell Size (m)". The y-axis in figure 8a was changed to "Cell Stdev (cm)". The terrain slope was clarified throughout the manuscript. The term "terrain variability" was modified to "ground surface variability" where appropriate in the text.**

• In the abstract the authors say that in the forest there was " ....modestly reduced performance", but the degradation is 4x. Delete "Modest".
**A. Modified as recommended.**

• Perhaps it is that I have been doing research and spent 9 years as a sub- and chief editor of a journal, but I still found the introduction longer than need be. Anyone reading this paper is likely to be deeply involved in snow research and knows snow changes a lot, that our model are good but drift without real data to guide them, etc. I suggest just getting to it in the introduction: delete the first 3 paragraphs of the introduction....or at least make the point we need high quality depth data more concisely.
**A. After reviewing the introduction, the authors agree. The first three paragraphs were removed.**

• Line 78-80: This sentence is run-on and awkward.
**A. Also noted by reviewer #1. This sentence was rewritten.**

• Line 87: My understanding is that now DGPS level positioning on UAS's removes the need for ground control points. If that is true, perhaps moderate this statement.
**A. There is an active, informal group of snow hydrologists who meet routinely to discuss SfM for SD mapping (led by Dan McGrath, Colorado State University). To date, that group's consensus is that despite the DGPS available on the newer and more expensive DJI systems, at least two to three ground control points are still**

**needed. The authors have chosen to keep the sentence.**

• Figure 1: Much better and very useful and handsome.
**A. Thank you.**

• Figure 2: Last word (bottom). Not sure what this means.
**A. The words in parentheses should have been "left" and "right". Because the figures are labeled as "(a)" and "(b)", these references were removed.**

• Section 3.2: Well done.
**A. Thank you.**

• Line 309: Compare this to the statement in the Abstract.
**A. The abstract was modified to match the referenced text.**

• Figure 5: Great figure now even better.
**A. Thank you.**

• Line 351: Insert "the"
**A. Modified as recommended.**

Anonymous Referee #4

*Thank you for the original review and the recommendations for minor revisions. We have provided detailed responses to the reviewer in **bold** following each of the reviewer's comments.*

This is an interesting paper on using UAS based lidar map snow depth. This is an emerging tool that will undoubtedly advance observation of fine scale snow distributions and hopefully understanding of small –scale snow processes ultimately. Overall this paper provides a detailed description of a campaign in 2018/2019 which is important to understand the accuracy of this instrument and operational, considerations. I do have some concerns in terms of its placement in the context of the current literature, the communication of some of the accuracy results, and some of the more technical aspects of the writing. I would consider these minor revisions and once resolved look forward to its publication! I will summaries some main points that need addressing followed by some technical notes.

Literature context. The introduction could be made more punchy and to the point. Large scale context of snow is important but we're only looking at how to measure it at a fine scale in this paper so making it more concise on the progression in instrument techniques that have been employed would be an improvement.
**A. After reviewing the introduction, the authors agree with this reviewer and reviewer #3. The first three paragraphs were removed and the paper now starts with the instrument techniques.**

The Harder et al 2020 paper is cited as the first example of a UAS-lidar snow depth study. This is a clear comparable study but besides being mentioned nothing is else is mentioned about it in the intro. What are the common comparison points that can be used to relate the two studies? What differentiates the two (ie. Sensor quality/cost to get comparable results as this is labeled as a modest cost system (define?) versus Harder et al who use a higher cost Riegl system)?
**A. Thank you for this comment. The Harder et al. (2020) paper was published only a few days before our revised article was submitted last summer. Based on this reviewer's and reviewer #1's comments, additional comparisons between this study and Harder et al. (2020) were added. We appreciate the opportunity to now revise our manuscript to include the requested comparison.**

**The introduction was revised as follows:**
Harder et al. (2020) compared snow depth estimates between lidar versus SfM techniques using in-situ snow depth observations in mountain and prairie environments, focusing on sub-canopy snow, which has been a challenge to measure in the snow remote sensing community. Using a considerably more expensive UAS-based lidar system (~$300K Canadian), they found that the lidar system tends to have lower errors than the SfM to capture sub-canopy snow distributions at moderate depth of snowpack (up to 2 m and 1 m of the maximum depth for mountain and prairie areas, respectively). In this study, we assess the ability of a more modest cost UAS lidar system (~$70K U.S. dollars) to map

snow depth focusing on shallow and ephemeral snowpack (< 20 cm).

**We also add a comparison of this study's results to their results in the discussion section as follows:**
This study's lidar snow depth performance metrics are comparable to those from the more extensive lidar surveys made Harder et al. (2020), In the field, our snow depth errors, 1 cm bias and 1.2 cm RMSD, were modestly better than those from their open sites snow depth 3 cm bias and RMSE values on the order of 10 cm. In the forest, our snow depth errors, 7 cm bias and 10 cm RMSD, were also modestly lower than those from their forest sites 9 to 13 cm bias and 15 cm RMSE. While it is difficult to make direct comparison across different study sites, snow conditions, and ground validation approaches, these early findings indicate that UAS lidar has the capability of mapping snow depths in open and forested regions and has improved performance as compared to previous SfM results particularly for vegetated surface. It is also noteworthy that this study's mapping was conducted using the Velodyne Puck series, a laser scanner adapted from the assisted and autonomous vehicle applications, rather than the specialized Riegl miniVUX-1UAV used by Harder et al. (2020) resulting in a complete mapping system that was approximately one-third the costs of their Riegl system.

Accuracy results: There are many ways to present the results of this works but a common approach in much of the UAS-SFM, (and lidar), ALS and TLS is to consider the RMSE and Bias metrics prominently and these are reported in the literature review. This paper does present these values in Figure 3 and the first paragraph of section 3.2. The metrics reported RSMD of 1.22cm and 10.5 cm in open field and forest respectively are quite comparable to other work but these connections are not made. Rather only the 1.22 cm open field RMSE is reported in the abstract – no mention of the larger forest error or anything in the conclusion. In lieu of direct comparison with insitu data there is a focus on the one sided confidence intervals that are reported at < 4cm. Confidence intervals can be important to isolating areas uncertainty but are not fully suitable to assess the skill/accuracy of the product. If the results could be clarified to emphasize the RMSD and MAD more clearly and the meaning/role of the confidence intervals to be clarified more strongly would make this better.

Technical writing Writing is good throughout but there are some sections and word choices that need work – specific examples given below but not exhaustive. Specificity of word section and consistency throughout will make things much clearer.

Technical edits
Line 33:"differentially" – word choice. Awkward with "differences" earlier in sentence
**A. The line was in an introductory paragraph that was removed when the introduction was modified.**

Line 48: "High vertical resolution snow mapping" word choice. Perhaps "Mapping snow with high spatial resolutions and vertical precisions...."
**A. The line was in an introductory paragraph that was removed when the**

**introduction was modified.**

Line 51: "Despite importance" -> "Despite the importance"
**A. The line was in an introductory paragraph that was removed when the introduction was modified.**

Line 54-56: " Using traditional, precise point measurements with a limited sample size, the experimental design requires a balance between the sampling extent and sample spacing (Clark et al. 2011). >: " Using a traditional experimental design with precise point measurements of a limited sample size requires a balance between sampling extent and sample spacing"
**A. The line was in an introductory paragraph that was removed when the introduction was modified.**

Throughout: Are we using UAV or UAS? Title and intro start with UAV and then it switches to UAS at line 82.
**A. Jennifer et al. We use UAV to indicate the vehicle and UAS to indicate the combination of the UAV and the sensors. The manuscript was reviewed for its use of UAV and UAS to be consistent with those definitions.**

Line 140-146: I see there were batteries changes in this mapping mission. How long does it take to survey this area? Important information to present and perhaps bring into discussion at end on how scalable this technology is with actual times/areas to benchmark a discussion.
**A. Good point.  The 2 hr mapping time as needed to address UAS lidar scalability was added to section 4.3.**

Line 151: "antiparallel" = perpendicular?
**A. Antiparallel means the same line, but in the opposite direction. The sentence was modified to read "Boresighting calibration was performed using returns from the first two parallel flight lines that were collected in opposite directions (i.e., antiparallel)."**

Line 152-154: The boreshighting and roll-pitch offsets need to be done for each flight? Or are these done once and they are stable thereafter?
**A. The boreshighting and roll-pitch offsets were done once for the entire survey.**

Line 155-156: What and how are points filtered?
**A. From unclassified, georeferenced point clouds, non-ground returns are filtered out of the data set using the Progressive Morphological Filter. The resulting data set contains only ground surface returns. The ground returns were coded according to LAS specifications, then merged with the non-ground returns. We have attempted to clarify in the text:**
Briefly, the PMF operates iteratively on sets of two parameters, window size and elevation thresholds to erode and dilate point cloud data sets to estimate surface topography. The result of the PMF is that non-ground returns (i.e., trees, shrubs, and

noise) are filtered out of point cloud data sets, so that only returns from ground surfaces remain. The two data sets, non-ground returns and ground returns from the original point cloud, are coded according to LAS specifications and merged. For a full explanation of PMF, see Zhang et al., (2003).

Line 174-175: (Figure 1) at end of sentence implies that it contains a methodology which it does not and is a little confusing.
**A. The text in the referenced sentence "(Figure 1)" was removed.**

Line 173-184: This section could be tightened up. Landscape classes where derived from CHM and simply > 15m was coniferous, deciduous was < 15m and field was what? 0m?
**A. We thank the reviewer for suggesting to tighten this section up. We tried to make our methodology a little clearer by editing the section. The section now reads as follows:**
The snow-off DTM was used to develop a 1 m resolution map of slope (Horn, 1981). Vegetation cover type (field/forest) was determined from optical imagery. A Canopy Height Model (CHM) was developed by subtracting the DTM produced using ground-classified points from the DSM produced using all lidar points. This results in a digital model consisting solely of canopy heights with no topography. The CHM generation used raster images with a 1 m resolution. The forested area was further classified as coniferous or deciduous for the study region. Within the forested area, the CHM was used to distinguish the upper canopy that did not lose needles/foliage from other forested regions with trees with no leaves using our snow-off survey that was collected with leaf off in the spring. A 3 by 3 maximum convolve filter was used to enhance the edges of canopy crowns and expand smaller regions that might have just one pixel of an intact canopy or a hole in a larger canopy (Palace et al., 2008). A 15 m threshold was used to differentiate between the upper level intact coniferous canopy and canopies that had lost their leaves. CHM pixels that were below this threshold were deemed deciduous canopies (see Figure S3 in supporting information for intermediate figure). The 5.6 ha forested area has a forest type that is 65% deciduous and 35% coniferous. "

Line 211-217: Is this section necessary? At this stage of snow depth accuracy evaluation do we even need to know that the soil was frozen?
**A. This section was added in response to an earlier reviewer who requested ancillary data on soil frost conditions to support the discussion of the magnaprobe. This section was retained.**

Line 227-233: I found this section/ equation/term definitions to be confusing and incomplete.
**A. Thank you for pointing out. Reviewer #1 has also gave specific suggestions regarding this section and terminology. Modifications were also made in response to Reviewer #1. The paragraph now reads as follows:**

The one-sided width of the 95% confidence limits ($CI_{95\%,+/-}$) for each grid cell's lidar derived estimate of the mean snow depth is a measure of uncertainty. The $CI_{95\%,+/-}$

values are used to compare the reliability of the snow depth estimates among cells. The $CI_{95\%,+/-}$ values were calculated using each grid cell's bare-earth and snow-on pooled sample standard deviation ($s_d$) and the number of bare-earth and snow-on lidar returns (n and m respectively) (Helsel and Hirsh, 2002).

$$CI_{95\%+/-} = t_{crit}s_d\sqrt{\left(\frac{1}{n}+\frac{1}{m}\right)} \tag{1}$$

A cell's pooled sample standard deviation ($s_d$) was calculated as

$$s_d = \sqrt{\frac{(n-1)s_{off}^2+(m-1)s_{on}^2}{(n+m-2)}} \tag{2}$$

where $s_{on}$ and $s_{off}$ are the standard deviations of the snow-on and snow-off lidar ground return elevations, respectively. The $s_n$ and $s_{off}$ values are a measure of the grid cell variability. This variability depends on the lidar instrument's relative accuracy (Maune and Nayegandhi, 2018), which includes intra-swatch accuracy (i.e., precision or repeatability of measurements) and inter-swath accuracy (i.e., differences in elevations between overlapping swaths), as well as surface elevation variations and terrain induced errors (Deems et al., 2013). The contribution from the individual sources of variability was not assessed in the current study.

Figure 7: seems low res in my pdf.
**A. Figure 7 was replaced.**

Line 380: surveying in locations prior to snow-on (and presumable mark with a stake?) risks modifying snow processes at that locations and poetically biasing the snowpack which may have relevance certain processes being studied, and risks destructive sampling impacts if same location is being repeatedly visited over a season.
**A. The author team agrees entirely with these comments. The team has had numerous internal and external discussions with other UAV researchers about sampling strategies that minimize the impacts, yet allow validation to occur. This is a challenge for validation high resolution snowpack measurements. To elucidate the downside of a priori surveying, the following section was rewritten to read "Surveying and marking sample locations prior to the winter season might reduce this effort. However, the use of sampling stakes risks modifying snow processes at the sample locations and potentially biasing the snowpack and incurring destructive sampling impacts if same location is being repeatedly visited over a season."**

Line 411-412: Confidence intervals for snow depth are high on slopes? What slope angles are we seeing this relationship? In the literature snow depth errors tend to be higher on slopes – see terrain induced errors section in Deems 2013 Deems, Jeffrey S., Thomas H. Painter, and David C. Finnegan. "Lidar measurement of snow depth: a review." Journal of Glaciology 59.215 (2013): 467-479.
**A. Yes, the confidence intervals were larger on slopes within a single cell. Slopes that exceeded 20$^o$ had a notable increase in their confidence intervals. While there was limited very steep terrain, there were a few large boulders in the forest. A reference**

**to Deems et al. (2013) discussion of terrain induced errors.**

Line 416: "higher" word choice- "larger" may be better
**A. Modified as recommended.**

Line 453: "antiparallel" = "perpendicular"
**A. Antiparallel is now defined in section 2.2, so this later language should now be clear.**

[revised manuscript text omitted]

Jennifer Jacobs 1/18/2021 5:18 PM

Jennifer Jacobs 1/18/2021 5:18 PM

Liz Burakowski 1/22/2021 2:56 PM
Comment [1]: accuracies … are

Liz Burakowski 1/22/2021 2:56 PM

Jennifer Jacobs 1/18/2021 5:18 PM

Jennifer Jacobs 1/18/2021 5:18 PM

Jennifer Jacobs 1/19/2021 10:46 AM

Jennifer Jacobs 1/19/2021 10:47 AM

Jennifer Jacobs 1/24/2021 9:59 AM

Jennifer Jacobs 1/24/2021 9:57 AM

Jennifer Jacobs 1/24/2021 9:57 AM

Jennifer Jacobs 1/24/2021 9:57 AM

Jennifer Jacobs 1/24/2021 9:48 AM

Jennifer Jacobs 1/24/2021 10:00 AM

Jennifer Jacobs 1/24/2021 10:06 AM

Jennifer Jacobs 1/24/2021 10:06 AM

Eunsang Cho 1/22/2021 12:31 AM

Jennifer Jacobs 1/24/2021 10:22 AM

Jennifer Jacobs 1/24/2021 6:19 PM

Jennifer Jacobs 1/24/2021 10:06 AM

Jennifer Jacobs 1/24/2021 10:22 AM

Jennifer Jacobs 1/24/2021 6:19 PM

Jennifer Jacobs 1/24/2021 6:20 PM

Eunsang Cho 1/22/2021 12:35 AM

Jennifer Jacobs 1/19/2021 11:03 AM

Eunsang Cho 1/22/2021 12:31 AM

[revised manuscript text omitted]